# Southern Ocean Cloud and Aerosol data: a compilation of measurements from the 2018 Southern Ocean Ross Sea Marine Ecosystems and Environment voyage

Stefanie Kremser[1], Mike Harvey[2], Peter Kuma[3,11], Sean Hartery[3], Alexia Saint-Macary[2,10], John McGregor[2], Alex Schuddeboom[3], Marc von Hobe[5], Sinikka T. Lennartz[6], Alex Geddes[4], Richard Querel[4], Adrian McDonald[3], Maija Peltola[7], Karine Sellegri[7], Israel Silber[9], Cliff S. Law[2,10], Connor J. Flynn[8], Andrew Marriner[2], Thomas C. J. Hill[12], Paul J. DeMott[12], Carson C. Hume[12], Graeme Plank[3], Geoffrey Graham[3], and Simon Parsons[3]

[1]Bodeker Scientific, Alexandra, New Zealand
[2]National Institute of Water and Atmospheric Research (NIWA), Wellington, New Zealand
[3]University of Canterbury, Christchurch, New Zealand
[4]National Institute of Water and Atmospheric Research (NIWA), Lauder, New Zealand
[5]Institute for Energy and Climate Research (IEK-7), Forschungszentrum Jülich GmbH, Jülich, Germany
[6]Institute for Chemistry and Biology of the Marine Environment, University of Oldenburg, Oldenburg, Germany
[7]Université Clermont Auvergne, CNRS, LaMP, Clermont-Ferrand, France
[8]School of Meteorology, University of Oklahoma, Norman, USA
[9]Department of Meteorology and Atmospheric Science, Pennsylvania State University, University Park, USA
[10]Department of Marine Science, University of Otago, Dunedin, New Zealand
[11]Peter Kuma Software & Science, Christchurch, New Zealand
[12]Department of Atmospheric Science, Colorado State University, Fort Collins, USA

**Correspondence:** Stefanie Kremser (stefanie@bodekerscientific.com)

**Abstract.** Due to its remote location and extreme weather conditions, atmospheric in situ measurements are rare in the Southern Ocean. As a result, aerosol–cloud interactions in this region are poorly understood and remain a major source of uncertainty in climate models. This, in turn, contributes substantially to persistent biases in climate model simulations such as the well known positive short-wave radiation bias at the surface, as well as biases in numerical weather prediction models and reanalyses. It has been shown in previous studies that in situ and ground-based remote sensing measurements across the Southern Ocean are critical for complementing satellite data sets due to the importance of boundary layer and low-level cloud processes. These processes are poorly sampled by satellite-based measurements and are often obscured by multiple overlying cloud layers. Satellite measurements also do not constrain the aerosol-cloud processes very well with imprecise estimation of cloud condensation nuclei. In this work we present a comprehensive set of ship-based aerosol and meteorological observations collected on the TAN1802 voyage of RV *Tangaroa* across the Southern Ocean, from Wellington, New Zealand, to the Ross Sea, Antarctica. The voyage was carried out from 8 February to 21 March, 2018. Many distinct, but contemporaneous, data sets were collected throughout the voyage. The compiled data sets include measurements from a range of instruments, such as (i) meteorological conditions at the sea surface and profile measurements; (ii) the size and concentration of particles; (iii) trace gases dissolved in the ocean surface such as dimethyl sulfide and carbonyl sulfide; (iv) and remotely sensed observations

of low clouds. Here, we describe the voyage, the instruments, data processing, and provide a brief overview of some of the data products available. We encourage the scientific community to use these measurements for further analysis and model evaluation studies, in particular, for studies of Southern Ocean clouds, aerosol and their interaction. The data sets presented in this study are publicly available at https://doi.org/10.5281/zenodo.4060237 (Kremser et al., 2020).

# Contents

# 1 Introduction

The Southern Ocean is the cloudiest region on Earth and is also distant from major anthropogenic sources of aerosol (Haynes et al., 2011). This makes the Southern Ocean an ideal environment for studying aerosol–cloud interactions (Krüger and Graßl, 2011; Fossum et al., 2018; Hamilton et al., 2014) and the role of marine aerosol in the radiation budget. The contribution of marine aerosol to Earth's radiation budget is both direct through aerosol scattering and absorption, and indirect via cloud droplet activation and their subsequent influences on cloud radiative processes (Murphy et al., 1998; Mulcahy et al., 2008; McCoy et al., 2015; Fossum et al., 2018). Marine aerosol can be classified as primary or secondary in origin (Fossum et al., 2018). Primary aerosols, such as sea spray, are directly injected into the atmosphere when breaking waves entrain air bubbles into the ocean surface, which subsequently form whitecaps and burst (Hultin et al., 2010; Salter et al., 2014). Secondary aerosols, such as sulfate aerosols, are formed from the nucleation of sulfur-containing gases in a gas-to-particle conversion process. One of the main precursors of sulfate aerosol in the marine environment is dimethyl sulfide (DMS), a by-product of an enzymatic compound produced within phytoplankton (dimethylsulfoniopropionate, DMSP; Read et al., 2008; Fossum et al., 2018). DMS is the main natural source of atmospheric sulfur, with a global average of 28.1 Tg of sulfur being emitted annually from the oceans into the atmosphere in the form of DMS (Lana et al., 2011). When DMS is emitted into the atmosphere, it undergoes a series of chemical reactions to form sulfur dioxide ($SO_2$), resulting in a typical lifetime of DMS in the atmosphere of 1–2 days (e.g. Chen et al., 2018). The $SO_2$ can then be further oxidised to form sulfuric acid, sulfate aerosol and methanesulfonic acid (MSA; e.g. Yan et al., 2020). Aerosol emitted into the atmosphere can grow in size via condensation and coagulation. The ability of any aerosol particle to serve as a nucleus for water droplet formation depends on its size, chemical composition, the local supersaturation, and meteorological conditions such as the cloud base updraft velocity (Rosenfeld et al., 2014). Aerosol has a significantly different impact on cloud formation and evolution, depending on whether it acts as an ice nucleating particle (INP), a cloud condensation nuclei (CCN), or both.

Despite their significant influence on climate, clouds still represent the largest source of uncertainty in modern climate models with aerosol–cloud interactions being a major factor in this uncertainty (Myhre et al., 2013; Haynes et al., 2011). For example, Hyder et al. (2018) recently identified that 70 % of the sea surface temperature biases observed in model simulations, performed in support of the Coupled Model Intercomparison Project 5 (CMIP5), can be attributed to the models not representing clouds and their properties correctly. These errors occur because climate models simulate too little cloud cover and contain biases in cloud albedo over the Southern Ocean (Bodas-Salcedo et al., 2012; Schuddeboom et al., 2019), resulting in projections that underestimate the reflected solar radiation at the top of the atmosphere (TOA; Haynes et al., 2011) and overestimate downwelling solar radiation at the ocean surface. This leads to excessive sunlight being absorbed by the ocean (Trenberth and Fasullo, 2010; Kay et al., 2016; Hyder et al., 2018) and subsequent higher sea surface temperatures than observed (Bodas-Salcedo et al., 2012; Mechoso et al., 2016). Previous studies have also shown the importance of accurate mixed-phase cloud parameterisations over the Southern Ocean in climate models to properly simulate cloud radiative properties over the Southern Ocean (Lawson and Gettelman, 2014; Kay et al., 2016; Schuddeboom et al., 2019; Noh et al., 2019). In mixed-phase clouds, both liquid droplets and ice crystals coexist with the liquid water often being supercooled. While observations in the Southern

Ocean are sparse, measurements reported by McCluskey et al. (2018), DeMott et al. (2018), and Welti et al. (2020) indicate that INP concentrations are exceptionally low over the Southern Ocean, much lower than previously estimated by Bigg (1973). The low concentrations of INPs over the Southern Ocean limit cloud droplet freezing, reduce precipitation, and enhance cloud reflectivity compared to regions of higher INP abundance (e.g. Vergara-Temprado et al., 2018; Vignon et al., 2020). This indicates that an accurate representation of INPs in climate models is necessary to properly simulate cloud radiative properties over the Southern Ocean. For example, climate models often produce too many ice crystals in mixed phase clouds that consume the liquid droplets and thereby change the radiative properties of clouds (Kay et al., 2016). Furthermore, due to the low INP concentrations identified to exist broadly over the Southern Ocean, there is a need to better understand secondary ice formation processes and their dependence on INP concentration and to improve their representation in climate models. Observations support the embedded occurrence of a variety of secondary ice formation processes in clouds over the Southern Ocean, which are otherwise dominated by supercooled water. These processes range from association with seeding of ice crystals from colder cloud levels that appears consistent with a rime splintering process (Finlon et al., 2020) to studies suggesting the vital importance of ice production via breakup following ice-ice collisions (Sotiropoulou et al., 2021; Young et al., 2019).

Reducing the uncertainty in the simulation of aerosol–cloud interactions requires detailed observational data sets against which models can be evaluated. However, this process is hindered over the Southern Ocean by the lack of ground-based and in situ measurements. While satellite-based measurements can provide some data over the region, they cannot provide detailed aerosol chemical composition data or be solely relied upon to examine low-level clouds (Kuma et al., 2020b; McErlich et al., 2020). There have been only a limited number of ship- and ground-based field campaigns over the Southern Ocean (see Table 1 for an overview). Observational campaigns which provide detailed measurements of low-level cloud, aerosol, aerosol precursors, INPs, and CCN are essential for model evaluation, especially for parameters that can be indirectly estimated, but not accurately determined from satellite-based measurements.

In this paper we present a new data set of atmospheric (cloud, aerosol and thermodynamic properties) and seawater measurements that were collected during the six-week Southern Ocean Ross Sea Marine Ecosystem and Environment voyage (TAN1802) from Wellington, New Zealand, to the Ross Sea, Antarctica, in 2018. The data sets presented here are publicly available at https://doi.org/10.5281/zenodo.4060237 (Kremser et al., 2020). Given the sparsity of data in the Southern Ocean region, this data set provides a valuable collection of atmospheric and underway measurements that can be used to better understand aerosol–cloud processes over the Southern Ocean. This paper includes a description of DMS and carbonyl sulfide (OCS) measurements as previous work has identified that DMS plays an important role as a sulfate aerosol precursor. Furthermore, DMS concentrations have a particularly large impact on model aerosol forcings, yet are poorly represented in climate models (Hoffmann et al., 2016; Bodas-Salcedo et al., 2019). Although not strictly related to aerosol–cloud interactions, OCS is a greenhouse gas and an important source of stratospheric sulfate aerosol (Crutzen, 1976; Brühl et al., 2012; Kremser et al., 2016). Ocean emissions represent the largest known single OCS source, and process models predict that the highest open ocean OCS fluxes occur in the Southern Ocean (Kettle et al., 2002; Lennartz et al., 2017). As the TAN1802 voyage was only the second research cruise probing OCS in the Southern Ocean (the first one is described in Staubes and Georgii (1993)), and the first with

**Table 1.** List of previous ship- and ground-based field campaigns related to aerosol–cloud interactions over the Southern Ocean.

| Campaign name | Year | Reference |
|---|---|---|
| British Southern Ocean cruise (BSO) | Oct 1992–Jan 1993 | O'Dowd et al. (1997) |
| Aerosol Characterization Experiment (ACE I) | Nov–Dec 1995 | Bates et al. (1998) |
| Finnish Antarctic Research Program (FINNARP) | Nov–Dec 2004 | Vana et al. (2007) |
| Surface Ocean Aerosol Production (SOAP) | Feb–Mar 2012 | Law et al. (2017) |
| Sea Ice Physics and Ecosystem Experiment (SIPEX II) | Sep–Nov 2012 | Humphries et al. (2016) |
| PEGASO voyage of RV BIO Hesperides | Jan–Feb 2015 | Fossum et al. (2018) |
| RV Investigator trial voyage into the Southern Ocean | Jan–Feb 2015 | Alroe et al. (2020) |
| Clouds, Aerosols, Precipitation, Radiation, and atmospheric Composition Over the southeRn oceaN (CAPRICORN I and II) | Mar 2015 Mar–Apr 2016 | Protat et al. (2017) Mace and Protat (2018) |
| Antarctic Circumnavigation Expedition (ACE 2016/17) | Dec 2016–Mar 2017 | Schmale et al. (2019) |
| Chinese Antarctic Research voyages by RV Xuelong | Dec 2017 Jan–Feb 2018 | Yan et al. (2020) |
| Measurements of Aerosols, Radiation, and Clouds over the Southern Ocean (MARCUS) | Oct 2017–April 2018 | McFarquhar et al. (2019) |
| Macquarie Island Cloud and Radiation Experiment field campaign (MICRE) | Mar 2016–Mar 2018 | Marchand (2020) |
| The Southern Ocean Ross Sea Marine Ecosystems and Environment voyage | Feb 2018–Mar 2018 | this work |

sufficiently high temporal resolution to thoroughly test and improve the existing models, we include the OCS measurements in the data set accompanying this paper.

## 2   TAN1802 voyage – New Zealand to the Ross Sea

The 2018 Southern Ocean Ross Sea Marine Ecosystem and Environment voyage, TAN1802, was a voyage with NIWA's (National Institute of Water and Atmospheric Research) research vessel *Tangaroa* from Wellington to the Ross Sea, between 8 February and 21 March 2018. The purpose-built research vessel is 70 m long, with a beam width of 13.8 m and a draught of 7 m. It can accommodate 40 people, including a mix of research staff and ship personnel. The specifications and principal features of the vessel are described at the *NIWA website*. Over the course of the TAN1802 voyage, the RV *Tangaroa* travelled 11,000 km and spent the majority of its time, i.e. 30 days, south of 60° S (Fig. 1). The focus of this research campaign was

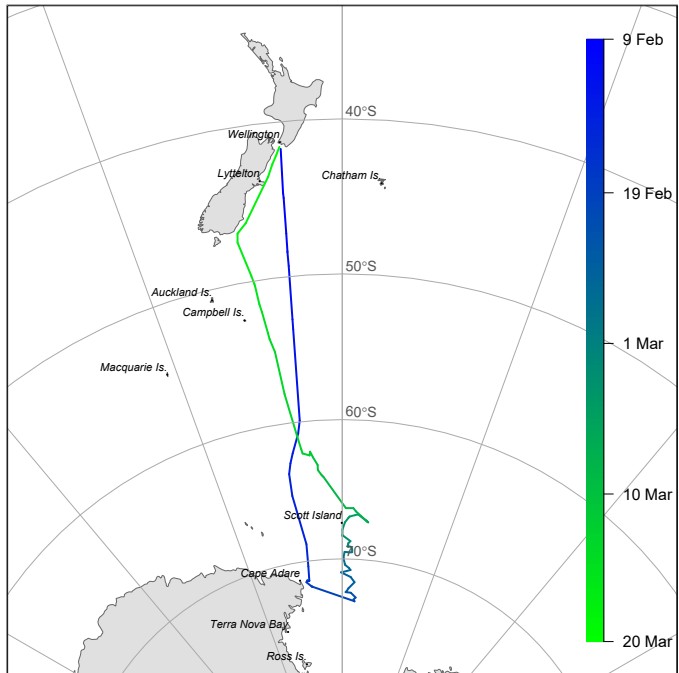

**Figure 1.** The ship track of the RV *Tangaroa* with dates indicated by colours. The 2018 Southern Ocean Ross Sea Marine Ecosystem and Environment voyage extended from the North Island of New Zealand to off the coast of Cape Adare (Antarctica).

to conduct measurements in the Southern Ocean, which is commonly defined to be south of 60° S latitude and encircling Antarctica.

**2.1 Voyage objectives**

One of the seven overarching research objectives of the TAN1802 voyage was to take atmospheric observations and samples to investigate interactions between marine aerosols and cloud formation over the Southern Ocean, thereby improving our understanding of aerosol–cloud interactions in this region. This study focuses on describing the ship-based measurements that were made in support of this one research objective, i.e. 'Aerosol-cloud interaction', with the following underlying research aims:

1. Characterise low level ($< 2\,\mathrm{km}$) and middle level ($2$ to $4\,\mathrm{km}$) clouds, aerosol, and radiation from ship-based continuous measurements using lidar, ceilometer, sky cameras, radiosondes, an automatic weather station, and radiometers.

2. Characterise aerosol sources which have a controlling influence on cloud properties through measurements of size, chemistry, and nucleating properties.

3. Investigate the importance of sea-salt and other primary aerosols as CCN.

4. Investigate the influence of local biogenic sulfur emissions to secondary aerosol abundance.

5. Measure boundary layer profiles of aerosol and thermodynamic properties through combination of lidar measurements and radiosonde flights and evaluate coupling between surface measured aerosol and low-level cloud capping within the marine boundary layer.

6. Link aerosol and surface trace gas properties to surface water biogeochemistry.

All measurements that were made to address these six research aims are summarised in Table 2a and 2b and a detailed description of the instrumentation and their measurements is given below.

## 2.2 Meteorological measurements and metadata

Meteorological measurements, including 1 minute records of air temperature, dew-point temperature, pressure, wind speed, wind direction, relative humidity (RH), sea surface temperature, downwelling shortwave and downwelling infrared radiation were made during the voyage using underway sensors and the automated weather station (AWS) installed on the ship. The vessel reports meteorological and oceanographic data through the Integrated Marine Observing System (IMOS, Smith et al., 2018). The AWS anemometer was positioned at 25.2 m above sea level (ASL) on the mast of the ship, while the other parts of the AWS were positioned at 15 m ASL. Measurements of the average relative wind speed and wind direction were made using a pair of ultrasonic anemometers (Gill WindSonic), reporting at 1 minute intervals. The Tangaroa Data Acquisition System (DAS) recorded Global Positioning System (GPS) coordinates, all AWS measurements, ship's hull sensor measurements, and other variables such as attitude (pitch and roll) every minute.

Data stored in the DAS were further processed as indicated in Fig. 2, before their use in, for example, sea-air flux calculations for DMS and OCS (Sect. 4.3.1 and Sect. 4.3.2). The *Tangaroa* DAS provides the true wind speed and direction based on vector correction of the measured wind speed. Directionally dependent speed-up factors for windflow over the ship have been characterised according to Popinet et al. (2004) (see Figure 17b and discussion in Popinet et al. (2004)) and incoming wind speed has been corrected through a look up table of azimuthally dependent speed-up correction factors. The true wind speed at the vessel anemometer height of 25.2 m was further corrected to the standard 10 metre value (u10) from the micrometeorological wind profile calculated by the COARE V3.6 algorithm (Fairall et al., 2003).

A pair of shortwave radiometers (0.285 to 2.8 μm - Eppley Precision Spectral Pyranometer, PSP) and a second pair of downwelling infrared radiometers (4 to 50 μm - Eppley Lab Precision Infrared radiometer, PIR) are installed on the ship. The pairing of the instruments enabled corrections to the measurements to be made by accounting for shadowing by the ship. Salinity was calculated using the SBE 21 Seacat thermosalinograph (Seabird Electronics Inc, WA, USA) conductivity and temperature measurements. The instrument is installed as part of the underway seawater measurement suite, with the surface water intake at a depth of about 7 m below the surface on the mid port side of the vessel. The salinity measurements have been periodically calibrated against salinity as determined from CTD (conductivity, temperature and depth sensor) measurements and have an estimated accuracy of 0.05‰. The "hull" temperature of the ship is measured by a remote SBE 38 thermometer (Seabird Electronics) located close to a pumped seawater intake near the bow, with a continuous flow to minimize heating artefacts and an expected accuracy of 0.2 °C.

**Table 2a.** Table of instruments related to aerosol–cloud interactions that were deployed on the RV *Tangaroa* and are described in this study.

| Instrument | Type | Location on the ship | Parameter | Section | Research aim |
|---|---|---|---|---|---|
| AWS | Automatic weather station | Monkey island | pressure, temperature, RH, wind | 2.2 | 1 |
| Eppley Precision Spectral Pyranometer | Shortwave radiometer | Monkey island | short wave radiation (0.285 to 2.8$\mu$ m) | 2.2 | 1 |
| Eppley Lab Precision Infrared radiometer | downwelling infrared radiometer | Monkey island | long wave radiation (4 to 50$\mu$ m) | 2.2 | 1 |
| InterMet iMet-1-ABxn | Radiosonde | Fantail | pressure, temperature, RH, wind, GPS | 3.1.1 | 1 & 5 |
| Windsond | Radiosonde (ascent and descent) | Fantail | pressure, temperature, RH, wind, GPS | 3.1.1 | 1 & 5 |
| Lufft CHM 15k | Ceilometer | Gilson gantry | attenuated backscatter, CBH | 3.2 | 1 |
| Sigma Space MiniMPL | Mini–Micropulse lidar | Monkey island | attenuated backscatter, CBH | 3.3 | 1 |
| Metek Micro Rain Radar | Rain radar | Port side of gallery | precipitation | 3.4 | 1 |
| Brinno BCC200 | Sky camera | Monkey island | Sky images | 3.5 | 1 |
| Allskypi | Sky camera | Monkey island | Sky images | 3.5 | 1 |
| Microtops-2 | Sun Photometer | Manual on deck | AOD, water vapour, fine and coarse mode AOD at 500 nm | 3.6 | 1, 2, & 3 |
| Picarro G2301 | Cavity Ring–Down Spectrometer | Middle laboratory | Atmospheric $CO_2$ & $CH_4$ | 3.7 | N/A[#] |
| SwellPro Splash Drone 3 | UAV | Foredeck | 0.38 to 17 $\mu$m, temperature, RH | 3.1.2 | 2 & 5 |
| Helikite | Tethered balloon-kite | Fantail | 0.38 to 17 $\mu$m, temperature, RH | 3.1.3 | 2 & 5 |

*RH - relative humidity; CBH - cloud base height; AOD - aerosol optical depth; $CH_4$ - methane; DMS - dimethyl sulfide; OCS - carbonyl sulfide; CO - carbon monoxide

# $CO_2$ measurements were mainly used for contamination detection.

**Table 2b.** Table 2 continued.

| Instrument | Type | Location on the ship | Parameter | Section | Research aim |
|---|---|---|---|---|---|
| Filter sampler | Filter | Bridge Mezzanine Deck (front) | Ice nucleating particles | 3.9 | 2 & 6 |
| PCASP-100X | Optical Particle Counter | Container laboratory | $0.1\,\mu m < D_p < 3.0\,\mu m$ | 3.8.1 | 2, 3, & 6 |
| CCN-100 | Cloud Condensation Nuclei Counter | Container laboratory | $0.2\,\% < s < 1.0\,\%$ | 3.8.2 | 2, 3, & 6 |
| CPC3010 | Condensation Particle Counter | Container laboratory | $0.01\,\mu m < D_p < 3\,\mu m$ | 3.8.3 | 2 & 6 |
| SMPS3936 | Scanning Mobility Particle Size Spectrometer | Container laboratory | $0.02\,\mu m < D_p < 0.5\,\mu m$ | 3.8.4 | 2 & 6 |
| NAIS | Neutral cluster and air ion spectrometer | Bottom of main mast | $2\,nm < D_p < 42\,nm$ | 3.8.5 | 2 & 6 |
| GC-SCD | Gas Chromatograph - Sulfur chemiluminescent detector | Container laboratory | Dissolved DMS | 3.10.1 | 4 & 6 |
| MICA | Mid-Infrared CAvity enhanced spectrometer | Middle laboratory | Atmospheric and dissolved OCS, $CO_2$, and CO | 3.10.2 | 4 & 6 |

*RH - relative humidity; CBH - cloud base height; AOD - aerosol optical depth; $CH_4$ - methane; DMS - dimethyl sulfide; OCS - carbonyl sulfide; CO - carbon monoxide

The meteorological measurements together with dissolved DMS measurements (Sect. 3.10.1) were used as inputs to the National Oceanic and Atmospheric Administration (NOAA) Coupled Ocean–Atmosphere Response Experiment (COARE) meteorological and gas exchange algorithm (Fairall et al., 2003, 2011; Blomquist et al., 2006) to derive meteorological values for standard reference heights (e.g. $u_{10}$), energy and gas exchange coefficients and sea-air fluxes of DMS (Sect. 4.3.1).

    AWS measurements were complemented by human weather observations, all-sky cameras, ceilometer, Mini–Micropulse

lidar, and micro rain radar measurements, which provided important information about visibility, sky conditions, clouds, cloud type, and the amount of precipitation or fog events. In addition, up to three daily regular radiosondes of type InterMet iMet-1-ABxn were launched throughout the voyage, as well as smaller balloons carrying Windsond radiosondes that were launched in synoptically interesting conditions, e.g. within low pressure systems (Sect. 3.1.1). An overview of all radiosonde releases during the voyage is provided in Table B1 and B2.

All meteorological data are available in netCDF format at UTC times and are provided with the data set accompanying this study. Section 4.1 below provides some detail about the meteorological conditions encountered during the voyage.

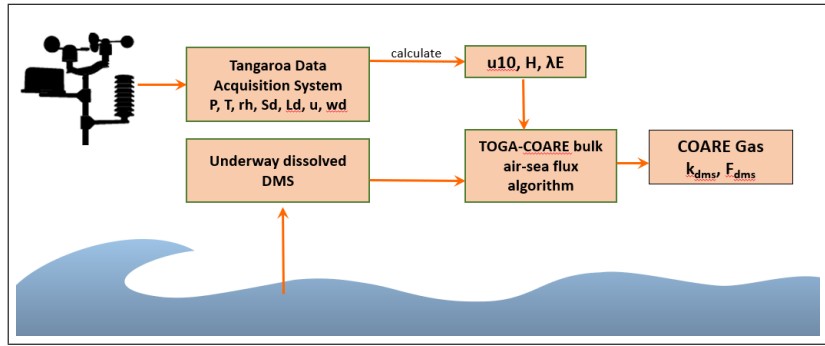

**Figure 2.** Summary of the processing scheme of the meteorological measurements (pressure (P), temperature (T), relative himidity (rh), wind (u), wind direction (wd), short- and longwave downwelling radiation (Sd, Ld)) from the AWS and radiometers measurements that were stored in the Tangaroa Data Acquisition System. Wind corrected to $10\,\text{m}$ ($u_{10}$), heat flux (H) and latent heat ($\lambda$E) were derived from these measurements and then, together with measured dissolved DMS concentrations, used to derived the sea-air fluxes of DMS.

## 3 Instrument descriptions

In addition to the instrumentation mentioned above, atmospheric measurements were conducted using a range of instruments, including a cavity ring-down spectrometer, cloud condensation nuclei counter, condensation particle counter, mobility particle size spectrometer, optical particle counter, neutral cluster and air ion spectrometer, a filter sampler, tethered balloon, and an unmanned aerial vehicle (UAV). During rare clear sky conditions, aerosol optical depth (AOD) measurements were made using a hand-held sun photometer. The instrumentation and measurement techniques of each instrument are described below. Furthermore, all data sets described here include some means of quality control and calibration procedures, which are also described below in the respective sections.

### 3.1 In situ measurements and remote sensing observations

#### 3.1.1 Radiosondes

Radiosondes are balloon-borne instruments that measure the vertical profile of temperature, relative humidity and pressure. Altitude, wind direction and wind speed are calculated from the GPS location of the sonde. A total of 58 radiosondes of type InterMet iMet-1-ABxn (hereafter referred to as iMet) and 12 of type Windsond were released on a weather balloon during the voyage (see Table B1 and B2). The iMet radiosondes were attached to $100\,\text{g}$ Kaymont weather balloons and released two to three times per day at about 07:30, 00:00 and 19:30 UTC. The typical height reached by the balloons was between 10 and $20\,\text{km}$ ASL. Of the total iMet radiosondes released, one failed right after launch, and one failed at $216\,\text{m}$ ASL. In addition, two iMet radiosondes had faulty or intermittent relative humidity readings. No iMet radiosondes were released north of $58°$ S or in unsuitable weather conditions, e.g. when wind speed was exceeding $35\,\text{kn}$ or in high swell. In addition to the iMet radiosondes, S1H3 Windsond radiosondes were launched sporadically throughout the voyage. The typical altitude reached by the Windsond

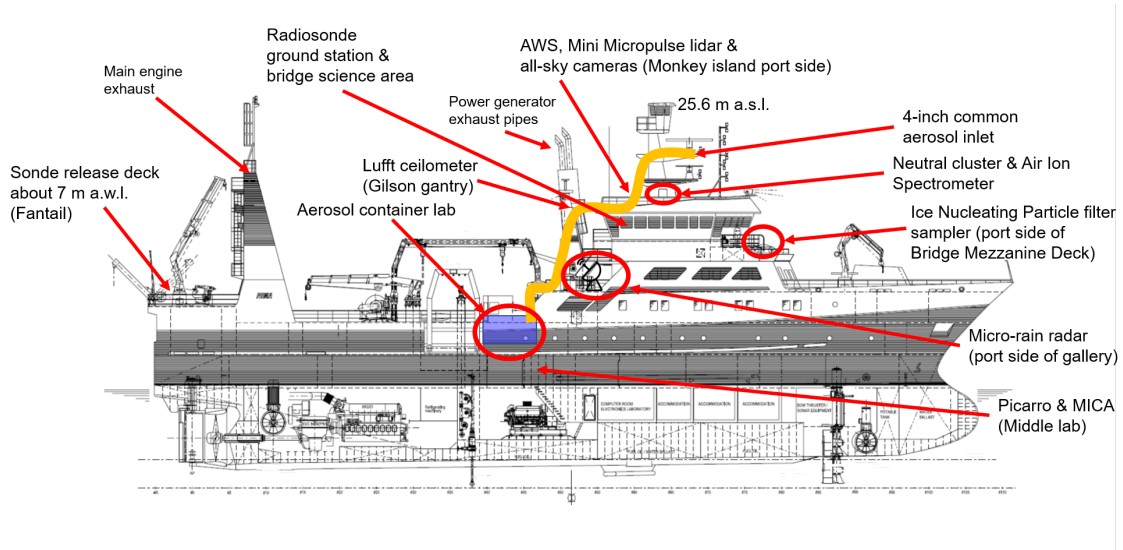

**Figure 3.** A diagram showing the locations of the atmospheric measurement equipment on-board RV *Tangaroa*. A line pump for the common aerosol inlet was used to pump sample air from above the bridge to the aerosol container lab.

radiosondes was 6 km. Five of the Windsonds were equipped with a second balloon to perform measurements during the descent, but only two descending profiles were successfully measured.

The iMet radiosondes communicated with the base station by radio at 403 MHz and included a pressure sensor with an accuracy of 0.5 hPa and a resolution of < 0.01 hPa. As described by the manufacturer, a thermistor was used to measure the temperature with an accuracy of 0.2 °C and a resolution of ±0.01 °C and a capacitive polymer sensor measuring relative humidity with an accuracy of ±5 % and a resolution of <0.1 %. The temporal resolution of the iMet sonde measurements is about 30 s, with a vertical resolution of about 60 m, except during periods of poor signal reception, which can decrease the temporal and vertical resolution.

The lightweight (about 12 g), low operating cost Windsond radiosonde provides real-time wind, temperature, and relative humidity profiles in the lower part of the troposphere with an operational ceiling of 9 km ASL. The system has an operational radio frequency configurable in the range of 400 to 480 MHz. The Windsond use a band-gap temperature sensor with a measurement range between -40° and +80 °C, an accuracy of 0.2 °C and a resolution of ±0.01°C. Relative humidity was measured using a film capacitor sensor with high accuracy (±1.8 %) and a resolution of 0.05 %. Pressure was measured directly using a microelectrome chanical piezoresistor pressure sensor with an accuracy of 1 hPa and a resolution of 0.02 hPa. The Windsond GPS ground station was not equipped with a GPS receiver; therefore, latitude and longitude was determined using an on-board GPS receiver pseudorange without differential correction. Wind speed and direction is determined independently from latitude and longitude using the GPS signal; wind speed accuracy is about 5 %. The accuracy of the wind direction depends on the GPS conditions and is therefore determined by the accuracy of the GPS sensor.

### 3.1.2 Unmanned aerial vehicle – UAV

During the voyage, two UAV flights were performed when the observed wind speed was below $5\,\mathrm{m\,s^{-1}}$. For the first flight, which took place on 4 March 2018, the UAV was equipped with an optical particle counter (OPC) of type Alphasense OPC-N2, a GoPro Hero4 camera and a customised radiosonde. The radiosonde was equipped with a SHT75 temperature and relative humidity sensor. Temperature can be measured between -40° and +40 °C with an accuracy of 0.3 °C and a resolution of $\pm 0.01°\mathrm{C}$ and relative humidity can be measured with an accuracy of 1.8 % and a resolution of 0.05 %. A customised radiosonde was required to be deployed on the UAV (rather than using a standard sonde) as it needed to interface with the OPC-N2 sensor and data had to be transferred over radio to the ground station. The Alphasense OPC-N2 is an OPC designed to count ambient particulate and drizzle sized cloud droplets between 0.38–17 µm in size. Ambient air is drawn into the sensor by a small rotary micro-fan at a flow rate of about $1.2\,\mathrm{L\,min^{-1}}$. The air enters the front of the device through a 6 mm orifice into an open optical cavity, where red laser light (around 650 nm) is incident on the incoming aerosol. Scattered light from the aerosol is collected via an elliptical mirror and a dual-element photodetector. These measurements are used to determine the particle size and particle number concentration.

While the expected battery lifetime of the UAV was 15 minutes, this was reduced to 6 minutes due to the low atmospheric temperature, resulting in a lower than expected altitude reached and unplanned landing on the ocean surface. After the battery regained enough power to take off again, the UAV was recovered. Measurements of aerosol concentration, temperature, pressure and humidity were recorded up to an altitude of about 70 m. No measurements were retrieved from the second UAV flight due to a faulty assembly of the propellers, which resulted in the loss of the aircraft.

While the operator had about seven flight hours of experience with the UAV, which is sufficient to obtain a UAV pilot certification in New Zealand, the conditions were challenging so that more experience, e.g. flight hours and practice in operating the UAV safely around obstacles, would have helped to mitigate some of the risks.

### 3.1.3 Helikite

Similar to the UAV flights, two helikite flights were conducted in suitable weather conditions subject to wind speed of below $5\,\mathrm{m\,s^{-1}}$. For the first flight, the helikite was equipped with an iMet radiosonde and an OPC-N2, providing profiles of aerosol concentration, as well as temperature, pressure and humidity profiles. The second helikite flight had to be terminated shortly after launch as the weather conditions changed rapidly, resulting in no measurements.

The helikite comprised a large $6\,\mathrm{m^3}$ balloon with a sturdy kite base. Lift can be achieved by inflating the balloon with helium and is aided by the additional lift of the kite. As a result of the large volume of the balloon, the total payload can be around 2 to 3 kg. The helikite was flown off the fantail and was anchored to an electric winch fitted with >1 km of high tensile strength Dyneema line. This system itself offers the opportunity to fly more expensive sampling equipment than typically deployed during a radiosonde flight where equipment is usually not recovered.

The first flight of the helikite occurred midway through the voyage on 26 February 2018. Conditions were good with wind speeds less than $5\,\mathrm{m\,s^{-1}}$. Due to the inexperience of the helikite operator, the helikite was flown in near neutral buoyancy,

i.e. the lift provided by the balloon was near or equal to the weight of the payload. As a result, the only lift received during the flight was from the kite alone. Once the helikite left the slipstream of the *Tangaroa* it rose slowly to an altitude of 260 m. At this stage, the additional weight of the tethered string counter-balanced all lift. After sampling for around 45 minutes, the system was reeled back in.

## 3.2 Ceilometer

During the voyage a ceilometer, which is a low-power lidar, made continuous measurements of the overlying atmospheric state. The ceilometer deployed on the voyage was a Lufft CHM 15k, which operated at an infrared wavelength of 1064 nm, with a maximum range of 15 km. The ceilometer was installed on the Gilson gantry behind the monkey island (Fig. 3), located approximately 16 m ASL. The ceilometer continually emits short light pulses vertically into the atmosphere, where light is scattered back by clouds, aerosol and air molecules. By detecting the run-time of the return signal, the ceilometer identifies the lowest altitude of a cloud as the layer with higher particle backscatter characteristics. The backscatter is calculated at 1024 vertical levels in the atmosphere (about 15 m vertical resolution). By applying detection algorithms using the operational software to the backscatter measurements, quantitative information on cloud base height (CBH), cloud fraction (CF), cloud layers, and boundary layer height can be determined. As the emitted signal is strongly attenuated by thick clouds, it is often not possible to observe the middle or tops of clouds. On some occasions, the movements of the ship (pitch and roll) affected the ceilometer measurements when there were horizontally inhomogeneous clouds, producing a vertical filament structure in the backscatter.

## 3.3 Sigma Space Mini–Micropulse Lidar

The Mini–Micropulse Lidar (MiniMPL) is a sophisticated laser remote sensing system that provides continuous, unattended monitoring of the profiles and optical properties of clouds and aerosols in the atmosphere. A micropulse lidar (MPL) transmits laser pulses that scatter (reflect) off particles in the atmosphere. The MPL then measures the intensity of backscattered light using photon-counting detectors and transforms the signal into atmospheric information in real time. During the campaign, aerosol backscatter data were collected using the Sigma Space MiniMPL, which is a compact version of the standard MPL described by Ware et al. (2016). The manufacturer specifications for the MiniMPL's maximum range is 30 km. However, accurate MPL measurements can rarely be obtained up to this height. This is because the retrievals are strongly impacted by absorption and scattering along the beam path, with the signal-to-noise ratio decreasing with height, resulting in a lower effective range. During the campaign this range was mostly limited to the first few kilometers due to dense low level clouds that saturated the return signal. Other periods of clearer skies had distinct cloud features at up to 8-9 km before fading into background noise above these features. Our data processing was limited to 10 km, as the voyage focused on marine boundary layer clouds as well as low and middle level clouds as defined in research aim 1.

The MiniMPL is a dual-polarisation micropulse lidar operating at a wavelength of 532 nm at 2.5 kHz (pulse energy is 3–4 µJ). Laser light that is scattered back towards the instrument is collected by an 80 mm diameter receiver (Spinhirne et al., 1995; Campbell et al., 2002; Flynn et al., 2007). The vertical range resolution was set at 15 m during the ship campaign. A

two-axis scanning head was mounted on top of the environmental enclosure containing the lidar, to provide variable-angle scanning throughout the voyage. Azimuth was fixed for observations (pointing outward from the side of the ship) and the scanning head was programmed with an elevation-only scanning routine that included the following angles: 0, 5, 10, 15, 20, 30, 40, 45, 50, 60, 70, 80, 90° elevation. The finer, 5° elevation step was used near the horizon, and then 10° steps from 20° to 90° (zenith). An observation was also made at 45° because it is convenient geometrically. At 0° and 90°, the observations were 12 minutes long, at other angles 6 minutes, resulting in the full scanning cycle taking 90 minutes. The elevation angle of each particular observation is recorded in the data file. Note that there were some instances during the campaign (overall 9 days) when a software failure caused the scanning system to not follow the programmed schedule.

The MiniMPL was not motion stabilised on the ship and so any ship movement is captured within each integration period of the measurements. While each individual laser pulse will be received near instantaneously and "freeze" the ship motion, the full number of pulses, i.e. scans, during the minute-long integration period will result in a number of profiles over a range of pointing angles due to ship motion, which will be all averaged together for that minute. This applies for the vertical pointing scans and the scans done at the distinct elevation angles.

The instrument produces native binary files ("mpl") with backscatter and housekeeping metadata, which can be converted to netCDF files using manufacturer supplied software (SigmaMPL) or third-party software (*mpl2nc*). The primary output quantity is the normalised relative backscatter (NRB) profile, representing the backscattering of light (in photon counts $\mathrm{km^2\mu s^{-1}\mu J}$), after correcting and normalising the measurements. An auxiliary GPS unit was connected to the lidar, whose output was recorded in the product files. The instrument was installed on the monkey island (Fig. 3).

The instrument ordinarily requires range-dependent calibration of backscatter in the form of dead time, overlap and after-pulse corrections, which account for the saturation of the photon counter, incomplete overlap of the outbound and inbound beams, and post-pulse reflections from the internal parts of the instrument, respectively. These were supplied by the manufacturer. An improved calibration was produced post-voyage, which addresses a technical issue with the manufacturer calibration (bit truncation of dead time polynomial coefficients) and a change in overlap which might have happened during transport and deployment of the instrument. The data product produced with the third-party *mpl2nc* software was calibrated with the improved calibration and is supplied with the data set.

The CHM 15k ceilometer and Sigma Space MiniMPL measurements were both processed using the Automatic Lidar and Ceilometer Framework (ALCF, Kuma et al., 2020b). While ALCF was developed to provide a tool to evaluate clouds simulated by climate models or reanalysis data using ceilometer or MiniMPL observations, ALCF can be run independently of any model input to process ceilometer or MiniMPL observations. ALCF can ingest the raw measurements, transform backscatter profiles to profiles comparable with different instruments, and output the results in netCDF format. ALCF is described in detail in Kuma et al. (2020b).

Two different data products are provided for both the ceilometer and MiniMPL data, level 0 and level 1:

– Ceilometer level 0: contains one file per 5 minutes of observations in the native netCDF format (.nc files). The 5 minute files provide one profile every 2 s, with a temporal resolution of 15 m.

- MiniMPL level 0: contains one file per hour of observations in the native binary (.mpl) format which can be processed using the proprietary SigmaMPL software or converted to netCDF format using a python tool mpl2nc. The hourly files provide one profile every 6 s with a vertical resolution of 15 m.

- MiniMPL (minimpl_mpl2nc): contains MiniMPL data that were processed using the mpl2nc source code to convert raw MiniMPL data files to netCDF files. The hourly files provide one profile every 6 s with a vertical resolution of 15 m.

- Level 1: contains ALCF processed raw ceilometer and MiniMPL data sets (one file per day) in netCDF format. The dataproducts included are time series of vertical backscatter profiles, backscatter standard deviation, cloud base height, cloudmask, and lidar ratio. The data were sub-sampled to 5 min intervals with a vertical resolution of 50 m."

## 3.4 Micro Rain Radar

During the voyage a Metek Micro Rain Radar 2 (MRR-2) made continuous measurements of the overlying atmospheric state between 7 and 27 February 2018. The MRR-2 is a vertical pointing FM-CW (Frequency-modulated continuous-wave) radar with a centre frequency of 24.23 GHz and a frequency modulation between 0.5–15 MHz. The scatter return signal can be processed to derive Doppler spectra at a number of predefined vertical ranges, from the ground to several hundred meters. For rain droplets, the relationship between terminal fall velocity and drop diameter is used to derive vertical profiles of the rain drop size distribution from the Doppler spectra. These drop size distributions can be integrated to derive rain rates even for very small amounts of precipitation, below the thresholds detectable by conventional rain gauges. The software supplied by the manufacturer completes all this processing and also makes estimates of other parameters, such as liquid water content. The temporal resolution of the measurements is 10 s. Measurements of snowfall using this instrument are more challenging because the particle backscattering cross sections depend on both their mass and shape, while terminal velocities relationship to particle size depends on their projected area. In the case of snowfall, we use the method of Maahn and Kollias (2012) to process the raw data to derive radar reflectivity, velocity, spectral width and snowfall rate estimates. The radar was installed on the port side of the gallery beneath the bridge (Fig. 3).

It should be noted that the Doppler velocity information is integrated over a period, meaning that variations in the ship motion will impact the spectral width of the signal, adding additional uncertainty to the derived MRR precipitation data. There are also signs of the Doppler velocity being degraded by ship motion around 22:00 UTC. Unfortunately, disdrometer measurements were not available on this voyage and therefore, the MRR was not calibrated. This also potentially adds uncertainty to the derived precipitation values when using these data quantitatively. However, the data are still very valuable for masking periods of precipitation as used and described in Hartery et al. (2020a).

## 3.5 Sky cameras

A pair of Brinno BCC200 cameras were installed on the starboard and port side of the monkey island (Fig. 3). The cameras were configured to capture an image of the sky every 5 minutes. The resolution of the images is $1280 \times 720$ pixels, and they are recorded in a video file (Motion JPEG). These images are complementary to the human weather observations, ceilometer,

and lidar data to evaluate cloud cover, cloud types and cloud base height during the voyage. An additional camera system, named allskypi, was also installed on the monkey island, adjacent to the MiniMPL. The allskypi system contained a ZWO ASI178MC (3096 × 2080 pixels) camera with a fisheye lens connected to a Raspberry Pi single–board computer. Allskypi acquired images at 7 exposure levels every 5 minutes, with in post voyage processing combining these exposure stacks into a single image by exposure fusion as described in Mertens et al. (2009). Over the course of the voyage over 60,000 images were taken, resulting in nearly 9,000 HDR images. When combined with ship positioning data, including roll and pitch, cloud fraction can be determined by simple thresholding techniques such as the ELIFAN algorithm presented in Lothon et al. (2019). This algorithm was adapted to the allskypi system to obtain cloud fractions by masking out pixels below an elevation of 20° to exclude ship structure and to avoid low elevation angles that thresholding techniques struggle to accurately resolve. Additional masks were also applied for the remaining ship structure and for the solar disk. Furthermore a record of whether or not the sun was obscured by clouds was produced by monitoring the image saturation over the solar disk. All-sky imagery, along with estimates of cloud fraction and sun obscuration obtained during this voyage were primarily used for quality assurance and quality control (QA/QC) of other sky viewing observations such as the ceilometer and MiniMPL measurements (as described in, e.g. Wagner and Kleiss, 2016). Cloud fraction derived from the sky camera product is also useful for model evaluation and when combined with the raw imagery and ceilometer data it could potentially be used to classify cloud types as described in Huertas-Tato et al. (2017).

### 3.6 AERONET Maritime Aerosol Network (MAN) hand-held sun photometer

When clear-sky conditions were present, column aerosol measurements were made using a portable sun-pointing Microtops-2 sun photometer, operating at five wavelengths. The instrument was calibrated prior to the voyage by NASA and operated according to the Aeronet Maritime Aerosol Network protocols with an attached GPS receiver to log positional information. Scans were usually taken in groups of five measurements and only made under clear-sky conditions with no clouds present near or around the sun taking care to avoid measurements through cirrus clouds. Clear sky conditions were rare and only observed for less than 2% of the time. Due to the otherwise high cloud cover occurrence during the voyage (see Figure 11 below), these measurements were performed only four times on three distinct days. Processed products include AOD at five wavelengths, water vapour content, the angstrom parameter and aerosol optical depth for the fine (sub-micron) and coarse (super-micron) modes calculated according to the spectral deconvolution algorithm of O'Neill et al. (2003). The data are available via the *MAN website for the TAN1802 voyage*. To date over 600 voyages, including the TAN1802 voyage described here, have contributed to the MAN database providing a valuable global resource for analyses (e.g. Smirnov et al., 2009, 2011) and use in validation and model development of important aerosol components such as oceanic sea-salt (Bian et al., 2019).

### 3.7 Cavity ring-down spectrometer – Picarro

By the voyages nature, the ship did not always head into the wind. As a result, there were distinct times throughout the voyage when winds from the stern outpaced the motion of the ship and therefore the sampling line of air sampling instruments was often exposed to exhaust from the ship. This problem was largely unavoidable, but the ship's measurements of wind-speed and

heading combined with high precision measurements of carbon dioxide ($CO_2$) were used to identify contamination episodes.
Experience from previous voyages (e.g. Law et al., 2017) has shown that the Cavity Ring–Down Spectrometer (CRDS) is ideally suited to detect ship exhaust contamination. For this and other reasons beyond the scope of this paper, a CRDS (G2301, Picarro) was installed on the ship and operated continuously throughout the voyage. The CRDS was installed in an equipment room off the middle lab (Fig. 3) measuring atmospheric mixing ratios of $CO_2$ and methane ($CH_4$) continuously at 1 Hz. Air for analysis by the CRDS was obtained from an inlet on the Forward Light Tower above the Bridge ($\sim$20 m ASL) via an airline to the middle lab. Air was pumped down from the airline at about 2 L $min^{-1}$, of which 150 mL $min^{-1}$ (determined by a mass flow controller) is used for analysis. Before the air from the airline was sampled by the analyser, it was dried to a dew point of 2–4 °C using a thermoelectric cooler and then dried further to a dew point between -30 and -40 °C using a back-flushed Nafion dryer in which remaining water vapour in the air is transferred to the CRDS exhaust air that had been dried by passing it through a molecular sieve trap. While the Picarro instrument does measure the concentration of water vapour in the air, in this system the water vapour measurement was only used as a diagnostic indicator of system performance. Solenoid valves controlled by the Picarro were used to select either pre-dried air for analysis, or air from one of three reference tanks, plus a target tank, for system calibration. A calibration sequence was automatically run twice per day.

The analyser has a built-in Windows 7 PC for data acquisition and control of the CRDS system. Measurements were stored in the form of hourly text files on the Picarro PC's solid state drive. File times are UTC, whereas the Picarro's internal PC was set to NZST (UTC + 12 h) and was synchronised to *Tangaroa's* time server. Picarro's sample time is around 1 s (there is some variability around this value), but this is shared among the three compounds measured ($CO_2$, $CH_4$, and $H_2O$), so the individual compound sample time is around 3 s. Once per day the data files were backed-up to the network drives of the ship and processed to produce diagnostic plots to check system operation and performance.

One netCDF file containing the level 1 data product of the Picarro measurements is provided, containing 5-min average of the $CO_2$ and $CH_4$ concentrations measured during the voyage. Data quality flags are provided for every substance, including flags to mark data subject to exhaust contamination.

## 3.8 Common Aerosol Sampling Conduit

Throughout the voyage, the container laboratory, which housed the majority of the underway aerosol sampling instrumentation, was positioned behind the mid-ships exhaust (2 m ASL). To prevent exhaust air from contaminating the in situ measurements of ambient marine aerosol, ambient air was drawn from the mast of the RV *Tangaroa*, through the conduit (Fig. 3) to the container laboratory, at a rate of $4.1 \times 10^{-2}$ $m^3$ $s^{-1}$. Size-dependent losses of particulate to conduit walls from an-isokinetic sampling, gravity, turbulence, and diffusion are described in detail in Hartery et al. (2020b). The average transit time for particulates through the 40 metre long common aerosol sampling conduit was <8 s. The inlet of the conduit was angled downwards to prevent the accumulation of precipitation within the inlet region.

The aerosol container laboratory (Fig. 4) was equipped with the following instruments: an Optical Particle Counter of type PCASP-100X, a Cloud Condensation Nuclei Counter of type CCN-100, a Condensation Particle Counter of type CPC-3010 and a Scanning Mobility Particle Size Spectrometer (SMPS). Within the aerosol laboratory, the main sampling conduit connected

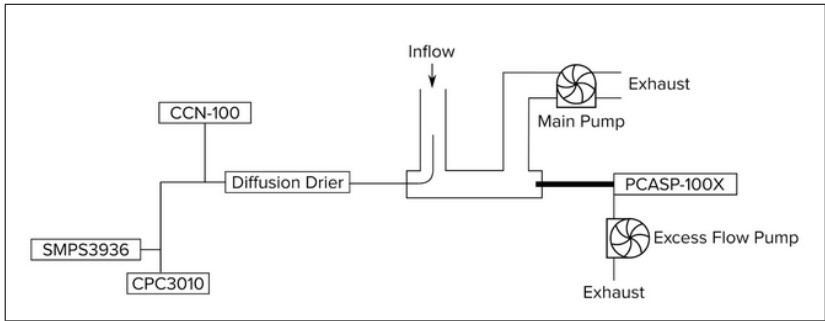

**Figure 4.** A schematic layout of the particle counting instruments in the aerosol container laboratory (not to scale).

to a plumbing manifold with three outflows: (1) a sample flow for the CCN-100, SMPS3936, and CPC3010; (2) a sample flow for the PCASP-100X; and, (3) a primary exhaust flow (Figure 4). The inlet of the sampling line for the CCN, SMPS, and CPC
was positioned within the stream of the main sampling conduit to improve sampling efficiency of particulate. Air sampled from the sampling line connecting the CCN, SMPS and CPC was dried with a custom-built diffusion drier prior to being sampled by each instrument. The inlet of the sampling line for the PCASP-100X was also positioned in the stream of the sampling conduit to improve sampling efficiency of particulate. The PCASP-100X used on the ship was an airborne version designed to be isokinetic for an instrument inlet speed of about $100\,\mathrm{ms^{-1}}$ with the instrument mounted external to the aircraft on a pylon.
For operating the instrument in the laboratory, airflow was drawn through the instrument inlet with an external ring compressor purge pump, which improved response time and isokinetic sampling efficiency of the PCASP by increasing the airflow in the region of the internal cavity hypodermic inlet. However, the air sampled by the PCASP was not dried. The temperature within the aerosol container laboratory was typically about $20\,°\mathrm{C}$, while the ambient temperature was about $0\,°\mathrm{C}$ between 15 February and 15 March. As such, the relative humidity of the air sampled by the PCASP was likely substantially lower than the ambient
relative humidity. The difference between the laboratory and ambient relative humidity would have partially dried the particles; though, this has not been explicitly quantified or accounted for in the data as it would require a priori knowledge of particle composition and hygroscopicity. As a result, there are potentially biases between the size of particles detected by the PCASP and particles detected by the SMPS. These biases are likely greater outside the period of 15 February to 15 March as a result of there being a smaller difference in temperature between the laboratory and ambient conditions. Finally, the remainder of the
air passing through the main sampling conduit was directed towards the exhaust via the main pump. A schematic layout of the plumbing is represented in Fig. 4.

In addition, a Neutral cluster and Air Ion Spectrometer (NAIS) was installed on at the bottom of the top platform of the ship. All of these instruments will be described in further detail below. Operation of different types of instruments covering overlapping, or often the same, particle size ranges offers a measure of mutual quality control on the measurements.

### 3.8.1 Optical Particle Counter

The abundance of particles in the size range 0.1–3.0 µm was measured with a Passive Cavity Aerosol Spectrometer Probe (PCASP-100X; Droplet Measurement Technologies). The PCASP and Scanning Mobility Particle Size Spectrometer (SMPS, see Sect. 3.8.4) are both spectral particle counters which provide the partial number concentration at given sizes, i.e. the number of particles observed within various sub-ranges over the total range of observable sizes. PCASP measures size according to the optical diameter (i.e. how it refracts and scatters light). The advantage of the PCASP is that it can record data quickly (1 Hz), while the SMPS instrument is slow. However, the disadvantage of the PCASP is that it can only measure particles larger than 100 nm.

The PCASP instrument recorded the number of observed particles in 30 particle size bins at a frequency of 1 Hz. While the PCASP measurement frequency is high, it is generally beneficial to integrate the PCASP over a period as long as 5 minutes to get better counting statistics and decrease the relative measurement uncertainty. As a result, the measurements in each size bin were block-averaged into 5 minute intervals in a post-processing stage. Between 9 February and 21 March 2018, there were a total of 12,000 5 minute intervals, throughout which the instrument recorded for a total of 11,400 intervals. Four additional measures of quality control were implemented in the data post-processing chain, viz:

1. Using the mole fraction of $CO_2$ in a coincident sampling line, measured by the Picarro instrument, to screen the 1 Hz sub-samples for contamination by ship exhaust (Hartery et al., 2020b). For 11,118 of the 5 minute intervals with data, the mole fraction of $CO_2$ was less than 405 ppm and the sample was flagged as "clean air".

2. Using the relative wind direction measured by the sonic anemometers in a simple wind sector analysis. Measurements that were taken when the relative wind direction from both the port and starboard anemometers were within 60° of aftward were removed. All other samples were flagged as having come from a "clean sector". Out of the 11,118 clean air samples (i.e. not contaminated by ship exhaust) 9,986 were from clean sectors.

3. Calculating the standard deviation of the 1 Hz sub-samples within each of the 5 minute intervals (Hartery et al., 2020b). Even for a steady concentration of particles, the number of particles counted within a given interval will vary according to Poisson counting statistics; thus, the standard deviation of the 1 Hz samples within the 5 minute interval should be approximately equal to the square root of the measured concentration. However, if the standard deviation of the 1 Hz sub-samples was more than three times greater than the square root of the concentration, then the 5 minute sample was discarded. This additional measure removed 184 samples.

4. Removing observations in the first size bin, as the lower threshold of particle detection in this bin is not well defined due to potential variations in the refractive index of the measured particle(s). Additionally, the 4th and 5th size bins were added together and redefined as a single bin, as the 5th size bin was in between linear gain stages of the particle counter, which led to spuriously low counts.

Overall, 81.7 % of the measurements taken remained after the post-processing described above. This is a reasonable data retention rate, considering the challenges of sampling just ∼10 m ahead of the mid-ships exhaust. After post-processing of the

measurements, the processed particle size spectra were corrected to standard temperature and pressure. In addition, the particle size spectra were corrected according to parameterisations of the sampling and transport efficiency of aerosol particles detailed in Brockman (2001). These calculations accounted for anisokinetic sampling conditions, diffusion, gravitational settling, and turbulence. Finally, the total particle concentrations in each size bin were normalized by the logarithm of the bin's width.

### 3.8.2 Cloud Condensation Nuclei Counter

The concentration of individual particles that can form into cloud droplets, i.e. cloud condensation nuclei (CCN), was measured at varying water vapour supersaturations with a CCN counter (CCN-100; Droplet Measurement Technologies). The CCN and Condensation Particle Counter (see Sect. 3.8.3) instruments are integrating particle counters, which provide the total concentration over a given size range. For a CCN-100 counter, the lower size threshold of observable particles is dependent on the chamber supersaturation and the hygroscopicity of aerosol. The benefit of the CCN-100 is that it provides a measure of the number of "cloud-relevant" particles.

Prior to being sampled by the CCN-100, particles were dried with a diffusion drier. The raw CCN-100 observations were recorded at 1 Hz. The instrument observed the total abundance of activated particles for water vapour supersaturations between 0.2–1.0 % at intervals of 0.1 %. Each interval was observed for three minutes, resulting in 1 scan every 30 minutes. Measurements within each supersaturation interval were only retained if the instrument was in thermal equilibrium. The raw, 1 Hz data were averaged into the 3 minute intervals for which supersaturation was constant, screened for contamination by ship exhaust according to the coincident $CO_2$ mole fraction, the relative wind direction, and the standard deviation of the 1 Hz subsamples. Finally, all of the screened observations at thermal equilibrium were merged to a common hourly date coordinate. As with the PCASP-100X, measurement uncertainties are proportional to the square root of the observed concentration.

The CCN-100 was calibrated by the manufacturer prior to the voyage and calibrated by the operator after the voyage. The calibration procedure followed the methodology described in Rose et al. (2008). Overall, the supersaturation of each stage was accurate to within 20 % of the set-value, e.g. the stated supersaturation of 0.3 % was accurate to within $\pm 0.06$ %.

### 3.8.3 Condensation Particle Counter

The total abundance of particles in the size range 0.01–3.0 μm was measured with a Condensation Particle Counter (CPC3010; TSI) at a frequency of 1 Hz. Similar to the data processing procedure for the PCASP-100X, the raw data were screened for contamination by ship exhaust according to the coincident $CO_2$ mole fraction, the relative wind direction, and the standard deviation of the 1 Hz subsamples. The screened data were then averaged over 5 minute intervals and merged to the common date coordinate. On 1 March 2018, the laser beam dump became partially dislodged within the optical cavity of the CPC3010 and the operator was unable to resolve this issue at sea. As this led to spurious counts, the data following 1 March were excluded from the data set.

### 3.8.4 Scanning Mobility Particle Size Spectrometer

The abundance of particles in the size range 0.020–0.50 μm was measured with a Scanning Mobility Particle Size Spectrometer (SMPS3936; TSI). The SMPS instrument sizes particles according to how mobile the particle is in air. The instrument measured the total abundance of particles passing through an electrostatic classifier (EC3080L; TSI) with a condensation particle counter (CPC3772; TSI). For a specific voltage setting, only particles of a specific size and charge will pass through the EC3080L and be observed by the CPC3772. The instrument was set to observe the concentrations of particles at 32 logarithmically-spaced voltage levels. The concentration at each voltage level was observed over a period of 10 s, with an additional 2 s purge between voltages. The instrument scanned through the 32 set voltages once every 6.4 minutes. As with previous counters, the coincident $CO_2$ mole fraction time series was used to screen the raw 0.1 Hz data for contamination by ship exhaust. After screening, the concentration-voltage spectra were merged to a common 30 minute data coordinate. The inversion of the merged concentration-voltage spectra into concentration-diameter spectra was calculated in the post-processing stage, accounting for multiple-charged particles and diffusional losses to the bipolar diffusion charger within the SMPS (Stolzenburg, 1988). As with the PCASP-100X data, the processed particle size spectra were corrected to ambient conditions by applying the size-dependent loss corrections detailed in Hartery et al. (2020b).

### 3.8.5 Neutral cluster and Air Ion Spectrometer - NAIS

To detect the distribution of ions (charged particles and cluster ions) in the electric mobility range from 0.0013–3.2 $\mathrm{cm^2\ V^{-1}\ s^{-1}}$ and the distribution of aerosol particles in the size range from 2–42 nm, a Neutral cluster and Air Ion Spectrometer (NAIS, Airel Ltd., Mirme and Mirme, 2013) was deployed on the ship. The measurements are taken with a temporal resolution of 1.5 minutes. The instrument was installed at the bottom of the mast located on the *Tangaroa* monkey island (Fig. 3), with the inlet facing the port side of the ship. The NAIS has two identical cylindrical Differential Mobility Analyzers (DMA) for the simultaneous measurement of positive and negative ions. Each analyser has a sample flow rate of 30 $\mathrm{L\ min^{-1}}$ and a sheath flow rate of 60 $\mathrm{L\ min^{-1}}$. Such high flow rates are used to avoid diffusion losses and ensure significant signal to noise ratio, even when ion concentrations are low. The inner cylinder of each analyser is divided into four isolated parts, which keep a constant voltage during a measurement cycle. The outer cylinder is divided into 21 isolated rings connected to 21 electrometers. Naturally charged particles are moved by a radial electric field from the inner cylinder of the DMA to the outer cylinder. The current carried by the ions is further amplified and measured with electrometers. These data are converted first into electrical mobility and further into the size distribution of ions. For detection of neutral particles, particles are charged by ions originating from a corona discharge to an equilibrium that is used to calculate the total particle concentration in a given size range. The size of ions generated by the corona discharge is around 2 nm, masking the atmospheric signal of this size of neutral clusters. In addition to ion and particle measurements, each measurement cycle contains an offset measurement during which particles in the sample air are charged by a unipolar corona charger and electrically filtered for measuring the background level of the electrometers. The offset is used to evaluate noise levels and instrument functioning. Measures of quality control were implemented in that data post-processing chain. First, the mole fraction of $CO_2$ in a coincident sampling line, measured by

the Picarro instrument, was used to screen the NAIS data according to the suggested filtering protocol outlined in Sect. 4.2.1. Note that the filtering of the NAIS data differ from the filtering of pollution events for the PCASP and SMPS data, but the impact on the remaining measurements is negligible. Secondly, data above 15 nm were excluded from the final data set due to technical issues with one of the electrometers. Further quality control of the measurements was performed by following the data cleaning and quality check guidelines described in Manninen et al. (2016), which are mainly based on visually inspecting the measurements. Overall, 37 % of the measurements made were included in the QA/QC data set.

In this paper, we only present the particle measurements, and only the particle measurements are included in the data set (see Table A1), excluding the ions as they will be described and discussed in a different publication. The data are provided in one netCDF file that contains the particle size distribution with 29 size bins with diameters from 0.75 to 42 nm. The temporal resolution of the data stored is 1.5 min. The file also includes a data quality flag for diameters that should not be used because of the charger ions or instrument malfunction and a second flag for times of observed polluted periods.

### 3.9 Filter sampler

Ice nucleating particles were collected onto pre-cleaned, 0.2 µm pore diameter Nuclepore polycarbonate membrane filters, each overlying a 3 µm pore diameter clean support Nuclepore membrane, in open-faced filter holders (McCluskey et al., 2018). The filter sampler was placed in front of the bridge about 15 m ASL, at a position relatively clear of sea-spray generated by the ship hull and relatively free from contamination (e.g. ship exhaust), and connected via vacuum tubing to a pump inside the sub-bridge mezzanine space. Twenty one filter samples were obtained between latitudes $41°$ to $73°$ S, with sample collection periods ranging from 13.5 to 50 hours, at an average flow rate of $14 \, \text{L min}^{-1}$. The total volume sampled during each collection was recorded using a gas meter placed after the pump. Samples were stored and shipped frozen to the Colorado State University (CSU) for measurement. Three field blank filters were collected (i.e. filter units opened and closed on deck before filters were removed and stored as for sample filters) and used to adjust for background INPs.

Temperature dependant number concentrations of INPs active via immersion freezing (one spectrum per filter) were obtained with the CSU ice spectrometer (IS) (Hiranuma et al., 2015; McCluskey et al., 2018). Filters were placed into sterile 50 mL polypropylene centrifuge tubes, 5 mL of 0.1 µm filtered deionized water added and particles re-suspended by tumbling end-over-end. Next, 50 µL aliquots (typically 64) of suspensions, and 15-fold dilutions of the suspensions, were dispensed into sterile, 96-well polypropylene trays and the trays placed into the cooling blocks of the IS. Blocks were cooled at $0.33° \, \text{C min}^{-1}$ and the freezing of wells recorded with a CCD camera system. The lower limit of measurement was typically -28° C, with the upper limit defined by sampling statistics. The number of frozen wells at each temperature were converted to the number of INPs $\text{mL}^{-1}$ of suspension using Eq. 13 in Vali (1971). Then this value was corrected for background INPs using a regression of the combined results from the three field blank filters (about 1 INP/filter at -15°C, about 4 INPs/filter at -20°C, and about 55 INPs/filter at -25°C). The corrected measurement was then converted to INPs $\text{L}^{-1}$ air at ambient conditions using the total volume filtered. Ninety five percent confidence intervals were obtained by applying Eq. 2 in Agresti and Coull (1998).

Two netCDF files are provided for the filter measurements in the Zenodo archive, both containing level 1 data, i.e. data that have been quality controlled as described above. One netCDF file (IceNucleiFilterEnviron_TAN1802_2018_level_01.nc)

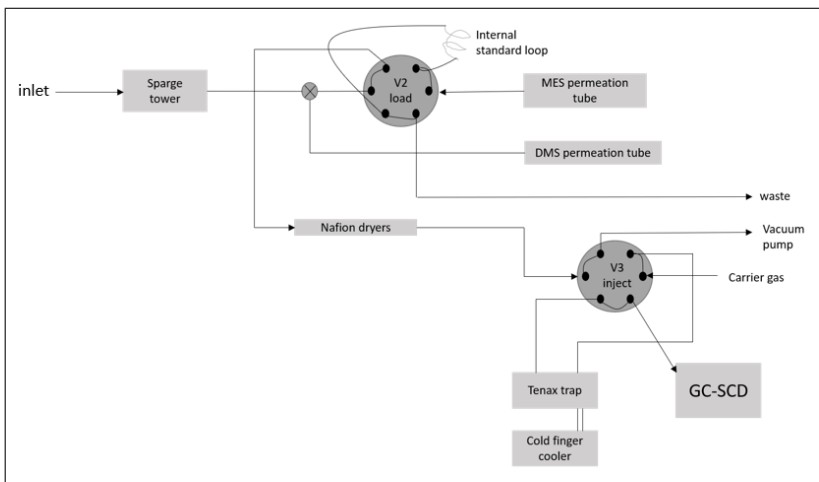

**Figure 5.** A schematic layout of the DMS analysis system for measuring dissolved DMS, including: the sparge tower, internal and external standards, Nafion dryers, Tenax trap and the GC-SCD. (not to scale).

contains details of sampling while the other (IceNucleiFilter_TAN1802_2018_level_01.nc) provides the ice nucleating particle concentrations.

### 3.10    Dimethyl sulfide (DMS) and carbonyl sulfide (OCS) measurements

#### 3.10.1    Gas chromatograph – GC–SCD

Underway continuous measurements of dissolved dimethyl sulfide (DMS) were made using a gas chromatograph (GC-SCD;
Walker et al., 2016). A schematic of the DMS analysis system is shown in Fig. 5. The instrumentation consists of a custom built automated preconcentrator (semi-automated purge and trap system), which is coupled to an Agilent Technology 6850 gas chromatograph (GC) with an Agilent 355 sulfur chemiluminescent detector (SCD). The equipment was set up on the shelter deck of the ship in the "aerosol container" which had a constant surface seawater supply from a depth of about 5 m. For the dissolved DMS measurements, surface seawater was sampled continuously each day from approximately 08:00 to 21:00, with
a 1 hour interruption when water samples taken by the CTD (conductivity, temperature and depth sensor) rosette water sampler were analysed around mid-day. During the voyage, 96 discrete samples were collected using CTD Niskin bottles, where water samples were collected from four to six discrete depths in the layer between 10 and 100 m. These discrete seawater samples provide a means to derive DMS profiles throughout the 100 m ocean surface layer.

     To avoid contamination, seawater samples were gently filtered by pumping the water, using a peristaltic pump, through a
25 mm GF/F filter. The filter was changed after every four injections. A calibrated volume of 5.84 mL of the filtered water was transferred to a silanized glass chamber (sparge tower), which was fitted with a quartz frit and purged with zero-grade nitrogen (99.9 % pure). To prevent organic matter build-up the chamber and frit were cleaned daily. The gas–phase DMS sample was then dried by passing through Nafion® dryers and trapped on a teflon-lined Tenax® stainless steel trap at -20 °C for 5 minutes

and purged at 110 °C for GC analysis (DB-megabore sulfur column, 70 m length, 0.530 megabore diameter and film thickness 4.30 μm). The detector sensitivity was checked daily using two temperature controlled VICI® permeation tubes, one filled with methylethylsulfide (MES) and the other with DMS. MES was used as an internal standard, with samples doped during analysis to allow for correction of short-term changes in detector sensitivity, while the DMS permeation tube provided the external standard (Walker et al., 2016). On average over the duration of the voyage, the detection limit was 0.079 ($\pm$ 0.016) pgS s$^{-1}$.

To establish the detector response to sulfur, a calibration curve (instrumental signal versus concentration) was generated using solutions prepared from pure hydrolysed dimethylsulfoniopropionate (DMSP). The calibration curve was used to determine the concentration in an unknown sample by comparison with a set of standard samples of known concentration. Here, DMSP was diluted to produce six different standard solutions ranging between 0.1 and 9.54 nmol L$^{-1}$ in 20 mL gas-tight glass vials. Two pellets of sodium hydroxide (NaOH) were added to each vial to hydrolyse the DMSP to DMS before sealing the vials with aluminium caps. The standard solutions were then treated the same way as the samples by injection into the stripping system. The relationship between the standard concentrations and the instrument response was then used to determine the concentration of DMS measured in the samples taken during the voyage.

Overall, the quality of the DMS measurements is very good. The data quality procedure that was implemented only removed six data points from the whole data set obtained during the voyage.

One netCDF file containing level 1 data is provided for the DMS measurements. The file contains the dissolved DMS concentrations which were quality controlled and calibrated as described above as well as the calculated DMS fluxes and its gas exchange coefficient $k$. Details on the calculation of the DMS flux and the gas exchange coefficient are given in Section 4.3.1.

### 3.10.2 Mid–Infrared CAvity enhanced spectrometer – MICA

The MICA (Mid–Infrared CAvity enhanced spectrometer, which is a prototype of a commercially available ABB Los Gatos OCS Analyzer) instrument measures carbonyl sulfide (OCS), carbon monoxide (CO) and CO$_2$ employing Off Axis Integrated Cavity Output Spectroscopy (OA-ICOS, Baer et al., 2002; O'Keefe et al., 1999; Paul et al., 2001). Air samples are internally pumped through a 305 mm long and 51 mm diameter cavity at a mass flow rate of about $6 \times 10^{-6}$ kg s$^{-1}$ with the cavity pressure regulated to 80 hPa. The beam of a quantum cascade laser (QCL) ramped over the wavenumber range 2050.2–2051.2 cm$^{-1}$ is coupled into the cavity, the light exiting the cavity on the opposite side is collimated onto a HgCdTe photodiode. Two highly reflective dielectric cavity mirrors allow for an effective path length of approximately 1000 m. Trace gas mixing ratios are retrieved from infrared spectra online using manufacturer Los Gatos software. In addition, raw spectra are saved every 15 seconds to allow for consistency and quality checks of the recorded data.

For the TAN1802 voyage, MICA was deployed in the temperature controlled aerosol container laboratory, alternating measurements of the marine boundary layer and the surface ocean at intervals of 10 minutes for air and 50 minutes for water using a fully autonomous setup that consists of a pump, switching valves and a spray-head seawater equilibrator (Lennartz et al., 2017). The intake of the airline was located at 20 m ASL at the starboard forward mast on the monkey island (Fig. 3). Seawater from about 5 m depth was supplied to the equilibrator at a flow rate of 2-3 dm$^3$ min$^{-1}$. To ensure that concentrations remain at

near equilibrium, the gas phase was constantly recirculated between the equilibrator headspace and MICA. A filter (PallAcro, 0.7 μm) was placed directly in front of the MICA inlet to remove particles and droplets. Teflon was used for all tubing, and materials known to cause OCS contamination such as rubber, were avoided. From gas phase mixing ratios in the equilibrated air, dissolved concentrations were calculated using Henry's law constants.

MICA was calibrated in the laboratory before and after the TAN1802 voyage, to ensure data quality, determine measurement accuracy and precision, and cross check correction functions accounting for some known non-linearities at low and high concentrations. OCS mixing ratios ranging from 0.25–5 ppb were prepared using permeation devices. In addition, a NOAA certified standard containing 450 ppt OCS was used to ensure consistency with data from OCS sampling networks. Instrument response was consistent for all standards with accuracy better than 30 ppt for mixing ratios below 750 ppt and 4 % for higher mixing ratios, translating to about 2 pmol dm$^{-3}$ for dissolved concentrations. Precision was determined by sampling the NOAA standard as 90 ppt at the nominal 1 Hz sampling rate and 15 ppt (about 1 pM) with 2 minute averaging, with no significant drifts observed at longer time scales over a 4 hour sampling period. CO mixing ratios in the range 10–2500 ppb and CO$_2$ mixing ratios in the range 10—2500 ppm were prepared from a 5±0.05 ppm CO and 5000±2.5 ppm CO$_2$ standard (Air Products) by dilution with clean Argon gas (containing no detectable CO and CO$_2$). Taking into account uncertainties of the standard and the dilution system, respective accuracies for CO and CO$_2$ are derived to be 10 ppb and 6 ppm, corresponding to 0.01 nmol dm$^{-3}$ and 0.1 μmol dm$^{-3}$ for dissolved concentrations, respectively. During the 4 hour experiment with the NOAA standard, which also contains CO and CO$_2$, a 1 Hz precision of 5 ppb for CO and 1 ppm for CO$_2$ was determined. This reduces to 1 ppb and 0.2 ppm respectively at the 2 minute temporal resolution at which MICA data for TAN1802 are provided. Pre- and post-campaign calibrations for all gases were in excellent agreement. Raw IR spectra recorded during the cruise were inconspicuous and consistent with the mixing ratio data retrieved online.

Two netCDF files are provided for the MICA measurements with data that have been processed and quality controlled as described above. One netCDF file contains atmospheric and dissolved OCS, CO and CO$_2$ concentrations at 2 minute temporal resolution (mean, standard deviation and standard error are given for each gas and each 2 minute time interval). The second netCDF file contains the OCS sea–air flux derived from the MICA measurements and wind speed using the sea–air gas exchange parameterisation of Nightingale et al. (2000) at 1 hour temporal resolution (see Table A1). Note that OCS observations from TAN1802 have already been included in a long term global data set of ship based observations of atmospheric and dissolved OCS published by Lennartz et al. (2020).

# 4 Data sets

## 4.1 Meteorological observations

Favourable meteorological conditions were encountered for much of the voyage with the entire study area (south of 60° S, hereafter referred to as the Southern Ocean) being free of sea-ice for the duration of the voyage. The ship faced a strong head wind (southerly) on the transect from 60° to 70° S. Overall, only three periods of enforced down-time occurred. The ship track together with sea surface temperature and underway sea surface salinity encountered during the voyage are shown in Fig. 6.

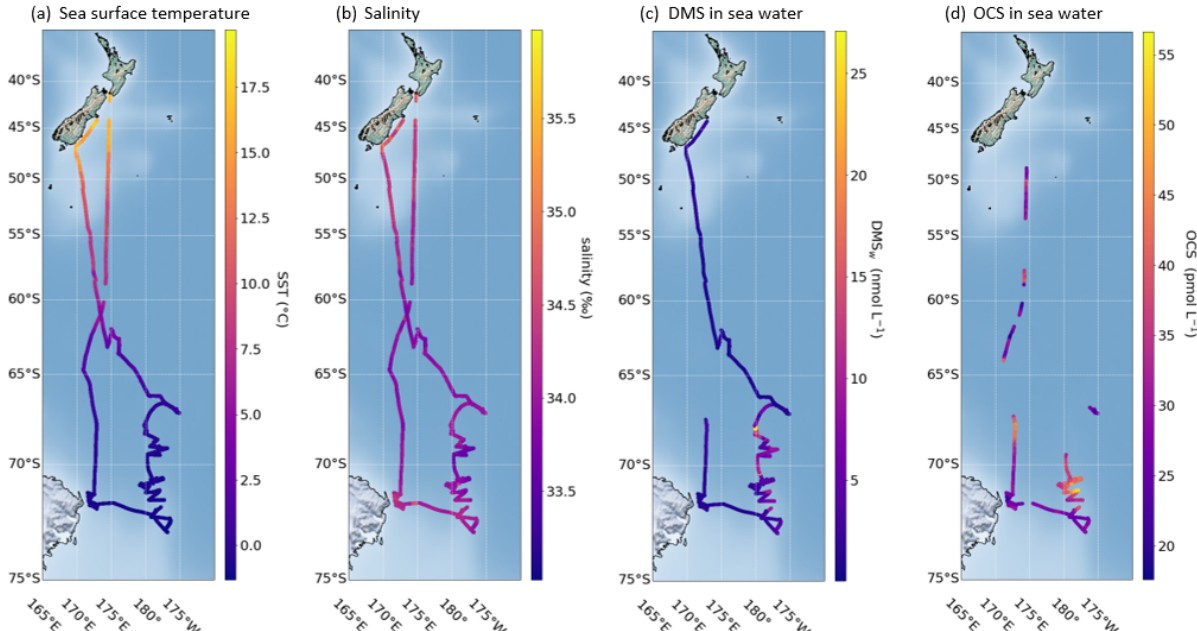

**Figure 6.** The track of the RV *Tangaroa* during the Southern Ocean Ross Sea Marine Ecosystem and Environment voyage (a) sea surface temperature, (b) salinity, (c) DMS measured in seawater, and (d) OCS measured in seawater. The measurements of DMS and OCS together with derived sea-air fluxes will be discussed in more detail in Sect. 4.3.1 and 4.3.2.

When reaching the Southern Ocean, the presence of sharp gradients in sea surface temperature (drop from >5 °C to about 0 °C) in proximity to the Antarctic Circumpolar Current (ACC) fronts is evident in the data. Salinity decreased from greater than 34.5 to 34 ‰ or less within the Southern Ocean, but increased close to the ice edge (to maximum of 35.9 ‰).

Time series of observed wind speed, pressure, relative humidity, temperature, sea surface temperature, and radiation along the complete voyage track are shown in Fig. 7. The vessel reached the Southern Ocean region on day five of the voyage (14 February 2018). The drop in air temperature and pressure when entering the Southern Ocean is clearly visible in Fig. 7(c, g). Over the Southern Ocean, air temperatures observed ranged mainly between +1 °C and -2 °C with a minimum of -7 °C (Fig. 7), with observed sea surface temperatures remaining around 0 °C. The median air and sea surface temperatures throughout the time spent in the Southern Ocean were -1.4 °C and -0.3 °C, respectively. The observed median wind speed at 10 m in the Southern Ocean was $9\,\mathrm{m\,s^{-1}}$ (interquartile range of 5.96), and the maximum wind speed at 10 m recorded in the Southern Ocean was $26\,\mathrm{m\,s^{-1}}$. The wind direction over the Southern Ocean corresponding to strong winds was mostly south and south/west as indicated by the wind barbs in Fig. 7(a). The southernmost latitude reached during the voyage was 73° S.

Figure 8 shows example temperature and relative humidity profiles between the ground and 17.5 km as measured by radiosondes, which were released south of 60° S. Fog events associated with moist air trapped near the surface by low-level temperature inversions are visible on 15 February and 5 March 2018 in the radiosonde data. The tropopause is also clearly

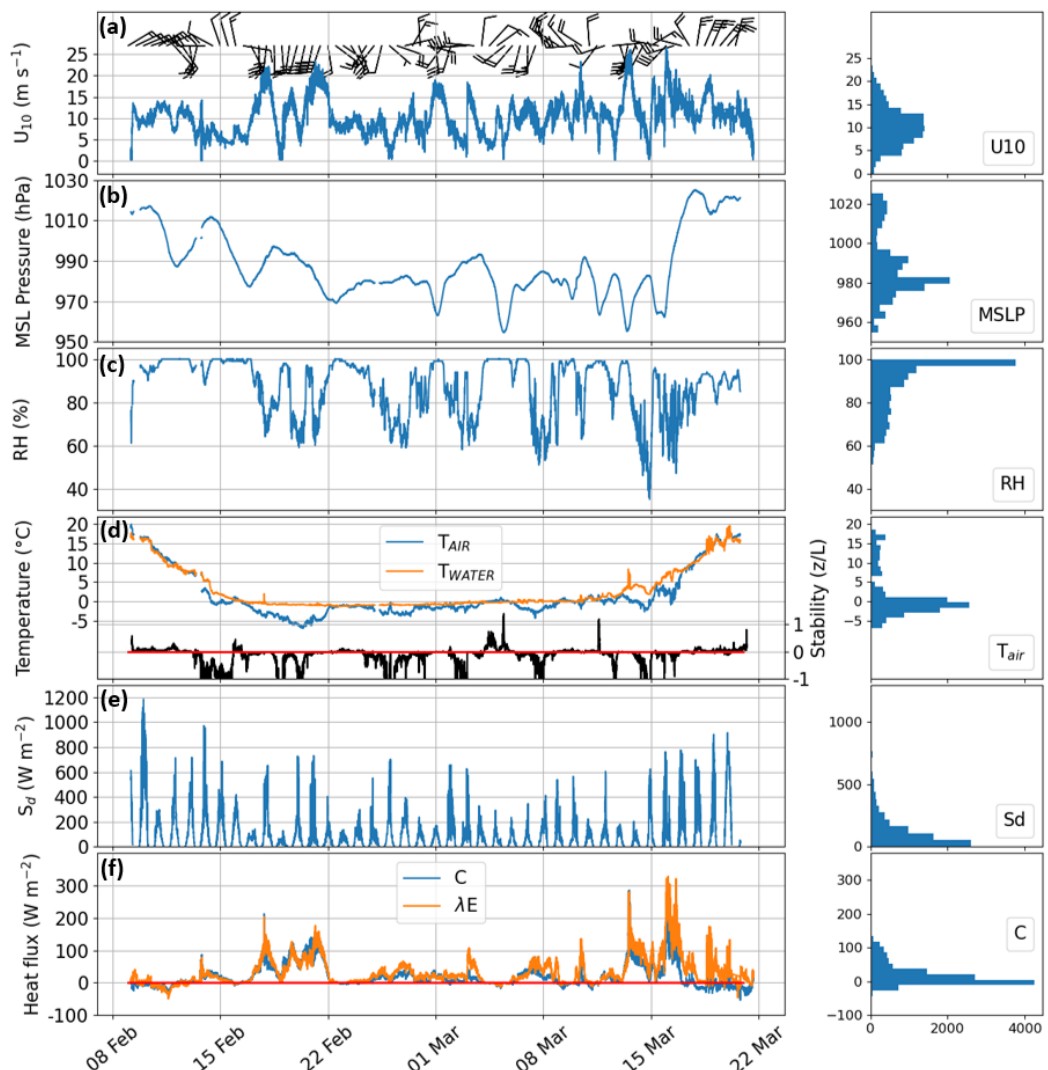

**Figure 7.** Summary of the meteorological conditions during TAN1802, showing (a) 12 hourly vector-averaged wind barbs for measured wind speed (barbs are in knots, 1 barb = 10 knots) together with the wind speed at 10 m ($u_{10}$), (b) mean sea level pressure (MSLP), (c) relative humidity (RH), (d) air temperature ($T_{air}$), sea surface temperature ($T_{water}$), and the Monin-Obukhov stability parameter ($z/L$) (black line), where $z$ is the height of wind measurement and $L$ is the Obukhov length scale (m); with positive and negative values indicating stable and unstable conditions in the lower atmosphere, respectively, and the zero line in shown in red, (e) short wave radiation ($Sd$), and (f) sensible ($C$) and latent ($\lambda E$) heat flux. Small panels on the right show the corresponding histograms for 5 minute resolution derived from all AWS measurements that were obtained during the voyage. The ship entered the Southern Ocean (region south of 60° S) on 14 February and leaving this region on 16 March 2018.

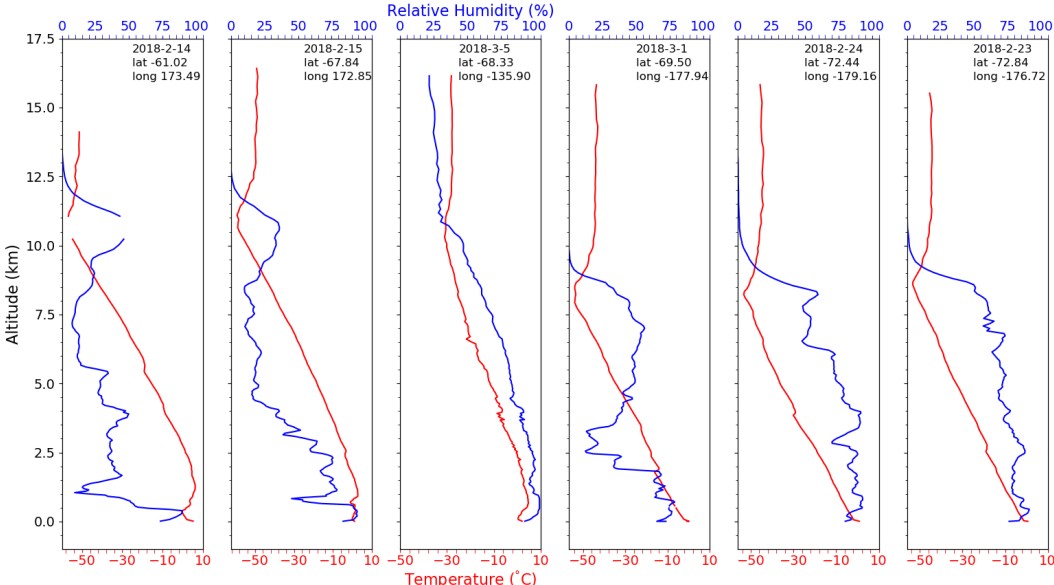

**Figure 8.** Example temperature and relative humidity profiles measured by the weather balloon radiosondes released during the voyage.

visible in the temperature profiles at around $11 \, \text{km}$ (15 February), dropping to between 8 and $8.5 \, \text{km}$ further south. Above the pronounced tropopause lies the stable and dry stratosphere with temperatures around -50 °C.

**Clouds and precipitation**

Observations of clouds were made throughout the voyage, with observations dominated by low-level cloud base heights and high cloud fraction, consistent with previous observations in the Southern Hemisphere (Protat et al., 2017; Klekociuk et al., 2020). Synoptic weather observations were performed throughout the voyage and were interpolated on regular 6-hourly synoptic intervals (00, 06, 12, 18 UTC), revealing that the most frequently observed cloud types were stratus (49 %), stratocumulus (25 %), and nimbostratus (26 %). The cloud fraction was derived by the allskypi system for solar zenith angles of less than 90°

with the mean over the voyage from south of 60° S being 92 %, with no single day averaging less than 79 %. Cloud fraction obtained from the ceilometer measurements (level 0 data product) compare well to the cloud fraction from the all-sky camera system with a mean of 0.95 for all measurements made south of 60° S. The occurrence of cloud fraction, in oktas, from allskypi and from the ceilometer is shown in Fig. 9.

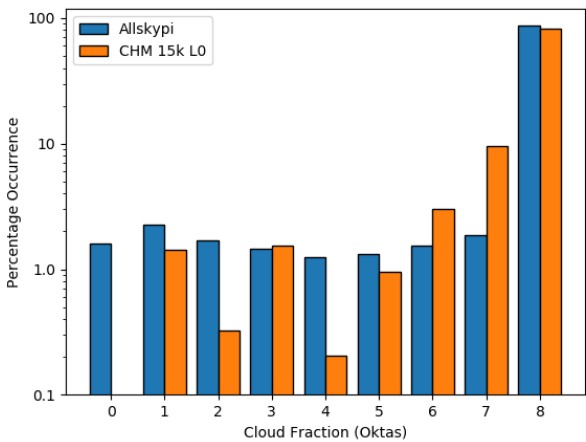

**Figure 9.** Cloud fraction derived from allskypi and the ceilometer (level 0 output), expressed in oktas, i.e. number of eighths of sky covered by cloud.

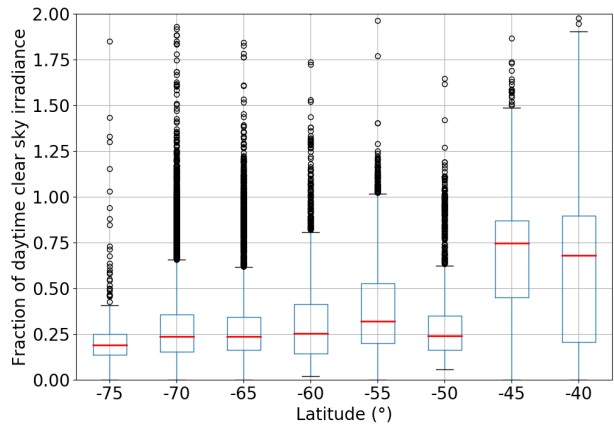

**Figure 10.** Box plot showing the upper to lower quartiles of the ratio of downwelling shortwave solar radiation (Sd in W m$^{-2}$) received by the Eppley Precision Spectral Pyranometers to the theoretical clear sky value calculated for an atmospheric transmission coefficient of 0.86, on a minute by minute basis throughout the voyage and categorised centred on 5° latitude bands (e.g. -40° category ranges from -42.5° to -37.5°). The number of data points included in each box plot are: 1629 (lat band -75°), 16032 (lat band -70°), 7837 (lat band -65°), 3325 (lat band -60°) and about 2000 data points collected in transit latitudes to the north.

Cloud observations are dominated by periods of complete cloud cover. In such cases, the all-sky camera and the ceilometer agree well with each other due to the lack of spatial variability (Fig. 9). When lower cloud fractions are observed and when there is spatial variability, the agreement between the ceilometer derived cloud fractions and the camera is reduced compared to events with complete cloud cover. This is due to the limited area that the ceilometer uses to compute cloud fraction compared to the camera system. The former uses a time weighted average of cloud occurrence to infer the spatial cloud fraction, essentially assuming that the spatial variability at a given moment is equivalent to the temporal variability over the preceding period. Furthermore, the difference can be also caused, in part, by the different geographical region sampled by the ceilometer and a sky camera (directly at zenith versus all-sky).

The on-board Eppley pyranometers were used to examine how the high occurrence of clouds affected the received solar radiation (Sd) compared to the expected clear sky value at the location of the vessel. The expected values were calculated from sun-earth geometry with beam and diffuse components. The clear-sky visible radiation transmission coefficient was taken as 0.86 in a simplified approach using a single value for the marine boundary-layer within the range expected (Longman et al., 2012) and verified by comparison to measurement on the rare few days with clear-sky around solar noon. Data were quality controlled, i.e. measurements made at low sun angles ($<3\,\mathrm{Wm^{-2}}$) and nighttime data were excluded. Ratios above 1 were rare and may be an error due to variation in the actual transmission coefficient with sun angle, aerosol loading, or from additional forward scatter off clouds present when the sun was not obscured. The ratio of measured to expected solar radiation is shown in Fig. 10, indicating a reduction in radiation received at the surface down to around a quarter of the anticipated clear sky value value due to the high prevalence of clouds (see Fig. 9) south of $50°$ S.

Using the cloud base height product derived from both the ceilometer and MiniMPL raw data using the ALCF tool (Kuma et al., 2020b), it is possible to look at the frequency of cloud occurrence binned by height in $200\,\mathrm{m}$ intervals, as shown in Figure 11. The observations clearly indicate that for the majority of the campaign period, there was a high abundance of low-level clouds, with over 95 % of the cloud base heights occurring at or below $1200\,\mathrm{m}$ and peaking below $200\,\mathrm{m}$.

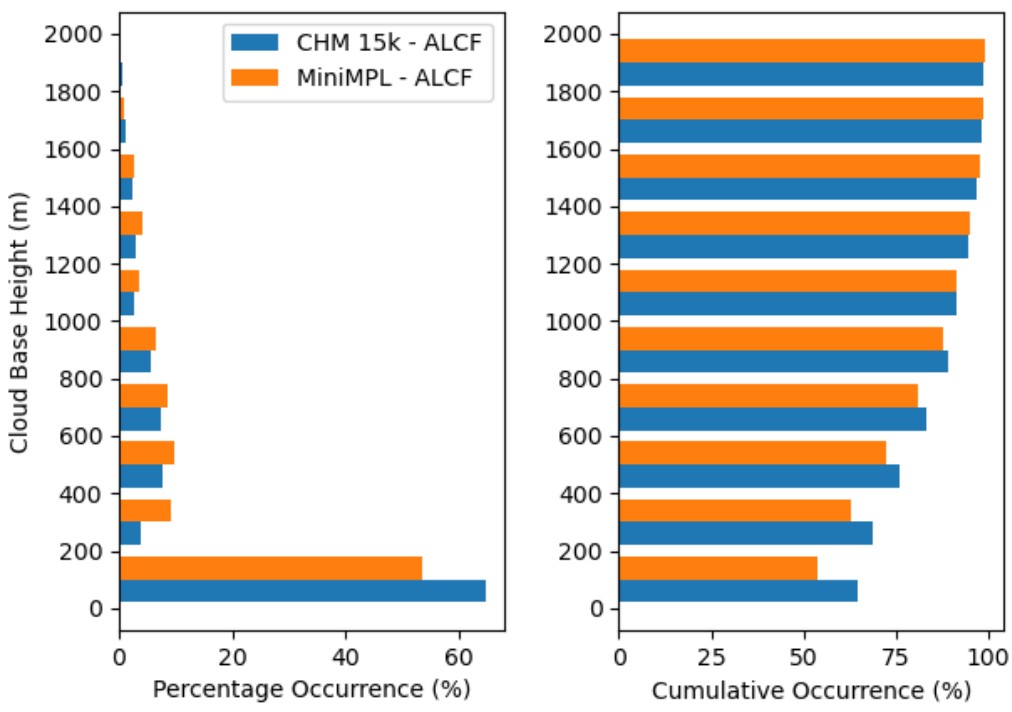

**Figure 11.** Histograms of cloud base height derived from the ceilometer and MiniMPL data processed by the ALCF software. The percentage occurrence (left) and cumulative occurrence (right) are shown for each 200 m bin, from 0 to 2000 m. Note that the near surface percentages can be affected by the incomplete overlap of the lidar in the first few hundred metres.

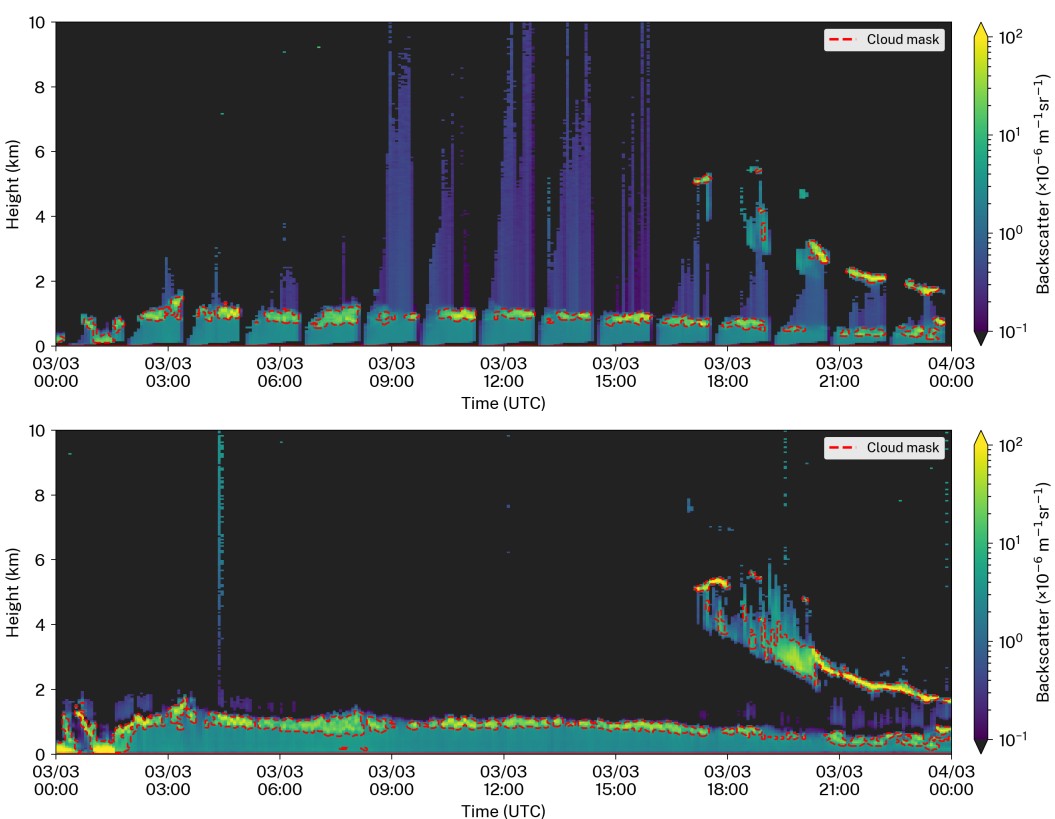

**Figure 12.** Automatically generated output plots from the ALCF tool showing backscatter ratios over 24 hours for 3 March 2018 for the MiniMPL data (upper panel) and ceilometer data (lower panel). Cloud mask is overplotted as dashed red lines. The periodic structure in the MiniMPL occurs because it was scanning over a range of elevation angles and so would saturate at lower altitudes when measuring at lower elevations angles (larger air masses occurring closer to horizon).

An example of the backscatter ratios measured by the two lidar instruments is shown in Fig. 12, wherein we demonstrate the differences in sampling between the two instruments. The MiniMPL was scanning over a range of elevation angles (see Sect. 3.3). The scans at lower elevation angles would saturate at a lower altitude due to the higher effective air mass being measured. Thus the periodic structure observed in the MiniMPL shown in the upper panel of Fig. 12, while the ceilometer, which did not have elevation scanning functionality and only measured in the zenith direction, shows a more continuous time-series. This particular day (3 March 2018), with its nearly unbroken cloud signal around 1 km, is representative of the overall cloud statistics from the voyage. The initial 2 hours (00:00–02:00 UTC) show surface level cloud or fog (Fig. 12). From 02:00 to 18:00 UTC, low level cloud between 1 and 1.5 km is present. At 18:00 UTC, in addition to the low level cloud, a higher cloud layer at 5 km is observed along with probable precipitation as it descends to 2 km by the end of the 24 hour period being shown. A challenge with measurements of this type is that there may also have been other high cloud layers throughout the day but they were not seen through the saturated low level cloud layer.

Precipitation was monitored throughout the voyage, but with relatively low occurrence throughout. Figure 13 displays the radar reflectivity, vertical velocity and spectral width for a range of altitudes over one 24 hour period collected near 71° S derived using the scheme detailed in Maahn and Kollias (2012). Figure 13 also displays snowfall estimates at the surface derived from the MRR-2 data. Note that the corresponding in situ precipitation measurement device on RV *Tangaroa* was not sensitive enough to snowfall to measure these very small rates of accumulation. The diagonal structures identified in Fig. 13 between approximately 19:00 UTC on 16 February and 01:00 UTC 17 February 2018 at altitudes above 2 km in the radar reflectivity are related to fall streaks, which represent the movement of precipitation towards the surface. The upward and downward motions observed in Fig. 13(b) are a distinctive characteristic of snowfall.

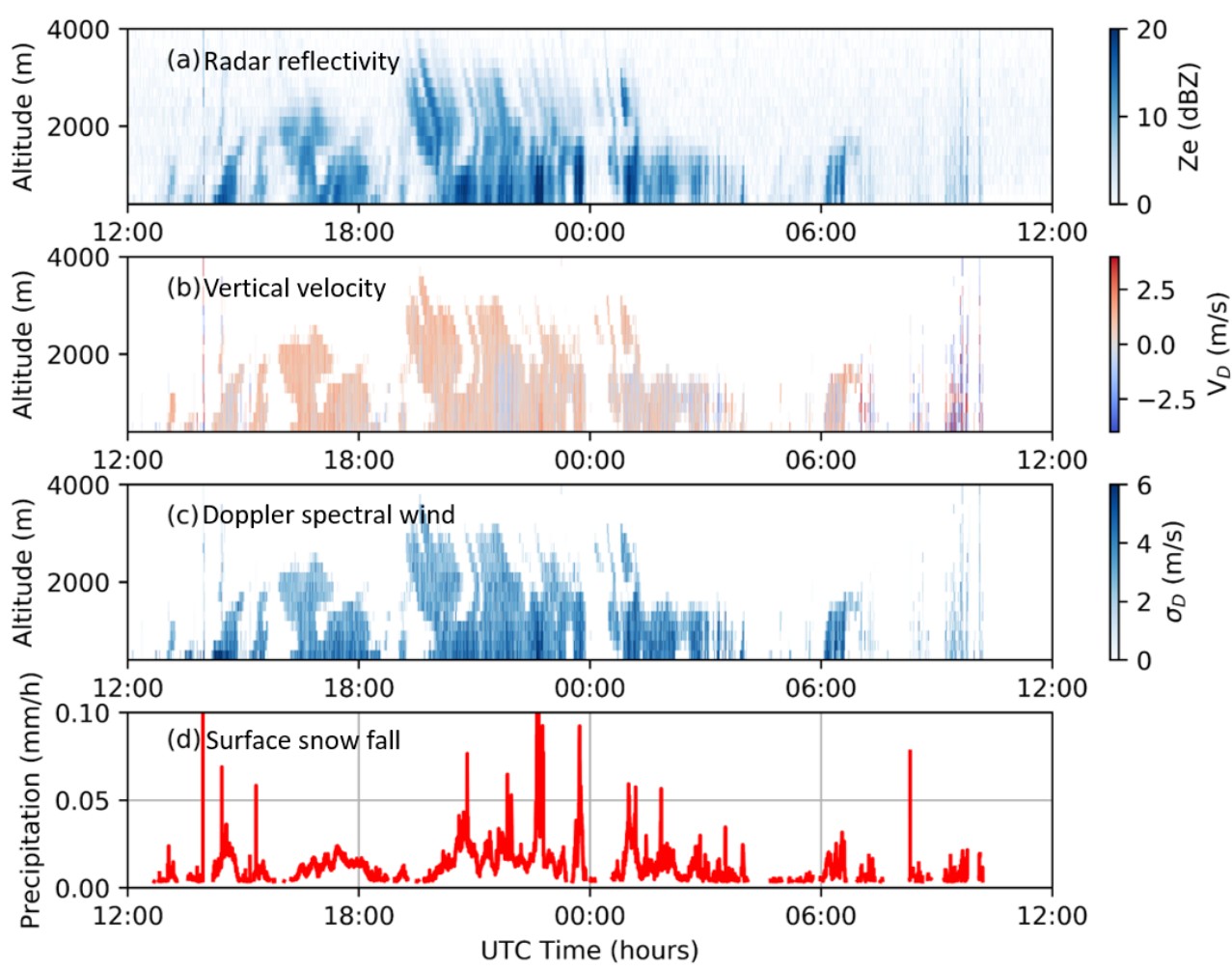

**Figure 13.** Altitude versus time contour plots of the (a) radar reflectivity, (b) vertical velocity, and (c) Doppler spectral width derived from the MRR-2 Doppler spectra over 24 hours between 16 February 2018 12:00 and 17 February 2018 12:00 UTC derived using the schemes detailed in Maahn and Kollias (2012). The corresponding empirical estimate of surface snowfall is identified in panel (d).

## 4.2 Particle size distributions and cloud condensation nuclei

### 4.2.1 $CO_2$ measurements for identifying contamination events

Throughout the voyage the mole fractions of atmospheric $CO_2$ were measured continuously by using a Picarro CRDS (Sect. 3.7). While the sampling line of the Picarro was separate to the particulate sampling line, it was in close proximity (within 5 m). Contamination from ship exhaust from the rear of the ship would have been sufficiently well-mixed in the turbulent air around the ship superstructure to affect both sampling lines. The use of $CO_2$ measurements together with wind speed and direction measurements are often used as a reliable method to identify periods of contamination in the air sampled by the sampling inlet for all aerosol measurements performed in the aerosol container lab.

Five minute mean $CO_2$ measurements for the entire voyage are shown in Fig. 14. Following an initial high value at the start of the voyage, due to proximity to land (Wellington), atmospheric $CO_2$ concentration rapidly decreased to close to the baseline value of 403 ppm, which was observed at NIWA's Baring Head atmospheric station at the time of the voyage. This baseline value is consistent with the voyage being conducted within the Southern Ocean/Antarctic source region for air selected for baseline analysis at Baring Head (Brailsford et al., 2012).

A large number of brief episodes of high $CO_2$ concentration (to >500 ppm) are apparent in the $CO_2$ data set shown in Fig. 14. These are attributed to contamination from the exhausts of the ship's engine and Dynamic Positioning System (DPS) generators being blown back towards the airline intake above the bridge in certain wind conditions. During the voyage the DPS was operated during Deep-Towed Imaging System (DTIS) deployments. Two tests were used to identify these exhaust contamination events in the Picarro $CO_2$ data, i.e. $CO_2$ measurements were deemed as pollution events if:

1. the $CO_2$ standard deviation of the 5 minute mean was greater than 0.1 ppm, AND

2. the $CO_2$ 5 minute mean was more than 0.1 ppm above the calculated 50-point median filter that was applied to the $CO_2$ 5 minute mean data.

In Fig. 14, these exhaust contamination conditions are indicated in red, while data considered as good are indicated in blue. The exhaust contamination tests are effective in identifying all of the data points attributable to exhaust contamination, at the expense of including a small number of points that may be considered good data.

All particle measurements described below were screened according to contamination events using the method described here or in their respective sections above.

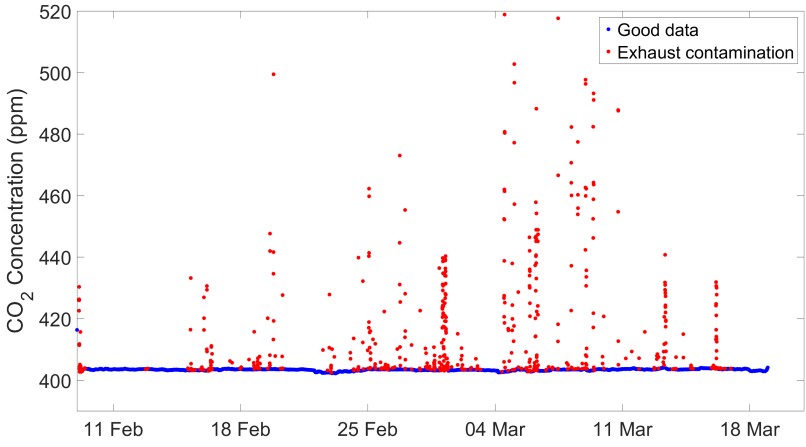

**Figure 14.** Calculated 5 minute mean of atmospheric $CO_2$ concentration measured by the Picarro. Colour coding indicates quality controlled data representative of baseline air or data points flagged as ship exhaust contamination.

### 4.2.2 Particle size distributions

As described in Sect. 3.8.1, a reasonably rigorous data quality procedure was implemented to screen the continuous aerosol observations for potential contamination by ship exhaust. The percentage of PCASP-100X measurements that have been removed due to ship exhaust contamination is shown in Fig. 15(a), while Fig. 15(b) shows the time series of the total concentration of particles observed by the PCASP-100X. These values represent the concentration of particles in the size range 0.1–3 µm. In Fig. 15(b), the quality-assured time series is overlaid by the measurements that were removed according to the data quality procedure. Overall, the procedure was highly successful in removing contaminated aerosol observations; however, the procedure also removed a small number of useful observations.

The quality assured measurements from all of the aerosol instruments operated throughout the voyage are shown in Fig. 16a–e. In Fig. 16a, the time series of the PCASP-100X observations is shown. Note that the concentrations in each size range have been normalized by the log width of the size bin. A notable instrument artefact within the PCASP-100X measurements is the persistent local peak in concentrations between 0.5–0.6 µm. Similar to the lack of particles observed in the 5[th] size bin (see Sect. 3.8.1), this is likely a result of gain stitching errors between the multiple linear amplifiers the PCASP uses to detect particles across such a broad range of sizes. The user may choose to exclude this size bin in further analysis.

In Fig. 17, the median particle concentration size spectrum measured by the PCASP-100X, SMPS, and NAIS is shown for the whole voyage. This spectrum can be used to compare particle concentration measurements between the various particle counters. Overall, there was reasonably good agreement between the particle size distributions measured by the PCASP-100X and SMPS 3936. However, on average the PCASP reported 1.6 times as many particles in the 100-300 nm range as the SMPS, and it is recommended that the SMPS data are used in this size range.

In Fig. 17, it appears that there is significant disagreement between the SMPS3936 and the NAIS in the 10–15 nm particle size range. However, this is most likely a result of additional deposition of these particles within the sampling conduit and inefficient transmission through the SMPS itself. The NAIS measurements, which were conducted from the mast of the ship, are likely more accurate in this size range. The SMPS data for particles smaller than 20 nm are available, but should be interpreted with caution.

Finally, while a median of all aerosol size distributions was shown in Fig. 17, many different types of air masses were encountered throughout the voyage. In Fig. 18, air mass back trajectories presented in Hartery et al. (2020b) were exploited to calculate the fraction of time air masses spent over different geographic regions. These data will help researchers contextualize our observations and enable better cross-comparison with other studies. For instance, on 17 February 2018 there was an abrupt change in the air mass as the vessel arrived at Cape Adare. The change in air mass resulted from a prompt switch to southerly winds upon our arrival (Fig. 7). This resulted in a rapid increase in the number of accumulation mode particles and CCN (see Figs. 16a, b and d). However, it is difficult to attribute the change in CCN and accumulation mode particulate to the change in air mass alone, as prior to 17 February the ship was surrounded by a near-continuous fog. Accumulation and coarse mode particulate are scavenged when they activate as cloud droplets during fog formation. It is likely that the presence and subsequent absence of fog is the dominant driver of changes to the particle size distribution on 17 February, with a change in air mass providing a secondary influence. As fog occurred frequently throughout the voyage (Kuma et al., 2020a), care is needed when interpreting the air mass results presented in Fig. 18 to properly disentangle these two effects.

Overall, we expect these measurements to be of great value to the scientific community as they cover the entire particle size spectrum. Complementary to Hartery et al. (2020b), the combination of sub- and super-micron particles can be used to test existing sea spray emissions parameterizations, or derive new ones specially adapted to the Southern Ocean. In particular, the role of biological processes on sea spray emissions and properties for seawater temperatures and phytoplanktonic populations specific to the Southern Ocean can be explored using this data set. Moreover, the combination of these total aerosol size spectra with the cloud condensation nuclei spectra could potentially be used to investigate particle activation within high-latitude, low-level marine stratocumulus. In addition, a new particle formation event was observed on 11 February 2018 as the RV *Tangaroa* left the continental shelf. This event is highlighted in Fig. 19 and could be studied in further detail to better understand the conditions which favour new particle formation.

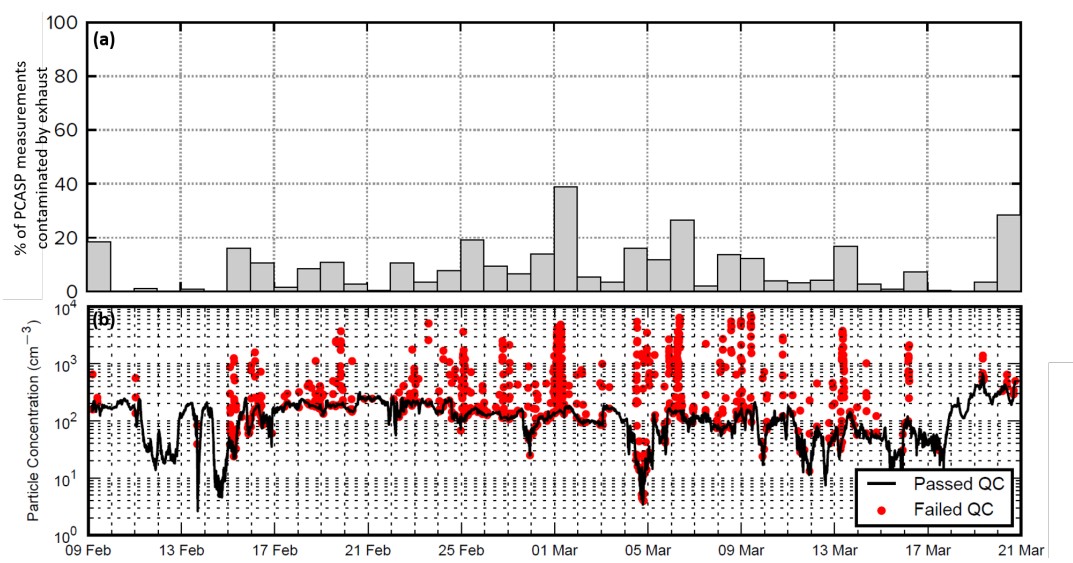

**Figure 15.** (a) Percentage of PCASP measurements that were contaminated by ship exhaust. (b) Comparison of raw (red) particle concentration and quality-controlled (black) data after a quality procedure to remove potentially contaminated aerosol samples was implemented based on wind direction, the co–incident $CO_2$ time-series and Poisson counting statistics. This procedure is described in detail in Sect. 3.8.1.

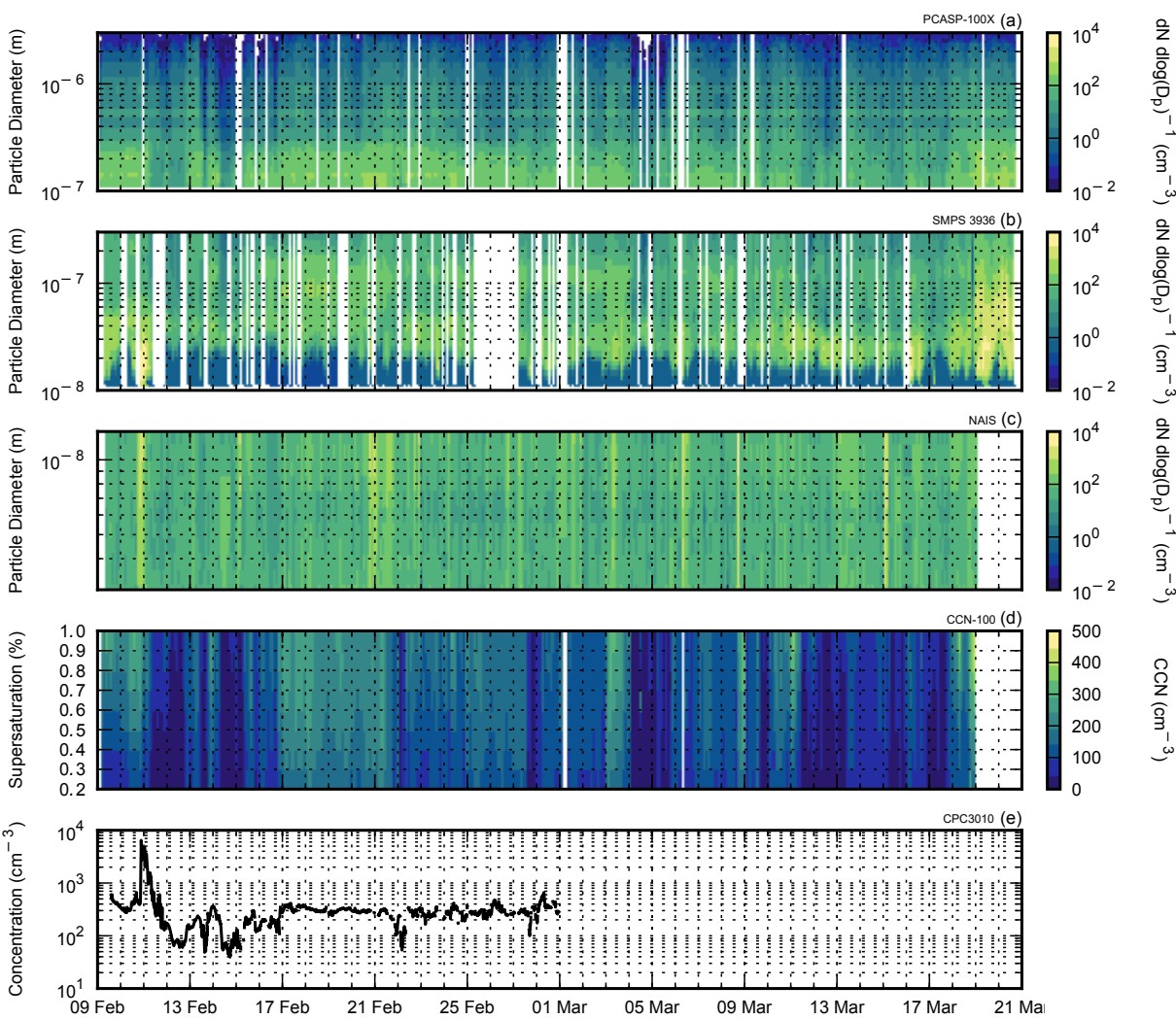

**Figure 16.** Spectral and total particle concentrations measured by various particle counters throughout the Southern Ocean Ross Sea Marine Ecosystems and Environment voyage. Data gaps resulted from intrusions of polluted air into the sampling line or from instrument/system errors. (a) PCASP-100X, (b) SMPS 3936, (c) NAIS, (d) CCN-100, and (e) CPC3010. See Sect. 3.8 for more details.

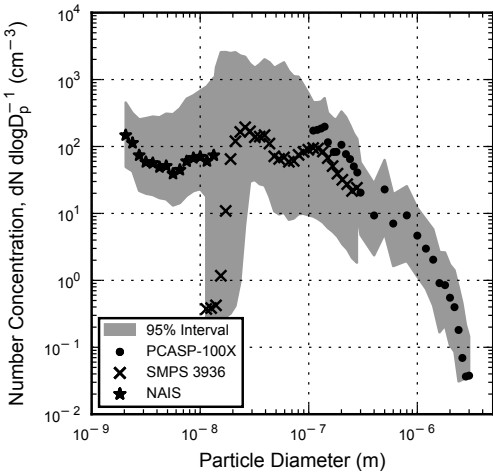

**Figure 17.** The median particle size distribution measured by the PCASP-100X, SMPS 3936, and NAIS across the entire voyage. Note that the spectrum above 3 µm is not shown and was not evaluated in this study.

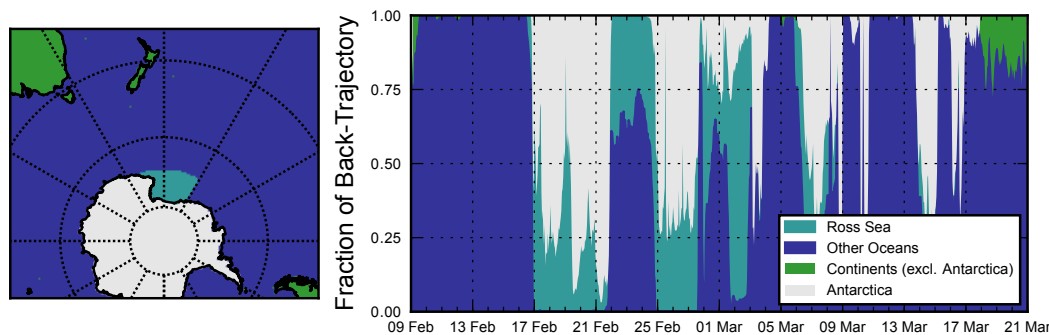

**Figure 18.** Left: Geographic regions (as indicated by the legend) used for the back-trajectory modelling over the Southern Ocean. Right: The fraction of time air masses spent over different geographic regions in the five days prior to the measurements. The back-trajectory modelling is described in detail in Hartery et al. (2020b).

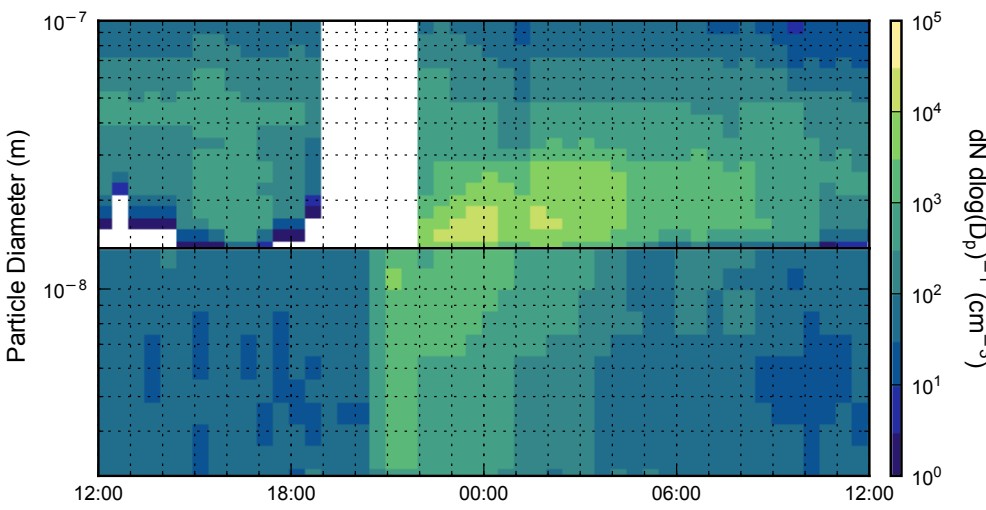

**Figure 19.** Same as Fig. 16(b, c) but for one day only. On 1 February 2018 at approximately 21:00 UTC, a new particle formation event that was detected by the SMPS (top) and NAIS (bottom) spectrometers.

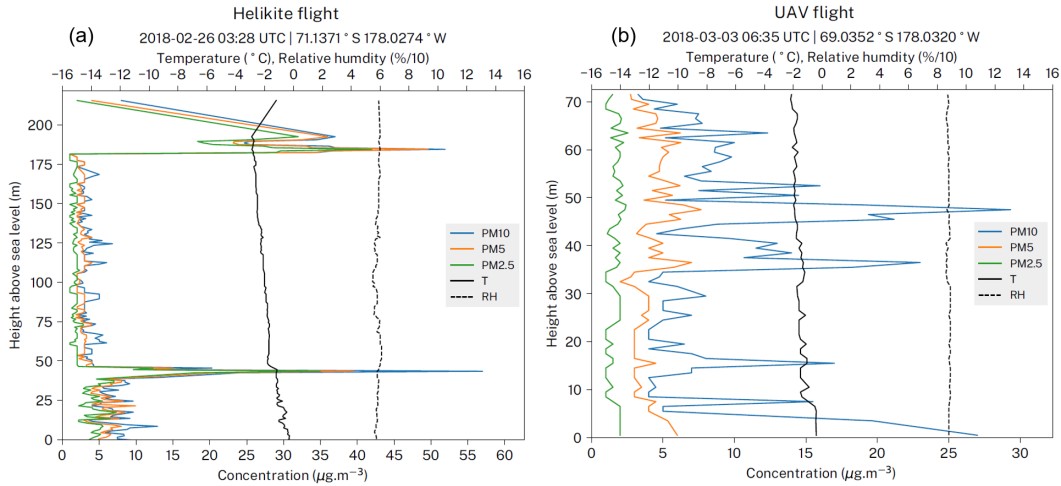

**Figure 20.** In situ boundary layer aerosol concentration, atmospheric temperature and relative humidity in two locations in the Ross Sea sampled with (a) helikite flight and (b) UAV flight. The sampling was performed using an Alphasense OPC-N2 unit connected to a radiosonde which were attached to the aircraft. Shown are the particulate matter (PM) concentration $PM_{2.5}$, $PM_5$ and $PM_{10}$, temperature (T) and relative humidity (RH). Height is based on the GPS coordinates of the radiosonde.

### 4.2.3  $PM_{10}$, $PM_5$ and $PM_{2.5}$ measurements

On two occasions we sampled in situ boundary layer aerosol concentrations with a UAV and a tethered helikite up to a height of about 70 m and 200 m, respectively (Sect. 3.1.2 and 3.1.3). Particulate matter (PM) concentration, temperature and relative humidity as a function of height measured during these flights are shown in Fig. 20. Background concentration observed were
about $2\,\mu g\,m^{-3}$ during the UAV flight and $5\,\mu g\,m^{-3}$ during the helikite flight, decreasing with height to about $1\,\mu g\,m^{-3}$ at 70 m and $2\,\mu g\,m^{-3}$ at 175 m on the respective flights. The spikes in PM10 concentration of up to $25\,\mu g\,m^{-3}$ and $50\,\mu g\,m^{-3}$ in Fig. 20 suggest data contamination by ship exhaust. In particular, the helikite profile is affected by two spikes at about 40 m and above 175 m. Despite this limitation, these measurements are valuable due to the scarcity of airborne in situ aerosol concentration measurements in this region. Only two successful flights were conducted due to adverse weather conditions,
which prevented flying the UAV or helikite during most of the voyage.

With the flights performed, we demonstrated the use of UAV and helikites to sample the atmosphere and showed that it is possible to use these aircraft for measurements over the Southern Ocean. However, the deployment is limited by strong winds, high swell, and low temperatures, which limit the battery lifetime. Despite that, UAVs and helikites provide useful means to measure aerosol concentration in the boundary layer, which cannot be easily measured with other methods.

### 4.2.4 Ice nucleating particle concentrations

The interactions between aerosols and clouds are some of the least understood atmospheric processes, especially those involving ice nucleating particles (INPs), which facilitate cloud ice formation. By triggering primary ice at temperatures above the homogenous freezing point (about -38°C), INPs strongly affect cloud reflectivity, longevity, and the initiation of precipitation. To measure the low concentrations of INPs expected over this region we chose long-period filter-based collections. Further, to obtain the desired detection limit of 0.0001 INPs $L^{-1}$ we re-suspended filters in as small a volume of water as practicable and tested the bulk of it for INPs. The immersion freezing device used here is designed to process relatively large aliquots of suspension over a wide temperature range (see Section 3.9).

INP concentrations in four latitude bins measured during the TAN1802 are shown in Fig. 21. South of 50° S, the INP concentrations were consistently low: $\leq 0.1\,\mathrm{m^{-3}}$ at -15° C, typically 0.2–1 $\mathrm{m^{-3}}$ at -20° C, and typically 5–25 $\mathrm{m^{-3}}$ at -25° C. These concentrations fall in the lower half of the range measured in samples taken over the same period during voyages of the RV Aurora Australis from Hobart, Tasmania to the Australian Antarctic Division base at Mawson (Antarctica) and to Macquarie Island (DeMott et al., 2018) - using identical sampling and measurement protocols. The observed INP concentrations are also comparable to those recorded south of Tasmania in March and April 2016 (McCluskey et al., 2018), and to INP concentrations at -15° C from samples taken during TAN1502 in the same general regions as TAN1802 (Welti et al., 2020). Welti et al. (2020) also recorded similarly low INP concentrations at -15° C during the Antarctic Circumnavigation Expedition of 2016-2017. By contrast, the original work of Bigg (1973) found much higher levels, of about 14 INPs $\mathrm{m^{-3}}$ at -15° C, in the same region of the Southern Ocean as traversed during TAN1802.

The three filters taken as the ship cruised between 40 and 50° S had markedly higher INP concentrations (Fig. 21). All had sampled air that had passed over New Zealand for part or most of the sampling period (results from back trajectories predicted using The HYbrid Single Particle Lagrangian Integrated Trajectory (HYSPLIT) model applying the Global Data Assimilation System at a grid resolution of 0.5°, and initiated at 50 m ASL).

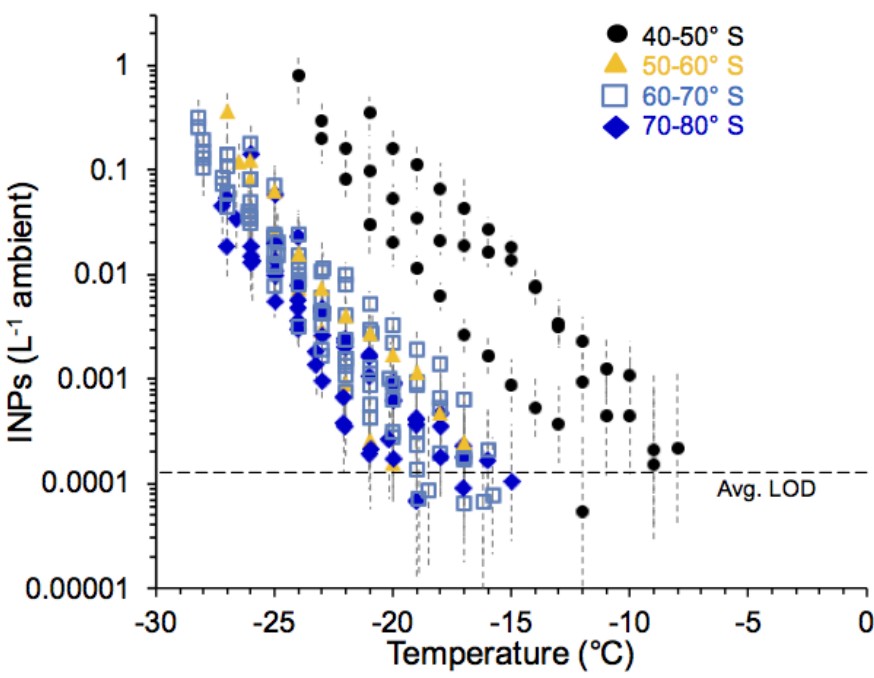

**Figure 21.** Cumulative spectra of ice nucleating particle (INP) concentrations in the boundary layer measured during the voyage. The spectra are divided by latitude to highlight the uniformly low INP concentrations south of 50° S and the raised levels in samples taken when the ship was east of New Zealand. LOD stands for limit of detection.

### 4.3 DMS and OCS observations

#### 4.3.1 Surface seawater DMS and derived fluxes

Dissolved DMS concentrations in surface seawater as measured during the TAN1802 voyage along the ship track are shown in Fig. 6(c). Throughout the voyage, DMS measurements obtained from the SCD (underway) and the data obtained from the CTD rosette sampler bottles at $10\,\mathrm{m}$ (not shown) were in good agreement and DMS profiles obtained from the CTD samples generally showed a near-surface maximum with lowest concentrations at or just below the surface mixed layer. Throughout the majority of the voyage, observed dissolved DMS concentrations in surface water were low between about 0.06 and 2 nmol $\mathrm{L}^{-1}$. The highest DMS concentrations were measured in the Eastern Ross Sea, in the transect between Iselin Bank and Scott Island, with a maximum concentration of $27\,\mathrm{nmol\,L^{-1}}$ (Fig. 6).

The DMS sea–air flux estimates ($F_{DMS}$) were derived by applying the COARE gas exchange coefficient for DMS to the DMS gradient at the ocean surface ($\Delta DMS$):

$$F_{DMS} = k_{DMS,COARE} \times \Delta DMS \tag{1}$$

were $\mathrm{k}_{DMS,COARE}$ is the gas exchange coefficient for DMS. The sea–air DMS concentration difference $\Delta DMS$ is equivalent to:

$$\Delta DMS = DMS_w - \frac{DMS_a}{H_{DMS}} \tag{2}$$

where $H_{DMS}$ is the temperature dependent dimensionless Henry's law solubility coefficient for DMS (Dacey et al., 1984), $DMS_w$ is the measured DMS concentration in seawater and $DMS_a$ is the DMS concentrations in air. The transfer velocity $k_{DMS,COARE}$ was calculated using the NOAA COAREG version 3.6 algorithm (Fairall et al., 2003, 2011; Blomquist et al., 2006) and parameterized in terms of local wind speed scaled to $10\,\mathrm{m}$ height as described in Bell et al. (2015). The transfer velocity $k_{DMS,COARE}$ was then adapted for DMS using the Schmidt number for local seawater temperature and salinity at $6.0\,\mathrm{m}$ depth (Saltzman et al., 1993). For the flux calculations the $DMS_a$ concentrations were set to zero as the atmospheric concentration is negligible compared to the concentrations in the ocean surface (ppt to $\mathrm{nmol\,L^{-1}}$).

As shown in Fig. 22 the transfer velocity is strongly dependent on wind speed. There is a positive correlation for the data set as a whole, with the transfer coefficient exhibiting the largest values at high wind speeds. As the sea–to–air DMS flux, shown in Fig. 23, depends on the DMS seawater concentrations, its distribution is very similar to that of dissolved DMS. The maximum DMS flux in the Southern Ocean is $69.4\,\mathrm{\mu mol\,m^{-2}d^{-1}}$, which corresponds to the maximum DMS concentrations measured in the Eastern Ross Sea. Yang et al. (2011) calculated an averaged DMS sea–to–air flux using the eddy covariance method of $2.9\pm2.1\,\mathrm{\mu mol\,m^{-2}\,d^{-1}}$ derived from measurements made during the Southern Ocean Gas Exchange Experiment voyage, North of the Weddell Sea in March–April 2008. The median flux estimated here for the summertime Southern Ocean (south of 60° S) was of similar magnitude with $3.57\,\mathrm{\mu mol\,m^{-2}d^{-1}}$.

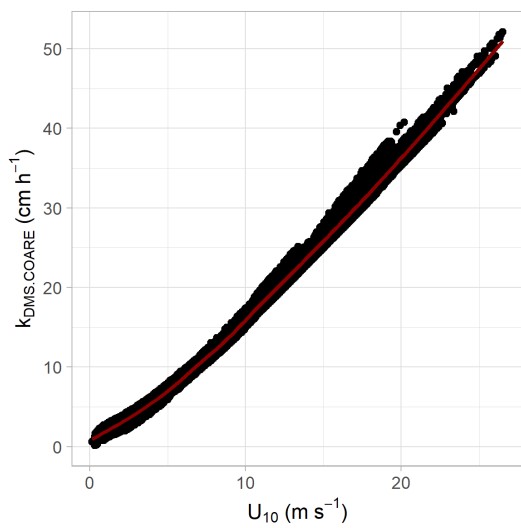

**Figure 22.** DMS gas transfer velocity $k_{DMS.COARE}$ against the horizontal wind speed at $10\,\text{m}$. The red line represents a spline fit to the data.

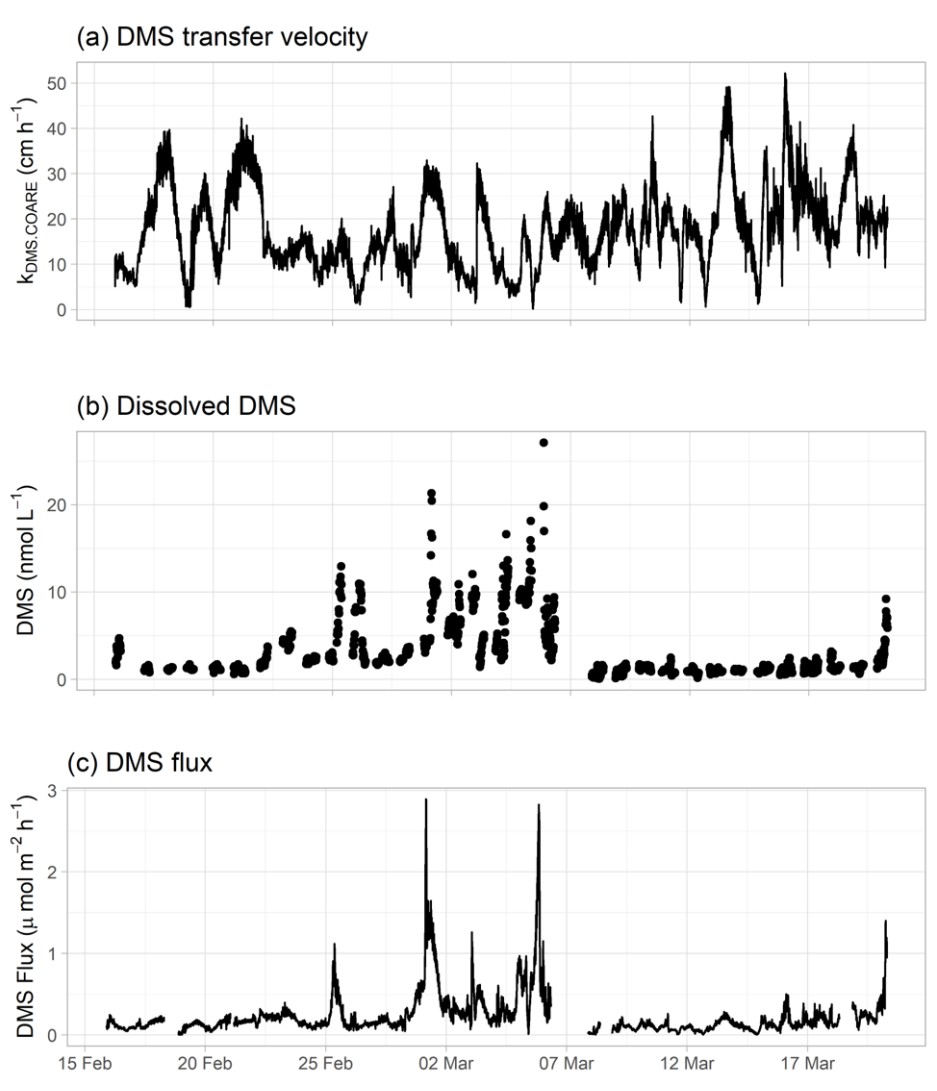

**Figure 23.** Time series of (a) DMS transfer velocity, (b) dissolved DMS concentration and (c) estimated DMS flux to the atmosphere plotted against time in UTC.

### 4.3.2 Surface seawater and atmospheric OCS

MICA operated from 9 February until 10 March 2018, when it was turned off after a cavity pressure drop caused by an internal blockage that could not be fixed at sea. Several measurement gaps between 9 February and 16 February were caused by power and seawater supply issues. Between 1 March and 8 March, salt jammed the 3-way switching valve in the air sampling position until the valve was cleaned with millipore water and ethanol. Quality assured dissolved OCS concentrations in surface seawater as measured during the TAN1802 voyage along the ship track are shown in Fig. 6(d).

MICA observations for the period 16 February to 1 March 2018, the time period without significant interruptions in either air or seawater sampling, are shown in Fig. 24. For OCS, atmospheric mixing ratios remain nearly constant around 500 ppt and dissolved concentrations vary between 20 and 60 pmol dm$^{-3}$. OCS is nearly always supersaturated and follows a characteristic diel cycle of a photochemically produced gas. Within the region sampled between 16 February to 1 March, uncalibrated fluorescent dissolved organic matter (fDOM) data from an in-line sensor show diel variability but low spatial variability during this period (data not shown). fDOM refers to the fraction of CDOM (chromophoric dissolved organic matter) that fluoresces. As expected, Fig. 24 shows a relationship between OCS concentration and irradiance, as CDOM is the main precursor to OCS photoproduction (Ferek and Andreae, 1984). Besides photoproduction, wind speed (red line in Fig. 24) is a key driver to the observed variability of integrated daily fluxes (grey bars and numbers in Fig. 24). Daily fluxes are derived using the sea–air gas exchange parameterisation of Nightingale et al. (2000), and were integrated from 12:00 pm to 12:00 pm UTC. In the cold sub-Antarctic waters, the strongly temperature dependent OCS hydrolysis (Elliott et al., 1989) becomes slow with a lifetime of several days, and sea–air exchange becomes the dominant OCS removal process in the surface seawater. This explains the observed behaviour of dissolved OCS concentration with high supersaturation, only building up at low to moderate winds when photoproduction is greater than removal, and high OCS fluxes often coincide with lower seawater concentrations on windy days. Observations from the TAN1802 voyage will be used to assess whether the behaviour of OCS in the Southern Ocean is adequately represented by a state of the art photochemical model (Lennartz et al., 2017). A specific model setup forced with high resolution observations made during the cruise will help to improve and fine tune the model.

Besides OCS, MICA also measured CO and $CO_2$ with spikes related to contamination by the ship's exhaust are removed from the data set. Atmospheric CO mixing ratios are, on average, 27 ppb. which is 10–20 ppb lower than expected even for the pristine air in this region (e.g. Novelli et al., 1998). While we can not irrevocably rule out an artifact, we found no indication in the raw spectra or during calibrations for a measurement error beyond the 10 ppb accuracy. Dissolved CO concentrations in the nM range agree with earlier CO measurements in the Southern Ocean (Williams and Bainbridge, 1973; Swinnerton and Lamontagne, 1974; Bates et al., 1995; Wingenter et al., 2004). CO is also photochemically produced from CDOM (Wilson et al., 1970; Stubbins et al., 2006), but the low amplitude of the diel cycle and the sustained high supersaturation ratios of 10–80 even on days with high wind and moderate irradiation suggest significant production mechanisms in addition to photochemical production. Atmospheric $CO_2$ mixing ratios were close to 400 ppm throughout the cruise, which agrees with the Picarro measurements (Sect. 4.2.1) within uncertainties.

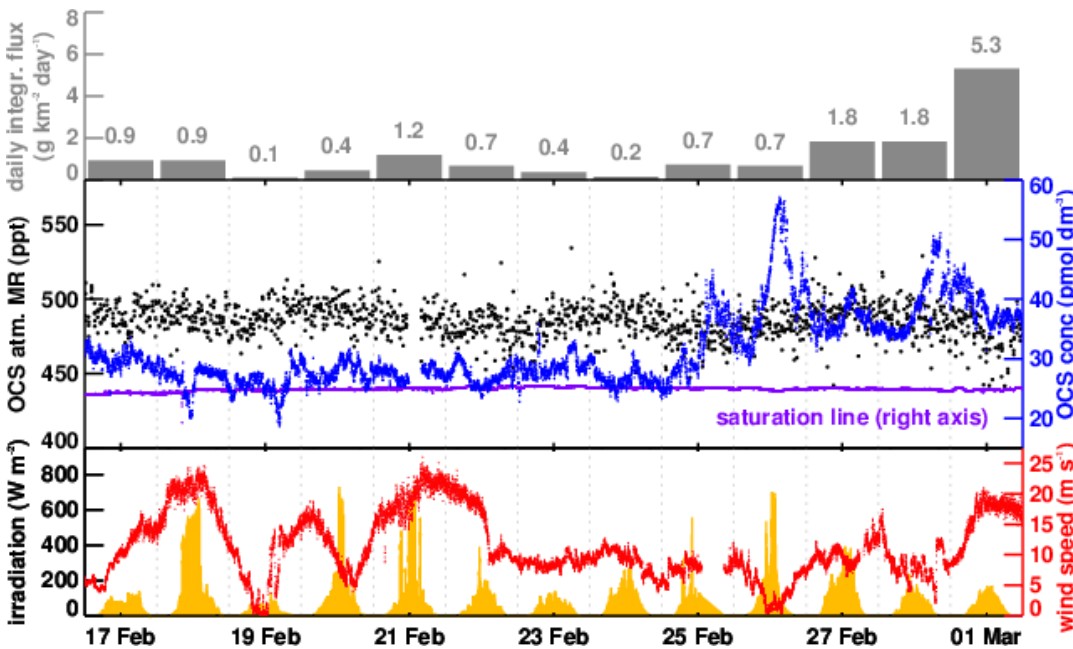

**Figure 24.** Time series of OCS atmospheric mixing ratios (black) and dissolved concentrations (blue) measured by MICA between 16 February and 1 March; data for other times are available but with significant interruptions (for more detail see text). Also shown are approximate saturation thresholds calculated from mixing ratios and seawater temperature (purple) in the same panel. The top panel shows the daily integrated sea–to–air OCS fluxes in $\mathrm{g\ km^{-2}\ day^{-1}}$ while the bottom panel shows the wind speed (red) and irradiation (yellow).

## 5  Summary

Ground-based and ship-based measurements of cloud and aerosol properties over the remote Southern Ocean are sparse such that satellite-based measurements are the primary source of data in this region. However, satellite-based measurements are inherently limited in their utility in several ways, e.g., while CCN concentrations can be indirectly estimated, they can not be accurately determined from satellite-based measurements. As a result, many questions can only be addressed using in situ and remote sensing ground-based and/or ship-based measurements that observe the atmosphere from below. Incomplete understanding of aerosol–cloud interactions over the Southern Ocean leads to a misrepresentation of aerosol and clouds processes in climate models. Such misrepresentations are manifested as biases in the representation of precipitation and radiation by climate models over the Southern Ocean.

A comprehensive description of meteorological, aerosol, cloud and precipitation measurements, made using a suite of sensors on board the New Zealand research vessel Tangaora during a six-week voyage over the Southern Ocean during February and March 2018, has been presented above. These ship-borne measurements are an important supplement to satellite-based measurements as they provide data on low-level clouds and aerosol composition in the marine boundary layer that cannot be inferred from satellite-based measurements alone. As such, the ship-borne measurements can be used to investigate some

of the processes that lead to biases in climate model representations of cloud–aerosol interactions that would otherwise not be amenable to diagnosis from satellite-based measurements alone. When both satellite- and surface-based measurements are used in conjunction with model studies, the synoptically varying vertical structure of Southern Ocean boundary layer and clouds, as well as variability of sources and sinks of CCN, aerosols, and the role of local biogenic sources can be investigated.

# 6   Code availability

The *mpl2nc* source code to convert raw MiniMPL data files to netCDF files is available at https://github.com/peterkuma/mpl2nc. The ALCF open source command line tool for processing of automatic lidar and ceilometer (ALC) data is availble at https://alcf-lidar.github.io/. The tool to convert micro rain radar data into netCDF format is available from https://github.com/peterkuma/mrr2c. The COARE gas exchange algorithm to calculate the transfer velocity for sea-air flux estimates can be obtained from the NOAA ftp server (ftp1.esrl.noaa.gov//BLO/Air-Sea/bulkalg/cor3_6). A Matlab script that can be used to run the COARE code is available in the ReadMe file that is provided with all data from the DAS. Open-source software to convert native radiosonde data into netCDF format is available at https://github.com/peterkuma/rstool.

# 7   Data availability

The TAN1802 voyage measurements described in this study are publicly available in netCDF format from Zenodo at https://doi.org/10.5281/zenodo.4060237 (Kremser et al., 2020). These are packaged in a set of product ZIP archives by instrument and processing level (see also Table A1 for an overview on what is available). The AERONET Maritime Aerosol Network (MAN) hand-held sun photometer data are available directly from the MAN website for the TAN1802 voyage.

**Appendix A:  Data products overview**

**Table A1.** Overview of data products available from the Zenodo TAN1802 data archive for different processing levels, i.e. level 0: Raw (unformatted) data, level 1: Raw data formatted into netCDF format and quality controlled as described in the main text of the paper, and level 2: Derived parameters such as sea–air fluxes.

| Name | Instrument | Processing level | Format |
|---|---|---|---|
| radiosondes | InterMet iMet-1-ABxn and Windsond Radiosonde | 0 and 1 | native/netCDF |
| picarro_crds | Picarro Cavity Ring-Down Spectrometer | 1 | netCDF |
| ceilometer | Lufft CHM 15k ceilometer | 0 | netCDF |
| ceilometer_alcf[a] | Lufft CHM 15k ceilometer | 1 | netCDF |
| mimimpl[b] | Sigma Space MiniMPL | 0 | binary |
| minimpl_alcf[a] | Sigma Space MiniMPL | 1 | netCDF |
| minimpl_mpl2nc[c] | Sigma Space MiniMPL | 1 | netCDF |
| mrr2 | Metek Micro Rain Radar | 0 and 1 | text files/netCDF |
| allskypi | Raspberry Pi sky camera | 1 | JPEG/netCDF |
| bcc200 | Brinno BCC200 sky camera | 0 | AVI |
| pcasp | Optical Particle Counter | 1 | netCDF |
| ccn100 | Cloud Condensation Nuclei Counter | 1 | netCDF |
| cpc3010 | Condensation Particle Counter | 1 | netCDF |
| smps3936 | Scanning Mobility Particle Size Spectrometer | 1 | netCDF |
| nais | Neutral cluster and Air Ion Spectrometer | 1 | netCDF |
| alphasense_opc | UAV and helikite | 1 | CSV/netCDF |
| inp_filter_sampler | Filter Sampler for INPs | 1 | netCDF |
| gc_scd | Gas Chromatograph - Sulfur chemiluminescent detector | 1 and 2 | netCDF |
| mica | Mid-Infrared CAvity enhanced spectrometer | 1 | netCDF |
| ocs_flux | Mid-Infrared CAvity enhanced spectrometer | 2 | netCDF |
| das | Data Acquisition System | 1 | netCDF |
| weather_obs | Human observations | 0 | csv |
| trajectories[d] | Fraction of back trajectories spent over certain regions | 2 | netCDF |

[a] Output obtained from ALCF tool, [b] contains calibration files (minimpl_calibration), [c] output from *mpl2nc* tool, and [d] more information in Hartery et al. (2020b).

**Appendix B:  Radiosonde releases, helikite and UAV flights**

**Table B1.** Release date, time and location of all radiosondes releases as well as helikite and UAV flights.

| Instrument | Time (ISO 8601 UTC) | Latitude | Longitude | Station number | Launch number |
|---|---|---|---|---|---|
| iMet | 2018-02-13 01:46 | 58°02.02'S | 174°13.10'E | 009 | #1 |
| iMet | 2018-02-14 00:47 | 61°00.95'S | 173°29.44'E | 021 | #2 |
| iMet | 2018-02-15 00:20 | 64°40.58'S | 170°57.39'E | 030 | #3 |
| iMet | 2018-02-15 07:33 | 65°29.39'S | 171°47.62'E | 036 | #4 |
| iMet | 2018-02-16 00:10 | 67°50.61'S | 172°50.92'E | 041 | #5 |
| iMet | 2018-02-16 07:40 | 68°38.97'S | 172°47.87'E | 044 | #6 |
| Windsond | 2018-02-16 18:47 | 70°24.06'S | 172°35.52'E | 046 | N/A |
| Windsond | 2018-02-16 21:16 | 70°46.86'S | 172°33.24'E | 047 | N/A |
| iMet | 2018-02-17 00:11 | 70°58.15'S | 172°32.02'E | 048 | #7 |
| Windsond | 2018-02-17 00:52 | 71°13.38'S | 172°30.36'E | 049 | N/A |
| Windsond | 2018-02-17 04:38 | 71°25.86'S | 172°32.52'E | 051 | N/A |
| Windsond | 2018-02-18 06:00 | 71°26.82'S | 171°59.22'E | 056 | N/A |
| Windsond | 2018-02-18 21:27 | 71°30.60'S | 171°45.18'E | 063 | N/A |
| iMet | 2018-02-19 00:31 | 71°30.87'S | 171°40.37'E | 066 | #8 |
| Windsond | 2018-02-19 03:04 | 71°38.64'S | 171°42.66'E | 068 | N/A |
| iMet | 2018-02-20 07:31 | 71°56.21'S | 171°55.89'E | 078 | #10 |
| Windsond | 2018-02-21 00:00 | 71°50.34'S | 174°23.04'E | 085 | N/A |
| iMet | 2018-02-22 03:39 | 72°41.05'S | 178°12.36'W | 086 | #11 |
| iMet | 2018-02-22 07:50 | 72°59.14'S | 177°15.18'W | 088 | #12 |
| iMet | 2018-02-23 02:18 | 73°00.47'S | 177°07.17'W | 093 | #13 |
| iMet | 2018-02-23 07:31 | 72°50.67'S | 176°43.39'W | 095 | #14 |
| iMet | 2018-02-24 00:01 | 72°19.68'S | 178°52.17'W | 100 | #15 |
| iMet | 2018-02-24 07:26 | 72°26.35'S | 179°09.89'W | 103 | #16 |
| iMet | 2018-02-25 00:02 | 72°11.56'S | 178°29.18'W | 111 | #17 |
| iMet | 2018-02-25 06:17 | 72°05.03'S | 178°03.53'W | 112 | #18 |
| Windsond | 2018-02-25 22:02 | 71°16.32'S | 177°59.52'W | 115 | N/A |
| Helikite | 2018-02-26 02:34 | 71°13.81'S | 178°03.58'W | 117 | N/A |
| iMet | 2018-02-27 00:02 | 71°00.34'S | 179°37.80'E | 120 | #19 |
| iMet | 2018-02-28 00:10 | 70°56.05'S | 179°57.72'E | 131 | #21 |
| iMet | 2018-02-28 07:53 | 70°53.14'S | 178°59.57'W | 135 | #22 |
| iMet | 2018-03-01 19:34 | 69°31.04'S | 178°17.11'W | 140 | #24 |
| iMet | 2018-03-02 00:01 | 69°29.84'S | 177°56.63'W | 143 | #25 |
| iMet | 2018-03-02 07:59 | 69°29.14'S | 179°33.22'W | 145 | #26 |

Legend: Not available (**N/A**), iMet-1-ABxn radiosonde (**iMet**), Windsond radiosonde (**Windsond**).

**Table B2.** Release date, time and location of all radiosondes releases as well as helikite and UAV flights. (cont.).

| Instrument | Time (ISO 8601 UTC) | Latitude | Longitude | Station number | Launch number |
|---|---|---|---|---|---|
| iMet | 2018-03-02 19:35 | 69°08.76'S | 177°48.10'W | 146 | #27 |
| iMet | 2018-03-03 00:09 | 69°04.66'S | 178°06.52'W | 150 | #28 |
| UAV | 2018-03-03 00:34 | 69°02.11'S | 178°01.92'W | 151 | N/A |
| iMet | 2018-03-03 07:32 | 69°01.09'S | 178°58.81'W | 153 | #29 |
| iMet | 2018-03-03 19:48 | 68°56.47'S | 178°56.01'W | 154 | #30 |
| iMet | 2018-03-04 00:34 | 68°51.46'S | 178°44.80'W | 158 | #31 |
| iMet | 2018-03-04 19:47 | 68°20.03'S | 179°58.32'W | 163 | #32 |
| Windsond | 2018-03-05 07:31 | 68°10.74'S | 179°58.08'W | 171 | N/A |
| iMet | 2018-03-06 20:08 | 66°44.56'S | 177°09.52'W | 173 | #34 |
| Windsond | 2018-03-06 22:21 | 66°36.78'S | 177°27.30'W | 176 | N/A |
| iMet | 2018-03-07 00:04 | 66°38.73'S | 177°22.30'W | 178 | #35 |
| Windsond | 2018-03-07 04:44 | 66°43.56'S | 177°10.44'W | 181 | N/A |
| iMet | 2018-03-07 07:40 | 66°44.51'S | 177°04.00'W | 183 | #36 |
| iMet | 2018-03-07 19:49 | 66°45.39'S | 177°01.48'W | 187 | #37 |
| iMet | 2018-03-08 00:09 | 66°57.30'S | 176°13.38'W | 190 | #38 |
| iMet | 2018-03-08 07:37 | 66°55.15'S | 176°16.24'W | 194 | #39 |
| iMet | 2018-03-09 00:13 | 67°07.69'S | 175°40.01'W | 205 | #40 |
| iMet | 2018-03-09 07:36 | 67°02.58'S | 175°36.87'W | 206 | #41 |
| iMet | 2018-03-09 19:50 | 67°10.77'S | 175°25.57'W | 210 | #42 |
| iMet | 2018-03-10 00:18 | 67°08.99'S | 175°30.05'W | 214 | #43 |
| iMet | 2018-03-10 07:33 | 66°52.08'S | 176°28.49'W | 215 | #44 |
| iMet | 2018-03-10 19:44 | 66°22.13'S | 177°45.99'W | 217 | #45 |
| iMet | 2018-03-11 00:41 | 66°17.64'S | 178°31.03'W | 221 | #46 |
| iMet | 2018-03-11 07:32 | 66°16.37'S | 179°21.91'W | 223 | #47 |
| iMet | 2018-03-11 19:45 | 65°27.60'S | 179°25.27'E | 227 | #48 |
| iMet | 2018-03-12 00:12 | 65°11.40'S | 179°06.63'E | 231 | #49 |
| iMet | 2018-03-12 07:38 | 64°36.50'S | 178°13.59'E | 234 | #50 |
| iMet | 2018-03-12 20:11 | 63°40.00'S | 176°06.68'E | 236 | #51 |
| iMet | 2018-03-13 00:35 | 63°31.75'S | 176°05.52'E | 240 | #52 |
| iMet | 2018-03-13 07:50 | 62°59.00'S | 176°07.15'E | 242 | #54 |
| iMet | 2018-03-14 00:12 | 62°03.87'S | 174°58.88'E | 243 | #55 |
| iMet | 2018-03-14 07:47 | 62°17.83'S | 175°08.36'E | 244 | #56 |
| iMet | 2018-03-14 20:02 | 63°10.46'S | 174°27.50'E | 246 | #57 |
| iMet | 2018-03-15 00:23 | 62°56.37'S | 174°18.31'E | 251 | #58 |
| iMet | 2018-03-15 07:42 | 61°50.72'S | 173°46.18'E | 252 | #59 |

Legend: Not available (**N/A**), iMet-1-ABxn radiosonde (**iMet**), Windsond radiosonde (**Windsond**).

*Author contributions.* All co-authors contributed data from one or more instruments and provided relevant figures and material for the manuscript. PK participated in the organisation of the voyage and deployment of instruments, performed observations during the voyage, post-processed a part of the data set, and developed the mpl2nc, mrr2c, rstool and ALCF software packages. SH maintained and ran the aerosol instruments during the voyage and prepared all aerosol data sets (except from the NAIS instrument). MP prepared the NAIS data, and together with KS prepared the required material for the manuscript. KS shipped the NAIS from France and installed the NAIS on the *Tangaroa* prior to the voyage and performed remote quality checks of the data during voyage. JM participated in the voyage, and prepared and quality controlled the $CO_2$ measurements. AM was responsible for collecting DMS and OCS measurements during the voyage. AS-M was in charge of the QA/QC of the dissolved DMS measurements, and together with MH and CSL provided the figures and material for the paper. MH led the collaborative proposal for the aerosol–cloud component of the voyage and calculated the DMS fluxes. RQ provided the MiniMPL instrument for the voyage and processed the data, AG developed the allskypi system and software and prepared the allskypi data. STL and MvH provided, prepared and quality checked the MICA instrument assembly and prepared MICA related data. AMcD prepared the rain radar data. IS and CF took part in the processing and calibration of the MiniMPL data. TH, PDeM, and CH analysed and provided the INP data. GG designed and build the particle sensing AlphaSense radiosonde equipment used for measurements with the UAV. SP prepared and installed the meteorological equipment, such as ceilometer, micro rain radar, Brinno sky cameras and provided logistical support. SK wrote the manuscript with contributions from all co-authors.

*Competing interests.* The authors declare that they have no conflict of interest.

*Acknowledgements.* We thank the vessel master Evan Solly, all officers, crew and the voyage leader Dr David Bowden of the TAN1802 voyage. We would also like to thank Gordon Brailsford for providing the Picarro instrument and Josh McCulloch for designing the Alphasense OPC unit with an RTC clock for the use with the UAV. We thank Tony Bromley from NIWA for supplying and preparing the helikite and Windsond systems used during the voyage and International Met Systems (InterMet) for providing the radiosonde base station iMet-3050A. We acknowledge the financial support provided by the Deep South National Science Challenge via the Clouds and Aerosols project. The voyage was supported through a New Zealand Crown Funding Agreement and associated voyage science was funded through the NIWA Research Programme in Ocean-Climate Interactions (2017/19 SCI). PK and SH both acknowledge independently awarded scholarships provided by the University of Canterbury and the Deep South National Science Challenge. PK's contribution was partially self-funded (Peter Kuma Software & Science, peterkuma.net). This project has also received support from ANR T-ERC Sea2Cloud and from the European Research Council (ERC) under the European Union's Horizon 2020 research and innovation programme (grant agreement No 771369). TH, PDeM and CH acknowledge partial support from U.S. National Science Foundation Award 1660486.

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
