# Peer review of "Southern Ocean Cloud and Aerosol data: a compilation of measurements from the 2018 Southern Ocean Ross Sea Marine Ecosystems and Environment voyage"

_Earth System Science Data, 2020_

## Referee Comment (RC1) · Gerald Mace (Referee) · 19 Feb 2021

I have reviewed the manuscript describing the data set collected during the TAN1802 voyage. The data collected on this voyage to the Ross Sea in January and February 2018 and the manuscript describing the data are quite interesting and very well done. Data sets like this to remote areas are rare and extraordinarily valuable because they document unique regions and processes. This data set highlights interactions between the marine biogeochemistry and the atmosphere that are largely unobserved. It is to the authors' credit the they are willing to make this data available to the community.

[Figure]

Community access to this data will certainly maximize the value of the data.

Overall, the manuscript is well done and nicely describes the data set. I think that a bit more detail could be included for not a lot of work that could highlight how the data set fits into a larger context. I highlight those points below in my comments. Overall, I suggest no major revisions, and I think the manuscript should be published after minor revisions.

Specific Comments:

Abstract Line 5: Satellite measurements certainly see the low level clouds without much trouble. The satellite measurements just can't constrain the aerosol-cloud processes very well. Maybe a bit of clarification on this point.

Line 55. No argument from me about the need to know INP better, but doesn't the low concentrations of INP suggest that it is very important to understand secondary ice processes much better than we do in this region of low INP?

line 76: Do you mean to say that "process model predict that the highest open ocean OCS fluxes are in the Southern Ocean"? Your sentence is missing "that" after "predict"

Line 93 (first bullet): "low/medium" is a rather ambiguous term. Do you mean clouds based in the boundary layer or clouds that are not cirrus or...?

Table 2 and Science Objectives: It might be useful to readers to number your science objective listed in 2.1 and then in Table 2 include a column that connects science objectives with measurements - something like a science traceability matrix.

Line 104: The heading text ("Meteorological conditions") is not descriptive of what is in the subsection.

Line 112: Is 1-minute the highest rate at which motion and position are recorded?

Line 123: Is there a reference describing the gas exchange algorithm?

Line 154: Line 154: You don't say how often the data are recorded from the radioson-des. The temporal or vertical resolution would be useful.

Line 184: Oh No! Major Bummer!

Line 220: Do you mean the maximum detection range? Statement makes it sound like it does not record below the upper troposphere.

Line 232: How often did the instrument/software fail?

Line 252: would be good to note the temporal and vertical resolutions of these files. Do the files include the attenuated backscatter of the lidars?

Section 3.4: Was the MRR on a stabilized platform? If not the Doppler information will be difficult to interpret due to ship motions. Also, is the MRR calibrated against some independent radar measurement? These are both critical details.

Line 277: This is a fairly sophisticated camera. Any chance the RGB imagery could be calibrated?

Line 370: This seem like an important step. Should it be described in more detail? What does "corrected to ambient" mean?

Line 449: so the the archived data are temperature dependent number concentrations of INP one per filter? Perhaps clarify? Was additional chemistry done on these filters? Could it be if not?

Figure 4: It would be useful to plot these over an ocean color or chlorophyl-a retrieval map. I eyeballed the map that can be created at https://oceancolor.gsfc.nasa.gov/l3/ and the monthly plot for February 2018 shows some interesting structure in your region that seems to correspond to your data.

Line 591: If the MRR is not stabilized the Doppler information is likely not suitable for the Maahn and Kolias algorithm. To my eye, the Doppler velocity looks to include ship motions but it is very hard to discern at the plotting resolution.

Line 618: What fraction of the data are deemed contaminated from exhaust? A day by day graphic showing the contaminated fraction would be useful.

Line 647: Increases in total number (e) and ccn (d) are pretty obvious on Feb 17. However increases in number in panels a and b are not obvious. Does this air mass change correlate with the onset of strong southerlies in Figure 5? May be worth noting if so. Also, on NASA Worldview I think maybe I see your ship track in the MODIS Aqua imager approaching Cape Adare. https://worldview.earthdata.nasa.gov/?v=258225.19165830087,-2145493.8406539974,501157.02538037306,-1958367.8679662866&p=antarctic&t=2018-02-17-T15%3A23%3A20Z

Figure 14: Having titles or insets on these panels listing what each is would be very helpful.

With Compliments, Jay Mace

––––––––––––––––––––––––––––––––

---

## Referee Comment (RC2) · Nadine Borduas-Dedekind (Referee) · 15 Mar 2021

Reviewers:

The reviewers of this manuscript are a research group in atmospheric chemistry at the University of British Columbia led by Dr. Nadine Borduas-Dedekind. The group met twice for a combined time of 3 h to review and discuss the manuscript, the instrumentation, the data and the data visualization. The discussion was led by undergraduate student Max Aragon Cerecedes and this report was compiled and edited by Dr.

Borduas-Dedekind. Additional graduate and undergraduate student co-authors of this review include (in alphabetic order): Ayomide Akande, Sophie Bogler, Isabelle Lao, Rickey Lee, Madri Jayakody and Jon Went.

General overview:

First and foremost, we congratulate the authors for the extraordinary team effort in collecting valuable cloud, aerosol and seawater data in the Southern Ocean. This paper presents an open access data set shared on Zenodo, and describes the instrumentation deployed and the data collected during the voyage of R/V Tangoroa in February-March 2018 from New Zealand to the Ross Sea, off the coast of Antarctica. As it stands, the data is presented at different processing levels depending on the instrumentation and will be undeniably useful for evaluating aerosol-cloud interactions in weather and climate models, including the biogeochemical cycling of sulfur compounds.

To help the authors have their data used more quickly, easily and efficiently, we have compiled a list of recommendations, clarifications and critical comments.

Our critical feedback on the structure of the paper includes working on the flow of the manuscript. Leading a large collaborative effort such as this manuscript has its challenges, including producing a unifying storyline of all the measurements undertaken during the voyage. In general, presenting the data by instrument is a useful and efficient way to categorise and present the data, and we command the authors for this structure. Nonetheless, even if multiple authors wrote different sections, we encourage the authors to ensure each section addresses all the information listed below, and we can suggest this type of format: (for example, we thought section 3.3. was particularly well written and structured)

1. Describe why this parameter was measured and what it will (or could) be used for in the future.

2. Describe the instrument operation in detail (ESSD serves as supplementary information, so every detail for operation should be included).

3. Comment on why the instrument was chosen over other alternatives or models.

4. Describe how the data was collected and, when applicable, processed

5. Show the data in figures/tables.

6. End each section by relating which goal is being addressed (from the list in pages 5-6, which can be numbered in order to refer to easily)

7. Avoid data interpretation, as required by the ESSD format. Note that some sections have added data interpretation (ex: 4.2.4) which should be removed.

To further help with the readability of the manuscript, a table of contents would be really useful as a reference for the future reader.

Furthermore, the authors describe instruments which were taken onboard, but which didn't collect data due to malfunctions or errors. The goal in relating this instrument information can be to further instruct scientists of lessons learned, and we think these instrument descriptions are worthwhile additions to the manuscript. However, we suggest that comments faulting inexperienced controllers should be omitted (in the spirit of sportsmanship) (for example lines 196-197). We also suggest to the authors to group "unused" instrumentation to a separate section called "lessons learned" or more objectively, "instrument malfunctions". Nevertheless, we appreciated the transparency offered by the authors of the instrumentation malfunctions. We wondered whether the authors may have one or two suggestions to add to these sections in order for future readers to be better prepared for their own voyage (Figure 18 - is it useful if the data will not be used? Perhaps not worth plotting?)

Specific manuscript comments:

Line 4: Could the authors comment on the direction of the "persistent biases"? An

added qualification such as bias low or high would be more precise.

Line 8: According to the Earth System Science Data manuscript preparation and file submission: Ship names are italic, but their prefixes are roman (e.g. RV Polarstern).

Line 52. Reference format is different from the rest of the paper and should be double checked.

Line 77-78 Which cruise was the first to probe OCS concentrations and sources? Could the authors add the reference? Or indicate it in the table 1?

Table 1: Very useful for context and for future readers and data users! The authors could also add their own ship campaign at the bottom of the table with a "this work" reference.

Line 86. Could the authors briefly explain their motivation for spending most of the voyage's time (30 days) south of 60°S? Does 60°S represent a reference point for some measurements for example?

Lines 92-93. The authors mention the characterisation of radiation but list only the lidar, ceilometer and sky cameras. How was this parameter measured? Wasn't there a radiometer or pyranometer on board which should be added to this list?

Lines 92-101. The authors mention seven research objectives (line 88) but only list six (92-101). Is there a goal missing? In addition, was there a priority within these goals?

Lines 92-101: Give each goal a number to refer to these goals in each instrument section throughout the manuscript.

Table 2: Can the authors ensure that all the items in the column "location on the ship" in Table 2 are also written in Figure 3? For example, where is the fantail located?

Line 117. How did the authors correct the wind speed according to the ship heading and speed? Also with (Popinet et al., 2004)? Please clarify and/or refer to the appropriate section.

Line 166. Wind speed accuracy is stated as 5%. Can the authors comment on whether this accuracy is typical for this instrument? What was the wind direction accuracy?

Line 180-185. The authors' UAV's battery was drastically reduced due to low atmospheric temperatures. Which UAV model was used, which battery?

Section 3.3 – Particularly well written and structured section (see our comment on general flow and structure at the beginning of this review).

Line 221. "The maximum range is 30km but the effective range was lower than that". Can the authors specific the effective range?

Line 273: Can the authors link the use of the sky camera to one of their objectives?

Line 280. Is the technique HDR? Mertens 2009 explicitly proposed exposure fusion as an alternative to HDR to produce a high‐quality, low dynamic range image. Did they authors use the correct reference here?

Line 282. The ELIFAN algorithm crops the sky pictures to remove the distortion before estimating cloud fraction. Did the authors remove the distortion of the allskypi pictures? What was the field of view of the fisheye lens? Could the authors comment on the cloud cover uncertainty (for example in Figure 7)?

Line 291: Can the authors specify how rare clear-sky conditions were on their voyage? If the authors' goals were to study aerosol-cloud interactions, could the authors briefly comment on the value of the sun photometer measurement?

Line 459: In section 3.10.1 title, the word chromatograth is misspelled.

Line 552. How was the % of cloud types calculated? Was this calculation performed by human observations, sky camera pictures and the ceilometer? Did the authors use an automatic algorithm to derive cloud types? Additional information would be useful to understand the data presented.

Section 4.2.4 (lines 673-693) has too much interpretation (not in the realm of ESSD's

scope) and can be rewritten to specify the sample collection details, the operating procedure of the instrument, why this instrument was chosen and the data collected. The data collected should include the frozen fractions as well as the INP concentrations.

Comments on Figures:

During the discussion of this manuscript as a group, we gathered images of each instrument and found this process to be very helpful in visualizing the instruments. We can recommend to the authors to do the same, by adding pictures for each instrument to each section. (The authors can also contact us for these pictures (borduas@chem.ubc.ca and aragon@gamma.ttk.pte.hu , as we've gathered them already for our discussion.)

Figure 2: The figure design can be improved for clarity by identifying the level of data analysis (raw vs calculated). The boxes can also be aligned for a cleaner figure.

Figure 3: Do the authors want to further describe the aerosol container lab? Or alternatively show a picture of the inside? The ship's exhaust should be highlighted as it is a big part of the discussion and of the data interpretation. The authors could also add a real picture of the ship (we had to google for a picture) for improved visualization.

Figure 5: Along the left panel, the mean values can be added for clarity and readability. The numbering can be rethought, for example labels (b, d, f, h, i, l) could be removed.

Figure 6: This data is useful to highlight how the tropopause is shallower closer to the pole. With that point in mind, could the authors arrange the panels as a function of latitude instead of as a function of time of the voyage?

Figure 8: Can the authors add the number of points included in each boxplot? Can the authors also specify the values of the whiskers? We weren't sure where the 0.86 coefficient for the atmospheric transmission coefficient came from; could the authors add a description and a reference? Finally, how do the authors explain values above one?

Figure 9: We struggled to understand this figure and perhaps it can be made clearer. What are the bins representing? Could they be better depicted as a histogram/bar graph? Could the percentage and cumulative occurrences be displayed on two graphs? The x-axis at the top and bottom of the plot for percentage is different, maybe color coding the axis labels to the plot line could help. One of the plot lines doesn't show up in the legend.

Figure 11. It would be worth adding a title to each plot to clarify the graph. The letter a of the plot a) has a smaller size than the other ones and Y-axis titles aren't aligned well. The colour bars should avoid white, otherwise the information cannot be seen (particularly true for the vertical velocity plot (b)).

Figure 12: Small note that there is a blue dot on at x=0 value. Could the authors double check? Can the authors comment on how realistic a value of 520ppm of $CO_2$ from an exhaust is?

Figure 13: Could the authors remove the graph lines to help clarity?

Figure 14. In the figure caption, the figures should be labelled a, b, c, d and e. We can also recommend to the authors to add the name of each instrument along each panel for better readability. We recommend plotting the CCN data on a separate graph.

Figure 15: We appreciated this figure to visualise the merger of the datasets on particle diameters and numbers. Thank you!

Figure 17: Nice!

Figure 21: It would be useful to have titles on the plots themselves.

Figure 22: This figure contains a lot of information and additional panels would help the clarity of this data.

Dataset and code availability comments

We recommend to the authors to add photographs of each one of their instruments for

improved visualization of the equipment used in this voyage.

Weather_obs_level_0 → What are the codes of weather types (1-4)?

The automatic weather station data appear to be complete and all information is available.

Line 299. Why is the sun photometer data only found in the Maritime Aerosol Network? Is it possible to add it to the authors' Zenodo data set too?

Line 780. ALCF tool was downloaded, checked and confirmed after communication with one of the authors, Peter Kuma. The script now worked well. Thank you for sharing this resource!

Line 783. The authors provide the website for COARE gas exchange algorithm but in Table A1 in the "das" Data Acquisition System ReadMe_file the authors also provide the Matlab script to calculate fluxes. It might be worth mentioning this script in the Code Availability.

We would also encourage the authors to explore the possibility of providing their data as an open API through https://developers.zenodo.org/

We end this review by once again commanding the authors and scientists for their hard work and effort in gathering this dataset. We wish the authors all the best with their future data analyses and with addressing their scientific research goals.

---

## Author Comment (AC1) · 13 May 2021

**1   Response to Gerald Mace**

I have reviewed the manuscript describing the data set collected during the TAN1802voyage. The data collected on this voyage to the Ross Sea in January and February 2018 and the manuscript describing the data are quite interesting and very well done. Data sets like this to remote areas are rare and extraordinarily valuable

because they document unique regions and processes. This data set highlights interactions between the marine biogeochemistry and the atmosphere that are largely unobserved. It is to the authors' credit the they are willing to make this data available to the community.

Community access to this data will certainly maximize the value of the data.

Overall, the manuscript is well done and nicely describes the data set. I think that a bit more detail could be included for not a lot of work that could highlight how the dataset fits into a larger context. I highlight those points below in my comments. Overall, I suggest no major revisions, and I think the manuscript should be published after minor revisions.

We would like to thank Gerald Mace for taking the time to review this paper and for his helpful comments that have led to improvements of the paper. His comments are repeated below in blue with our reply in black.

**Specific Comments**:

Abstract Line 5: Satellite measurements certainly see the low level clouds without much trouble. The satellite measurements just can't constrain the aerosol-cloud processes very well. Maybe a bit of clarification on this point.

We have rewritten the sentence for clarification following the reviewer's suggestion and would like to further comment that while passive satellite instruments can observe the cloud top, and active satellite instruments whole clouds (if they are not too opaque) active instruments (such as CloudSat, CALIPSO) cannot observe low clouds well. For CloudSat this is because of ground clutter in the first 1 to 2 km from the ground, and for CALIPSO this is because the laser signal cannot generally pass through anything but thin clouds. Passive satellite instruments and CALIPSO cannot observe low clouds if they are obscured by mid- or high-level clouds. Ground-based remote sensing observations can provide cloud base height and make it possible to identify more accurately

cloud types such as stratocumulus and stratus (which are common in the Southern Ocean region) and their horizontal and vertical extent. For a more detailed discussion we refer to McErlich et al. (2021).

Reference

McErlich, C., McDonald, A., Schuddeboom, A., Silber, I. (2021). Comparing satellite‐ and ground‐based observations of cloud occurrence over high southern latitude. Journal of Geophysical Research, 126, e2020JD033607. https://doi.org/10.1029/2020JD033607.

Line 55. No argument from me about the need to know INP better, but doesn't the low concentrations of INP suggest that it is very important to understand secondary ice processes much better than we do in this region of low INP?

It is a good point raised by the reviewer, and we added two sentences in the revised manuscript to highlight the need for a better understanding and representation of secondary ice formation processes.

line 76: Do you mean to say that "process model predict that the highest open ocean OCS fluxes are in the Southern Ocean"? Your sentence is missing "that" after "predict".

We reworded the sentence for clarification.

Line 93 (first bullet): "low/medium" is a rather ambiguous term. Do you mean clouds based in the boundary layer or clouds that are not cirrus or...?

The reviewer is correct that this objective is unclear, and we have changed the wording. The voyage objective was to look at low and middle level clouds, using the WMO standard height definitions for polar clouds, i.e. low level clouds below 2 km and middle level clouds between 2 and 4 km. We have revised the research aim statement in the revised manuscript.

Table 2 and Science Objectives: It might be useful to readers to number your science

objective listed in 2.1 and then in Table 2 include a column that connects science objectives with measurements - something like a science traceability matrix.

That is a good suggestion by the reviewer, and we have followed his advice and now provide an additional column in Table 2 that links the instrument to the research objective listed in section 2.1.

Line 104: The heading text ("Meteorological conditions") is not descriptive of what is in the subsection.

We changed the heading from "Meteorological conditions throughout the voyage" to "Meteorological measurements and metadata".

Line 112: Is 1-minute the highest rate at which motion and position are recorded?

No, sensors record at different time resolutions and raw data are also available at higher frequencies from some sensors, e.g. data on ships attitude (pitch, roll, heave) and heading from a POS/MV Model 320 are recorded at 2 Hz. However, the DAS (The Tangaroa Data Acquisition System) stores the data at one-minute intervals of common variables, which is referred to in the manuscript and which is the temporal resolution of the data set provided. For examining general spatial relationships in the data, 1- or 5-minute resolution is a useful common period available for multiple sensors.

Line 123: Is there a reference describing the gas exchange algorithm?

The reference describing the gas exchange algorithm is mentioned in section 4.3.1 - we have now included the appropriate references upfront as well.

Line 154: Line 154: You don't say how often the data are recorded from the radiosondes. The temporal or vertical resolution would be useful.

We have followed the suggested by the reviewer and included additional information in the revised manuscript.

Line 184: Oh No! Major Bummer!

Indeed, the loss of the UAV was disappointing but with the measurements from the first flight, we can show the value of UAV measurements in the Southern Ocean and we have learned a lot about the operation of an UAV in the Southern Ocean. The UAV setup was experimental, and the aim was to explore the potential of using an UAV in the SO. For future operations of an UAV in the Southern Ocean region we would recommend that the UAV is first tested in low temperature conditions and admittedly training the person who is going to operate the UAV is important too.

Line 220: Do you mean the maximum detection range? Statement makes it sound like it does not record below the upper troposphere.

Yes, the reviewer is correct that the wording of this sentence is misleading. We re-worded the sentence for clarification.

Line 232: How often did the instrument/software fail?

During the six-week voyage, MPL measurements are not available or are unreasonable on 9 days, due to the software failure mentioned in the manuscript. We edited the sentence in the revised manuscript and also added that information to the data set description on Zenodo.

Line 252: would be good to note the temporal and vertical resolutions of these files. Do the files include the attenuated backscatter of the lidars?

We added additional information about the resolution to the revised manuscript. In response to the reviewer comment, the MPL and ceilometer data files include the normalised relative backscatter (NRB) signal, which is the attenuated backscatter after multiple corrections (as explained in l. 23-237 in the original manuscript). The ceilometer backscatter variable is the attenuated backscatter as well.

Section 3.4: Was the MRR on a stabilized platform?

No, the MRR was not on a stabilized platform.
If not the Doppler information will be difficult to interpret due to ship motions. Also, is the MRR calibrated against some independent radar measurement? These are both critical details.

The Doppler information is integrated over a period, which means that variations in the ship motion will be more likely to impact spectral width (i.e. the variance of the Doppler velocity) rather than the value derived. We agree that this will add additional uncertainty to the derived precipitation data, but not necessarily more than other underlying assumptions. Note MRR are generally not calibrated against other instruments apart from disdrometers. However, we did not have one of these instruments onboard and therefore the MRR was not calibrated. This also potentially adds uncertainty into the mix. An extra paragraph was added to the revised manuscript.

Line 277: This is a fairly sophisticated camera. Any chance the RGB imagery could be calibrated?

Unfortunately, this is not possible. The primary use was to use ratios between the color values, with thresholds tuned to the camera itself, so calibration was unnecessary. A color profile could be created and applied, but then the thresholding algorithm would need to be retuned for them to relate to the cloud fraction output.

Line 370: This seem like an important step. Should it be described in more detail? What does "corrected to ambient" mean?

We apologize for the vague language. Correcting to "ambient" means that we corrected the concentrations to standard temperature and pressure. This has been clarified in the revised manuscript. Furthermore, some additional text describing the size-dependent losses has been added.

Line 449: so the the archived data are temperature dependent number concentrations of INP one per filter? Perhaps clarify? Was additional chemistry done on these filters? Could it be if not?

Yes, that's correct. To be clearer, we have amended the following sentence from "Temperature spectra of INP concentrations active via immersion freezing were obtained..." to "Temperature dependent number concentrations of INPs active via immersion freezing (one spectrum per filter) were obtained. . .". No additional chemistry was performed on any filters, but is planned for the subset that coincide with the passage of the ship east of Macquarie Island in mid Feb, 2018 during the MICRE campaign. INPs were being collected on the island at the time, using the same filter system, and samples were later treated with heat and hydrogen peroxide to assess "biological", total organic and mineral INP contributions. It will be valuable to compare not only base INP concentrations but also relative abundances of these three components on the island and 70 east of it during the same period.

Figure 4: It would be useful to plot these over an ocean color or chlorophyll-a retrieval map. I eyeballed the map that can be created at https://oceancolor.gsfc.nasa.gov/l3/and the monthly plot for February 2018 shows some interesting structure in your region that seems to correspond to your data.

We thank the reviewer for this suggestion. However, we would prefer not to include any information about ocean color or chlorophyll-a as going into the biogeochem or biological links to any extent is beyond the scope of this study and will be likely part of future publications. This paper focuses on the description of the data sets obtained during the cruise and by providing information about the times and position coordinates of the ship, future users can easily explore correlations of these data to remotely sensed images of SST, ocean color or chlorophyll-a. As a result, we decided not to include any information about ocean color or chlorophyll-a as we expect some interesting research papers to follow this study.

Line 591: If the MRR is not stabilized the Doppler information is likely not suitable for the Maahn and Kolias algorithm. To my eye, the Doppler velocity looks to include ship motions but it is very hard to discern at the plotting resolution.

The Doppler information is integrated over a period, which means that variations in the ship motion will be more likely to impact spectral width (i.e. the variance of the Doppler velocity) rather than the value derived. We agree that this will add additional uncertainty to the derived precipitation data, but not necessarily more than other underlying assumptions. Note MRR are generally not calibrated against other instruments apart from disdrometers. However, we did not have one of these instruments onboard and therefore the MRR was not calibrated. This also potentially adds uncertainty into the mix. An extra paragraph was added to the revised manuscript for clarification.

Line 618: What fraction of the data are deemed contaminated from exhaust? A day by day graphic showing the contaminated fraction would be useful.

For the measured particle concentration, less than 20% of the measurements were removed due to contamination each day (a new figure has been added to the revised manuscript). Only on three days during the measurement period, more than 20

Line 647:Increases in total number (e) and ccn (d) are pretty obvious on Feb 17. However increases in number in panels a and b are not obvious. Does this air mass change correlate with the onset of strong southerlies in Figure 5? May be worth noting if so. Also, on NASA Worldview It hink maybe I see your ship track in the MODIS Aqua imager approaching CapeAdare https://worldview.earthdata.nasa.gov/ ?v=258225.19165830087,-2145493.8406539974,501157.02538037306,-1958367. 8679662866&p=antarctic&t=2018-02-17-T15%3A23%3A20Z

The air mass definitely changed upon our arrival to Cape Adare, but we should add some further context to this section as it is not just the air mass that is changing. In addition to a changing air mass, fog was also consistently present prior to Feb 17. The occurrence of fog can be inferred from the near 100% RH in Figure 7 from Feb 11 - 17. Cloud base height measurements with the ceilometer also confirmed the presence of fog. The presence of fog accentuated the differences in the number of CCN before and after Feb 17, since any CCN would have been activated within the fog leading up to the

17th. The presence/absence of fog is what is predominantly driving differences in the particle size distribution prior to and after Feb 17, with the changing air mass providing an additional influence.

We agree that it is hard to observe changes in the number of particles in the contour plots of Figures 16 a b since the contours are half an order of magnitude and there was some contamination from ship exhaust in the days immediately preceding Feb 17. However, we did see increases in the number of accumulation and sea spray mode particulate after Feb 17. Look specifically along the 100 nm and 1 um lines: there is a subtle but definite increase in the number of these particles after Feb 17. For a more convincing presentation, the difference in the number of accumulation mode particulate and CCN before and after Feb 17 is shown in Figure 1c of a manuscript submitted to JGR: Atmospheres, "Classification of the Below-Cloud Mixing State Over the Southern Ocean Using In-Situ and Remotely-Sensed Measurements" (https://www.essoar.org/doi/abs/10.1002/essoar.10502904.2). While most of the change to the total number of CCN was due to an increased availability of accumulation mode particulate, sea spray particles were also more abundant after Feb 17 due to the heavy southerlies. The correlation between the abundance of sea spray particles and wind history is further described in Hartery et al (2020).

Accompanying text has been added to the manuscript to highlight this.

Figure 14: Having titles or insets on these panels listing what each is would be very helpful.

We included the instrument name in the titles on each figure panel of Figure 14 (now Fig. 16).

---

## Author Comment (AC2) · 13 May 2021

**1   Response to Nadine Borduas-Dedekind and her students**

The reviewers of this manuscript are a research group in atmospheric chemistry at theUniversity of British Columbia led by Dr. Nadine Borduas-Dedekind. The group met twice for a combined time of 3 h to review and discuss the manuscript, the instrumenta-tion, the data and the data visualization. The discussion was led by undergradu-ate student Max Aragon Cerecedes and this report was compiled and edited by Dr.Borduas-Dedekind. Additional graduate and undergraduate student co-authors of this review include (in alphabetic order): Ayomide Akande, Sophie Bogler, Isabelle Lao,Rickey Lee, Madri Jayakody and Jon Went.

We would like to thank Dr. Nadine Borduas-Dedekind and her students for taking the time to review this paper and for their helpful comments and suggested changes that improved the paper. Their comments are repeated below in blue with our reply in black.

General overview: First and foremost, we congratulate the authors for the extraordinary team effort in collecting valuable cloud, aerosol and seawater data in the Southern Ocean. This paper presents an open access data set shared on Zenodo, and describes the instrumentation deployed and the data collected during the voyage of R/V Tangoroa in February-March 2018 from New Zealand to the Ross Sea, off the coast of Antarctica. As it stands, the data is presented at different processing levels depending on the instrumentation and will be undeniably useful for evaluating aerosol-cloud interactions in weather and climate models, including the biogeochemical cycling of sulfur compounds. To help the authors have their data used more quickly, easily and efficiently, we have compiled a list of recommendations, clarifications and critical comments.

Our critical feedback on the structure of the paper includes working on the flow of the manuscript. Leading a large collaborative effort such as this manuscript has its challenges, including producing a unifying storyline of all the measurements undertaken during the voyage. In general, presenting the data by instrument is a useful and efficient way to categorise and present the data, and we command the authors for this structure. Nonetheless, even if multiple authors wrote different sections, we encourage the authors to ensure each section addresses all the information listed below, and we can suggest this type of format: (for example, we thought section 3.3. was particularly well written and structured).

1. Describe why this parameter was measured and what it will (or could) be used for in the future.

2. Describe the instrument operation in detail (ESSD serves as supplementary information, so every detail for operation should be included).

3. Comment on why the instrument was chosen over other alternatives or models.

4. Describe how the data was collected and, when applicable, processed.

5. Show the data in figures/tables.

6. End each section by relating which goal is being addressed (from the list in pages5-6, which can be numbered in order to refer to easily).

7. Avoid data interpretation, as required by the ESSD format. Note that some sections have added data interpretation (ex: 4.2.4) which should be removed.

We thank the reviewers for this suggestion and we worked through the instrument description again and, where appropriate, we added additional material to the sections. We have now also clarified in Table 2 which research aim is addressed by which instrument and created a 'traceability matrix' with Table 2. We believe that Table 2 contains most if not all the information (including parameters measured) that a user of the data might need. The individual sections on each instrument then explain in more detail the measurements and their uncertainties and describe the measurement techniques, if a detailed explanation about the technique is warranted. We have used our judgement to balance supplying specific details needed for interpretation and the overall size of the paper. We ensured that we mention any modifications that were made to standard instruments as this is important information required for the reader and potentially user of the data sets.

The direct interpretation of some data sets is required to highlight and emphasize the quality, usability, and accessibility of the data sets, however, we kept the descriptions to a minimum and any detailed analysis that people might want to report on in future research articles were not included. With our presentation of the data sets and the limited interpretation provided, we want to demonstrate to future users the potential of this valuable data set obtained in such a remote location. To limit the length of the paper, we do not show every data set that is available in the Zenodo repository. The data sets are provided in netCDF files following the CF conventions and are described in such a way that it should be relatively easy for any user to plot the data using a tool such as panoply. We put a lot of effort into providing well formatted netCDF files including detailed descriptions. netCDF files are a commonly known format in the scientific community.

To further help with the readability of the manuscript, a table of contents would be really useful as a reference for the future reader.

Thank you for this suggestion and we have now included a table of contents at the beginning of the revised manuscript.

Furthermore, the authors describe instruments which were taken onboard, but which didn't collect data due to malfunctions or errors.

All instruments that are described in our manuscript collected data, at least for part of the voyage. Some instruments didn't record data continuously for various reasons (which are described in the paper) and we lost the UAV during the voyage. Despite losing the UAV we successfully present the value of profile measurements in the paper by showing the data set that we obtained during the first flight.

However, we do note that we present underway SST and SSS measurements but failed to describe the instrumentation. We have updated the revised manuscript accordingly and added a brief description of the instrument (including accuracy) and measurements.
*The goal in relating this instrument information can be to further instruct scientists of lessons learned, and we think these instrument descriptions are worthwhile additions to the manuscript. However, we suggest that comments faulting inexperienced controllers should be omitted (in the spirit of sportsmanship) (for example lines 196-197).*

We understand the concerns of the reviewer in that our statement may impact other researchers who want to perform experimental field measurements with a high risk of failure because of the environmental conditions, and it might send the wrong message. We would like to mention the difficulties the operators were facing as otherwise it might be difficult to understand why the data weren't included/potentially not all that useful. Also note that the operator is a co-author on this paper and aware and supportive of our statement.

*We also suggest to the authors to group "unused" instrumentation to a separate section called "lessons learned" or more objectively, "instrument malfunctions". Nevertheless, we appreciated the transparency offered by the authors of the instrumentation malfunctions.*

We agree with the reviewer that it is important to mention any malfunctions and failures of instrumentation during any measurement campaign. However, we would prefer to mention the failures and malfunction of instrumentation in their respective sections to avoid repetition in any additional sections and to avoid extending the already very long paper further. We are not presenting any 'unused' equipment, as mentioned before, every instrument described here provides some measurements.

*We wondered whether the authors may have one or two suggestions to add to these sections in order for future readers to be better prepared for their own voyage (Figure 18 - is it useful if the data will not be used? Perhaps not worth plotting?)*

We think that data in Fig. 18 are useful despite the potential contamination by ship exhaust at certain levels. They provide a rare profile of aerosol concentration measurements in the surface layer of the atmosphere in the Southern Ocean and they

demonstrate the value of helikite and UAV measurements despite their challenges.

**Specific manuscript comments:**

Line 4: Could the authors comment on the direction of the "persistent biases"? An added qualification such as bias low or high would be more precise.

In the abstract we refrained from talking about positive/negative biases as it depends on what variable one is looking at (as described in the introduction). For example, climate models produce too little clouds over the Southern Ocean leading to an underestimation (negative bias) of the reflected solar radiation at the top of the atmosphere and an overestimation (positive bias) of the downwelling solar radiation at the ocean surface. As the details about the biases are described in the introduction, we would like to keep 'persistent biases' in the abstract without any indication about negative or positive, but we included an example of such a persistent bias in the abstract of the revised manuscript.

Line 8: According to the Earth System Science Data manuscript preparation and file submission: Ship names are italic, but their prefixes are roman (e.g. RV Polarstern).

Thank you for catching this mistake and we have corrected this in the revised manuscript.

Line 52. Reference format is different from the rest of the paper and should be double checked.

It is not clear to us what the reviewer is referring to here as we believe the formatting is correct. We followed the ESSD guidelines.

Line 77-78 Which cruise was the first to probe OCS concentrations and sources? Could the authors add the reference? Or indicate it in the table 1?

The first voyage that collected OCS measurements was described in Staubes and Georgii (1993) and we have now included this reference in the revised manuscript.

Reference:

Staubes, R. and Georgii, H.-W.: Biogenic sulfur compounds in seawater and the atmosphere of the antarctic region, Tellus, 45B, 127–137, 1993.

Table 1: Very useful for context and for future readers and data users! The authors could also add their own ship campaign at the bottom of the table with a "this work" reference.

We like this suggestion by the reviewers and have now added 'this work' in Table 1.

Line 86. Could the authors briefly explain their motivation for spending most of the voyage's time (30 days) south of 60° S? Does 60° S represent a reference point for some measurements for example?

The focus of the voyage was to conduct measurements in the Southern Ocean. The Southern Ocean comprises the southernmost waters of the World Ocean, generally taken to be south of 60° S latitude and encircling Antarctica; the lands of which are south of 60° S as defined in the Antarctic Treaty System. That is why we are referring to the region south of 60° S as the Southern Ocean (see line 528) and that's why the ship spent most of its time south of 60° S. We included an additional sentence to clarify this to the reader.

Lines 92-93. The authors mention the characterisation of radiation but list only the lidar, ceilometer and sky cameras. How was this parameter measured? Wasn't there a radiometer or pyranometer on board which should be added to this list?

Well spotted. While we didn't mention the instrumentation at line 92-93, radiation measurements are included and described in the paper (e.g. instrument description in Section 2.2, Line 118 and Section 4.1). The measurements are also provided together with all other meteorological variables from the DAS (Tangaroa Data Acquisition System). We have now included both radiometers and the AWS in Table 2 and adapted the sentence in question.

Lines 92-101. The authors mention seven research objectives (line 88) but only list six (92-101). Is there a goal missing? In addition, was there a priority within these goals?

We apologize for the confusion. Overall the Tangaroa Marine Environment and Ecosystem Voyage aimed at addressing seven key research objectives:

1. Physical oceanography

2. Aerosol-cloud interactions

3. Microbial planktonic communities

4. Seabed habitats and fauna

5. Cetacean studies

6. Zooplankton

7. Mesopelagic fauna

These research objectives are not listed or described in the manuscript, as we only want to focus on the measurements that were taken in support of one key research objective, i.e. '2. Aerosol-cloud interactions'. The items listed between line 92-101 represent the individual goals to be addressed using the measurements described in this paper and thereby addressing the overall research objective. We have clarified the wording in the revised manuscript - we are now referring to research objective and underlying research aims in the revised manuscript.

Lines 92-101: Give each goal a number to refer to these goals in each instrument section throughout the manuscript.

We have followed the reviewer's advice and numerated the research aims. Rather than referring to these research aims throughout the manuscript, we have added another column to Table 2 to connect the research aims with the instruments/measurements.

Table 2: Can the authors ensure that all the items in the column "location on the ship"in Table 2 are also written in Figure 3? For example, where is the fantail located?

That is a good point by the reviewer. We have now updated Figure 3 and named all locations of the instruments.

Line 117. How did the authors correct the wind speed according to the ship heading and speed?

The true wind speed and direction is calculated through vector-based correction performed by the Tangaroa DAS which uses ship speed and heading relative to the true north.

Also with (Popinet et al., 2004)? Please clarify and/or refer to the appropriate section.

We have now added material that points to the appropriate section and Figure in Popinet et al. and modified the text in the manuscript.

Line 166. Wind speed accuracy is stated as 5%. Can the authors comment on whether this accuracy is typical for this instrument? What was the wind direction accuracy?

The accuracy of the wind direction is determined by the accuracy of the GPS sensor and the frequency of received samples and therefore no number is provided here http://windsond.com/windsond_catalog_Feb2019.pdf. The information sheet about the instrument can be found here. The absolute sensor accuracy and resolution for this instrument type and as provided are typical characteristics at 25° C.

We added the following sentence: "The accuracy of the wind direction depends on the GPS conditions and is therefore determined by the accuracy of the GPS sensor."

Line 180-185. The authors' UAV's battery was drastically reduced due to low atmospheric temperatures. Which UAV model was used, which battery?

The UAV model was Swellpro Splash Drone 3 as mentioned in Table 2. We used the original batteries as well as additional 4S LiPo batteries with a capacity of 5200 mAh.

Section 3.3 – Particularly well written and structured section (see our comment on general flow and structure at the beginning of this review). Line 221. "The maximum range is 30km but the effective range was lower than that". Can the authors specific the effective range?

We have extended the description about the effective range statement for clarification.

Line 273: Can the authors link the use of the sky camera to one of their objectives?

The measurements obtained using the sky camera were made in support of research aim 1. We have now associated the research aims with the instrumentation in Table 2.

Line 280. Is the technique HDR? Mertens 2009 explicitly proposed exposure fusion as an alternative to HDR to produce a highâ ÌEAËĞRquality, low dynamic range image. Did they authors use the correct reference here?

We thank the reviewers for pointing that out and apologize for the confusion, the referral to HDR was incorrect, it should have been exposure fusioning. The Mertens 2009 reference is correct we have reworded the sentence in the revised manuscript.

Line 282. The ELIFAN algorithm crops the sky pictures to remove the distortion before estimating cloud fraction. Did the authors remove the distortion of the allskypi pictures? What was the field of view of the fisheye lens?

Yes, the image is cropped, and masks are applied for the ship structure, solar disk and horizon. We limit the field of view to a zenith angle of $70°$ ($20°$ above the horizon). The actual field of view of the lens is 180 degrees. Close to the horizon, cloud thresholding techniques do not perform well, and the generous mask described also excludes ship structure. We have added additional material to the revised manuscript.

Could the authors comment on the cloud cover uncertainty (for example in Figure 7)?

Cloud cover uncertainty, when expressed in oktas will be low (order of $\pm$ 0.5 oktas) depending on cloud type and observing conditions. As cloud fraction is typically used for

QA/QC of other measurements, it does not warrant a full uncertainty analysis. Further-more, a full uncertainty analysis would require a manual analysis of imagery obtained during the voyage, which is typically unpractical.

Line 291: Can the authors specify how rare clear-sky conditions were on their voyage?

Clear sky conditions are described later in text and in Section 4.1. Figure 7 provides additional information about clear sky conditions, showing that clear sky, as observed by allskypi, occurred less than 2% of the time during the voyage. We edit the text in the revised manuscript accordingly.

If the authors' goals were to study aerosol-cloud interactions, could the authors briefly comment on the value of the sun photometer measurement?

As reviewers note, this paper is looking at characterization of aerosol, clouds and their interactions. Measurements were made over a broad range of size and properties. One of the ways in which aerosol can be compared between different voyages is through the sun photometry done under the Aeronet Maritime Aerosol Network (MAN) programme where centrally calibrated Microtops photometers are used. In spite to the very few measurements possible on this voyage due to cloudiness, those measurements have value as part of the MAN database that allows for analysis of the spatial distribution of aerosol properties by latitude/region etc. for AOD (aerosol optical depth) and derived properties such and relative fraction of coarse and fine aerosol. Over 600 voyages, including this one, have contributed to the MAN database to date providing a valuable global resource for analyses, see e.g. Smirnov et al (2011, 2009) and use in validation and model development of important aerosol components such as oceanic sea-salt (Bian et al., (2019)).

References:

Smirnov, A., Holben, B. N., Giles, D. M. et al.: Maritime aerosol network as a com-ponent of AERONET – first results and comparison with global aerosol models and

satellite retrievals, Atmos. Meas. Tech., 4, 583–597, https://doi.org/10.5194/amt-4-583-2011, 2011.

Smirnov, A., et al. (2009), Maritime Aerosol Network as a component of Aerosol Robotic Network, J. Geophys. Res., 114, D06204, doi:10.1029/2008JD011257.

Bian, H., Froyd, K., Murphy, D. M., Dibb, J., Darmenov, A., Chin, M., Colarco, P. R., da Silva, A., Kucsera, T. L., Schill, G., Yu, H., Bui, P., Dollner, M., Weinzierl, B., and Smirnov, A.: Observationally constrained analysis of sea salt aerosol in the marine atmosphere, Atmos. Chem. Phys., 19, 10773–10785, https://doi.org/10.5194/acp-19-10773-2019, 2019.

Line 459: In section 3.10.1 title, the word chromatograth is misspelled.

We have corrected that spelling mistake.

Line 552. How was the % of cloud types calculated? Was this calculation performed by human observations, sky camera pictures and the ceilometer? Did the authors use an automatic algorithm to derive cloud types? Additional information would be useful to understand the data presented.

The cloud types were identified by human observations (weather_obs_level_0). While the original observations were not on regular intervals, in the revised manuscript we interpolated these observations at 6-hourly synoptic times (00, 06, 12, 18 UTC). The calculated fractions of cloud type were calculated from these records. We added additional information to the revised manuscript. We added the following material to the paper: "Synoptic weather observations were performed throughout the voyage and revealed that the most frequently observed cloud types were stratus..."

Section 4.2.4 (lines 673-693) has too much interpretation (not in the realm of ESSD's scope) and can be rewritten to specify the sample collection details, the operating procedure of the instrument, why this instrument was chosen and the data collected. The data collected should include the frozen fractions as well as the INP concentrations.

Thank you for this considered feedback. In the ESSD guidelines they suggest that "material required to understand the essential aspects of the paper such as experimental methods, data, and interpretation should preferably be included in the main text". Hence, the guidelines do indicate that interpretation is acceptable to "understand the essential aspects". However, we appreciate ours may be excessive and have reduced it to encompassing only the comparisons with other studies, and the note that the three samples between 40 and 50° S had recent terrestrial influences.

Sample collection details and the operating procedure of the Ice Spectrometer (IS) are already given in Section 3.9, but we have added some details there of the background correction procedure. With regard to explaining why we chose the sampling approach, and the IS, we have added some introductory text to the start of section 4.2.4.

To correct for contaminating INPs present on filters, we applied a correction using a regression from the combined results from three field blank filters. Since this regression was a parametric equation (in comparison with the non-parametric count data obtained from the IS) this prevents us from providing frozen fractions for this background-corrected data.

**Comments on Figures:**

During the discussion of this manuscript as a group, we gathered images of each instrument and found this process to be very helpful in visualizing the instruments. We can recommend to the authors to do the same, by adding pictures for each instrument to each section. (The authors can also contact us for these pictures (borduas@chem.ubc.ca and aragon@gamma.ttk.pte.hu , as we've gathered them already for our discussion.)

We thank the reviewers for that suggestion and we have now included some schematics. Adding images for all instruments is not always appropriate as some of the instruments are very complicated and the whole instrument doesn't fit into one single picture. Without any proper labels, adding only images might confuse the reader. We also note
that adding images would extend the length of the paper significantly.

Figure 2: The figure design can be improved for clarity by identifying the level of data analysis (raw vs calculated). The boxes can also be aligned for a cleaner figure.

Thank you for this suggestion, we have re-designed the figure for clarity.

Figure 3: Do the authors want to further describe the aerosol container lab? Or alternatively show a picture of the inside? The ship's exhaust should be highlighted as it is a big part of the discussion and of the data interpretation. The authors could also add a real picture of the ship (we had to google for a picture) for improved visualization.

The midship and aft exhaust have been highlighted in Figure 3 in the revised manuscript. We unfortunately do not have a useful reference picture of the inside of the container laboratory as not all instruments could be captured in frame. Instead, we've added a schematic layout of the particle counting instrumentation along with the relevant plumbing. We did not include a real image of the ship for copyright reasons and because an image of the ship can easily be obtained from google. Furthermore we added additional material to Section 3.8.

Figure 5: Along the left panel, the mean values can be added for clarity and readability. The numbering can be rethought, for example labels (b, d, f, h, i, l) could be removed.

We have removed the labels as suggested by the reviewers, however, we decided not to include mean values for clarity as adding mean values would make the figure even more busy and much of the data are not normally distributed as shown by the histogram plots. Here we only want to present an overview of the data.

Figure 6: This data is useful to highlight how the tropopause is shallower closer to the pole. With that point in mind, could the authors arrange the panels as a function of latitude instead of as a function of time of the voyage?

Good point by the reviewers and we have followed their suggestions and re-ordered the individual panels.

Figure 8: Can the authors add the number of points included in each boxplot? Can the authors also specify the values of the whiskers?

The number of data points included in the boxplots refers to the 1 minute data values within each latitude band during daylight (i.e. when the radiation was greater than 3 W m-2). The greatest amount of voyage time was spent in the -70o latitude band (-67.5o to -72.5o). The number of data points included in each box plot are:

Latitude band / data points -40 (1975), -45 (1818), -50 (2077), -55 (1812), -60 (3325), -65 (7837), -70 (16032), -75 (1629).

The box is presenting the interquartile range (Q1 to Q3), the whiskers are $\pm$ 1.5 * interquartile range, and outliers are shown as points. We have updated the figure caption to include the number of data points as suggested by the reviewer.

We weren't sure where the 0.86 coefficient for the atmospheric transmission coefficient came from; could the authors add a description and a reference? Finally, how do the authors explain values above one?

The visible radiation transmission coefficient was a simplified approach modelled with a single expected value for marine boundary-layer surface measurements taken as 0.86 (e.g. Longman et al (2012)) and verified by showing good correspondence of radiometer measurement against the modelled clear sky radiation expected at the ship for the chosen transmission coefficient, on the rare few clear-sky days around solar noon. Data were quality controlled, low sun angles (<3 Wm-2) and nighttime data were excluded. Values above 1 were rare but can occur when the solar beam is not obscured but clouds are present to increase the forward scattered component or through variation in the transmission coefficient with sun angle and aerosol loading that were not accounted for. We have updated the text in the revised manuscript and added a reference.

Reference:

Longman, R.J., Giambelluca, T.W., Frazier, A.G. (2012) Modeling clear-sky solar radiation across a range of elevations in Hawai'i: Comparing the use of input parameters at different temporal resolutions. Journal of Geophysical Research: Atmospheres, 117(D2). https://doi.org/10.1029/2011JD016388

Figure 9: We struggled to understand this figure and perhaps it can be made clearer. What are the bins representing? Could they be better depicted as a histogram/bargraph? Could the percentage and cumulative occurrences be displayed on two graphs? The x-axis at the top and bottom of the plot for percentage is different, maybe color coding the axis labels to the plot line could help. One of the plot lines doesn't show up in the legend.

We have updated and re-designed the figure to address the reviewers' concerns. Cumulative and percentage occurrence are now separated into two different plots, and bars used to aid clarity. The histogram bins are now clearly visible. We modified the text in the manuscript accordingly.

Figure 11. It would be worth adding a title to each plot to clarify the graph. The letter a of the plot a) has a smaller size than the other ones and Y-axis titles aren't aligned well. The colour bars should avoid white, otherwise the information cannot be seen particularly true for the vertical velocity plot (b)).

Thank you for pointing out the font size error. We have corrected this and also changed the colour scale for the vertical component of the velocity.

Figure 12: Small note that there is a blue dot on at x=0 value. Could the authors double check? Can the authors comment on how realistic a value of 520ppm of CO2 from an exhaust is?

Yes, this is a real data point and represents the first 5-min average from the Picarro for this voyage. The blue colour indicates that it has been classified as good data, i.e. the measurements were not contaminated by ship exhaust. The CO2 concentration at

the beginning of the voyage was higher (416.3 ppm) than the baseline value of 403 - 404 ppm observed subsequently. This is to be expected due to the close proximity to Wellington, which has many sources of CO2 from industry, traffic etc. at the beginning of the voyage.

A 5-min mean value of 520 ppm is entirely reasonable for ship exhaust mixed with background air. "Instantaneous" ( about 1 s) Picarro measurements are often observed in the range 500 - 700 ppm on Tangaroa voyages. An upper bound on what is possible is obviously the CO2 concentration in pure ship exhaust (i.e. un-mixed with background air). This depends very much on engine operating conditions, but a typical value is around 5% vol., = 50000 ppm.

Figure 13: Could the authors remove the graph lines to help clarity?

We prefer to leave the grid lines in Figure 13 as it helps the reader to identify the data associated with the measurement. We have increased the size of the figure.

Figure 14. In the figure caption, the figures should be labelled a, b, c, d and e. We can also recommend to the authors to add the name of each instrument along each panel for better readability. We recommend plotting the CCN data on a separate graph.

Thank you for spotting the mis-labeling, which we have now fixed. We also included the instrument names in the title of each panel. We prefer to leave the CCN data plotted here together with the other particle concentrations as measured by the other instruments to better identify any relationships between these.

Figure 15: We appreciated this figure to visualise the merger of the datasets on particle diameters and numbers. Thank you! Figure 17: Nice!

Thank you, we appreciate your positive feedback.

Figure 21: It would be useful to have titles on the plots themselves.

We followed the suggestion by the reviewer and labeled each panel individually.

Figure 22: This figure contains a lot of information and additional panels would help the clarity of this data.

We have updated the figure for clarity and split the figure into separate panels.

**Dataset and code availability comments:** We recommend to the authors to add photographs of each one of their instruments for improved visualization of the equipment used in this voyage.

See response above; where appropriate we have added schematics for some instruments where we seemed it to be appropriate.

Weather_obs_level_0→ What are the codes of weather types (1-4)?

The weather type classifies the cloud situation as observed into four different categories (see below). It may be of limited interest to external users. We note that there are no clear sky records in the human observations. This is because, by coincidence, either none of them were made during the time periods of clear sky, or because clear sky as detected by a ceilometer or sky camera only means clear sky within their field of view. For human observations, a clear sky would mean there are no clouds visible in any direction.

We now provide an explanation of the numbers in the additional note section of the data set on Zenodo. The weather types are defined as follows: 1. Low-level stratus cloud, 2. Precipitation associated with nimbostratus, 3. Low-level stratocumulus, 4. Mid-level altocumulus or altostratus.

The automatic weather station data appear to be complete and all information is available. Line 299. Why is the sun photometer data only found in the Maritime Aerosol Network? Is it possible to add it to the authors' Zenodo data set too?

We would prefer not to include the sun photometer data in our Zenodo archive as the data are openly accessible from MAN repository at the link provided.

Line 780. ALCF tool was downloaded, checked and confirmed after communication with one of the authors, Peter Kuma. The script now worked well. Thank you for sharing this resource!

You are very welcome.

Line 783. The authors provide the website for COARE gas exchange algorithm but in Table A1 in the "das" Data Acquisition System ReadMe_file the authors also provide the Matlab script to calculate fluxes. It might be worth mentioning this script in the Code Availability.

Good idea and we now mentioned the script in the Code Availability section.

We would also encourage the authors to explore the possibility of providing their data as an open API through https://developers.zenodo.org/

Thank you for that suggestion. We believe that the data set becomes available through the API automatically, but we will investigate that further.

We end this review by once again commanding the authors and scientists for their hard work and effort in gathering this dataset. We wish the authors all the best with their future data analyses and with addressing their scientific research goals.

Thank you very much and we appreciate your positive feedback.